# A brainstem–hypothalamus neuronal circuit reduces feeding upon heat exposure

Marco Benevento[1], Alán Alpár[2,3], Anna Gundacker[4], Leila Afjehi[5], Kira Balueva[6], Zsofia Hevesi[1], János Hanics[2,3], Sabah Rehman[1], Daniela D. Pollak[4], Gert Lubec[5], Peer Wulff[6], Vincent Prevot[7], Tamas L. Horvath[8] & Tibor Harkany[1,9] ✉

Empirical evidence suggests that heat exposure reduces food intake. However, the neurocircuit architecture and the signalling mechanisms that form an associative interface between sensory and metabolic modalities remain unknown, despite primary thermoceptive neurons in the pontine parabrachial nucleus becoming well characterized[1]. Tanycytes are a specialized cell type along the wall of the third ventricle[2] that bidirectionally transport hormones and signalling molecules between the brain's parenchyma and ventricular system[3–8]. Here we show that tanycytes are activated upon acute thermal challenge and are necessary to reduce food intake afterwards. Virus-mediated gene manipulation and circuit mapping showed that thermosensing glutamatergic neurons of the parabrachial nucleus innervate tanycytes either directly or through second-order hypothalamic neurons. Heat-dependent *Fos* expression in tanycytes suggested their ability to produce signalling molecules, including vascular endothelial growth factor A (VEGFA). Instead of discharging VEGFA into the cerebrospinal fluid for a systemic effect, VEGFA was released along the parenchymal processes of tanycytes in the arcuate nucleus. VEGFA then increased the spike threshold of *Flt1*-expressing dopamine and agouti-related peptide (*Agrp*)-containing neurons, thus priming net anorexigenic output. Indeed, both acute heat and the chemogenetic activation of glutamatergic parabrachial neurons at thermoneutrality reduced food intake for hours, in a manner that is sensitive to both *Vegfa* loss-of-function and blockage of vesicle-associated membrane protein 2 (VAMP2)-dependent exocytosis from tanycytes. Overall, we define a multimodal neurocircuit in which tanycytes link parabrachial sensory relay to the long-term enforcement of a metabolic code.

Transforming sensory information into coping mechanisms through either reflex circuits or conscious neural processes is critical for survival. Core body temperature is a crucial parameter that is continuously monitored and adjusted in endotherm species. Major strides have been made recently towards deconstructing the neural mechanisms that reduce body temperature in disease (that is, fever[1,9]). However, less is known about how the brain orchestrates physiological responses to dissipate excess heat upon acute thermal challenges through environmental exposure—whether this is unavoidable (for example, when leaving air-conditioned microenvironments in the summer) or elective (for example, when entering a sauna for health benefits)—to maintain its physiological optimum.

The pontine parabrachial nucleus (PBN) is a primary centre for temperature sensing, and contributes to optimizing core body temperature by increasing vasodilatation[10] and reducing physical activity[10,11], energy expenditure[10,11] and food intake[11]. To achieve its physiological goal, the PBN communicates with—among others—hypothalamic areas, which exert metabolic control over the entire body and adjust hormonal axes[1,12,13]. During the past decades, neuronal contingents that respond to and/or activate bodily responses at certain temperature ranges have been identified in the ventromedial preoptic area, preoptic area (POA) and dorsomedial hypothalamic nucleus (DMH), with their outputs linked to autonomic and behavioural phenotypes[9,11,14]. By contrast, how extrahypothalamic sensory neuronal inputs to the hypothalamus become encoded into metabolic commands and tune their set points by proportional associations remain largely unexplored.

As well as its neuroendocrine cell contingents and their circuitries, the hypothalamus contains tanycytes, which form a cellular interface between the parenchyma and the ventricular and vascular systems of the brain by delimiting the wall of the third ventricle[2]. Tanycytes

[1]Department of Molecular Neurosciences, Center for Brain Research, Medical University of Vienna, Vienna, Austria. [2]Department of Anatomy, Histology, and Embryology, Semmelweis University, Budapest, Hungary. [3]SE NAP Research Group of Experimental Neuroanatomy and Developmental Biology, Semmelweis University, Budapest, Hungary. [4]Department of Neurophysiology and Neuropharmacology, Center for Physiology and Pharmacology, Medical University of Vienna, Vienna, Austria. [5]Programme Proteomics, Paracelsus Medizinische Privatuniversität, Salzburg, Austria. [6]Institute of Physiology, Christian Albrechts University, Kiel, Germany. [7]University of Lille, INSERM, CHU Lille, Development and Plasticity of the Neuroendocrine Brain, Lille Neuroscience and Cognition, UMR S1172, EGID, Lille, France. [8]Department of Comparative Medicine, Yale University School of Medicine, New Haven, CT, USA. [9]Department of Neuroscience, Karolinska Institutet, Solna, Sweden. ✉e-mail: Tibor.Harkany@meduniwien.ac.at

are polarized, with their soma being in direct contact with the cerebrospinal fluid (CSF) and their single basal process extending far into the hypothalamic parenchyma and terminating around blood vessels and/or neurons[5]. Thus, tanycytes are poised to facilitate the bidirectional exchange of metabolites, hormones and signalling molecules between the central nervous system and the periphery[2,3,5,6,15–19]. Even so, it remains unresolved whether tanycytes are regulated indirectly or by direct innervation to release any specific molecules to exert systemic commands[3,7,8,15–21]. The innervation of tanycytes is supported by ultrastructural studies that identify 'synaptoid contacts'[22,23], presynapse-like elements enriched in synaptic vesicles, along the basal processes of tanycytes[22–25]. However, the neuronal origin of any such contact, and the functional significance thereof, remains unknown.

Here we addressed the exact mechanism that triggers an endocrine response to last for lengthy periods—particularly the restriction of food intake—upon acute temperature rise[9,14,26]. We show that temperature-responsive glutamatergic neurons of the PBN innervate tanycytes, with their action potential-dependent synaptic entrainment inducing the vesicular exocytosis of VEGFA along their basal process directly onto *Flt1*-positive (*Flt1*[+]) dopaminergic and *Agrp*[+] neurons of the arcuate nucleus[11,14,27] (ARC). Consequently, an anorexigenic phenotype manifests for hours, which can be rescued by the inactivation of PBN neurons, as well as attenuation of *Vegfa* expression and release in vivo. These data suggest that tanycytes translate extrahypothalamic sensory modalities into chemical codes to reset the output of hypothalamic neurocircuits through direct communication with neurons.

## Acute heat restricts food intake

Prolonged (4 h) heat exposure has been linked to significant metabolic changes in mice[11,14,28–30]. Here we measured food intake sequentially in both male and female mice kept at 25 °C (close to thermoneutrality[31]) and upon exposure to 40 °C for 1 h (Fig. 1a). Heat challenge significantly reduced food intake in both sexes over a 24 h period (Fig. 1b and Extended Data Fig. 1a). The diurnal pattern of locomotion (hypoactivity immediately after manipulation followed by transient hyperactivity; Extended Data Fig. 1b) and the body weight (Extended Data Fig. 1c) of the mice were also affected. As in earlier studies[11,14], the temperature of exposed skin (Fig. 1c), particularly above brown adipose tissue depots and in the perianal region (Extended Data Fig. 1d), increased sharply when exposed to 40 °C for 1 h, and then dissipated rapidly (within about 15 min) to baseline. These data suggest that reduced food consumption is a component of thermodefensive behaviours, which should depend on an interplay between thermoregulatory and feeding centres of the nervous system.

## Acute heat activates α-tanycytes

Thermal challenge could precipitate an immediate early gene response in hypothalamic neurons if the neurons directly participate in a neurocircuit linking heat exposure to feeding. To address this, we confirmed that exposure to 40 °C for 1 h induced cFOS expression in the PBN, the primary neuronal domain for temperature sensing (Fig. 1d). Next, we examined cFOS[+] cell contingents in the hypothalamus in mice of both sexes exposed to 40 °C for 1 h, with controls kept at 25 °C. In addition to neurons in the DMH and ARC (Extended Data Fig. 2), which are known to express cFOS in response to heat[27,32,33], we found unexpected increases in cFOS signal in α-tanycytes, and to a lesser extent in β-tanycytes, in both the rostral and caudal subdivisions of the ARC (Fig. 1e,f and Extended Data Figs. 1e and 2). Acute cold (4 °C for 1 h) did not induce cFOS in tanycytes (Extended Data Fig. 2a), suggesting a specific association of tanycyte activity with heat. Tanycytes also accumulated phosphorylated ERK1 and ERK2 (pERK1/2) after exposure to 40 °C (Extended Data Fig. 3a,b), consistent with their participation in a thermoregulatory mechanism.

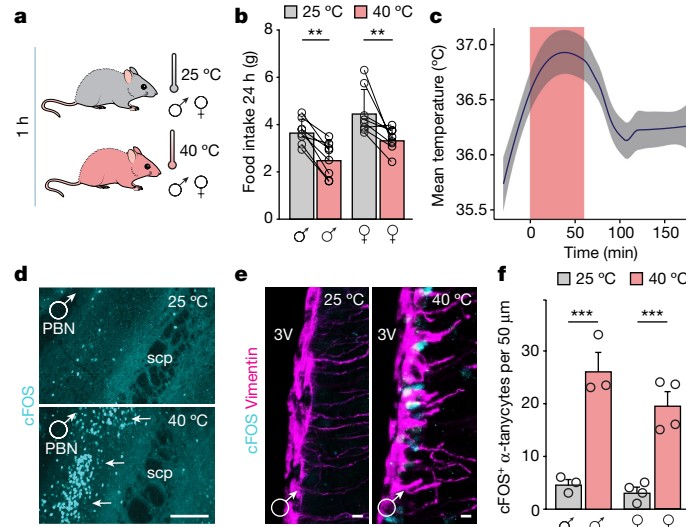

**Fig. 1 | Acute heat reduces food intake and selectively activates α-tanycytes.** **a**, Schematic of the experimental paradigm to link acute thermal challenge (1 h) to reduced food intake in wild-type male and female mice. **b**, Food intake in male and female mice during a 24-h period after exposure to 40 °C (1 h) versus continuity at 25 °C (*n* = 8 per sex). **c**, Core body temperature during (pink shaded region) and after thermal challenge. Data are presented as a facet-wrap plot with measurements taken at 15-min intervals (*n* = 8 male mice). **d**, Acute heat-induced cFOS expression in the PBN (arrows). scp, superior cerebellar peduncle. Scale bar, 120 µm. **e**, cFOS immunoreactivity in vimentin[+] α-tanycytes (at −2.30 mm relative to bregma) at the indicated temperatures (*n* = 3 male mice per group; data for female mice are in Extended Data Fig. 1e). Scale bars, 4 µm. **f**, The number of cFOS[+] α-tanycytes per 50 µm of ventricular surface in control versus heat-exposed mice of both sexes. Data are mean ± s.e.m., with circles (**b**,**f**) denoting individual data points. **b**,**c**,**f**, Detailed statistics are provided in Methods. *$P < 0.05$, **$P < 0.01$, ***$P < 0.001$.

## Parabrachial neurons innervate tanycytes

Ultrastructural data place synaptoid contacts in apposition to both α- and β-tanycytes[22–25,34,35]. However, neither the neurotransmitter content nor the functional significance of these structures is known. Using immunoelectron microscopy, we showed that synaptoid contacts along tanycytes are rich in vesicular glutamate transporter 2 (VGLUT2 (encoded by *Slc17a6*))-positive small vesicles (Fig. 2a). Their juxtaposition to the somata of tanycytes suggests that these structures, at least morphologically, could qualify as synapses.

Next, we sought to address the identity of the neurons that provide direct input onto hypothalamic tanycytes. We used virus-based transsynaptic labelling by delivering rAAV8-EF1a-mCherry-IRES-WGA-Cre[36] (for expression of mCherry and wheat germ agglutinin (WGA)–Cre fusion protein) in the lateral ventricle of either Ai14 or Tau-mGFP-loxP mice (Fig. 2b). Thus, the primary transduction site was marked by mCherry, and WGA–Cre was transported via synaptic transcytosis, resulting in Cre-dependent recombination in synaptically connected neurons. We found that WGA–Cre transduced vimentin[+] tanycytes to express mCherry 21 days later (Fig. 2c,d). WGA–Cre also induced recombination in hypothalamic neurons in the ARC (Fig. 2c,d), confirming that synaptoid contacts could be used as 'hitchhiking' vectors to reveal tanycyte-to-neuron connectivity locally. To test whether extrahypothalamic nuclei also innervate tanycytes, we injected retrograde AAVrg-CAG-GFP particles in the ARC area in mice (proximal to β-tanycytes; Extended Data Fig. 4a). We found significant labelling in the PBN (Fig. 2e and Extended Data Fig. 4a), consistent with prior anterograde tracing studies reporting rich projections from the PBN to the ARC, DMH and POA[1,10,13,14,28,29,37,38]. As well as providing information about the territorial distribution of PBN efferents, this enabled

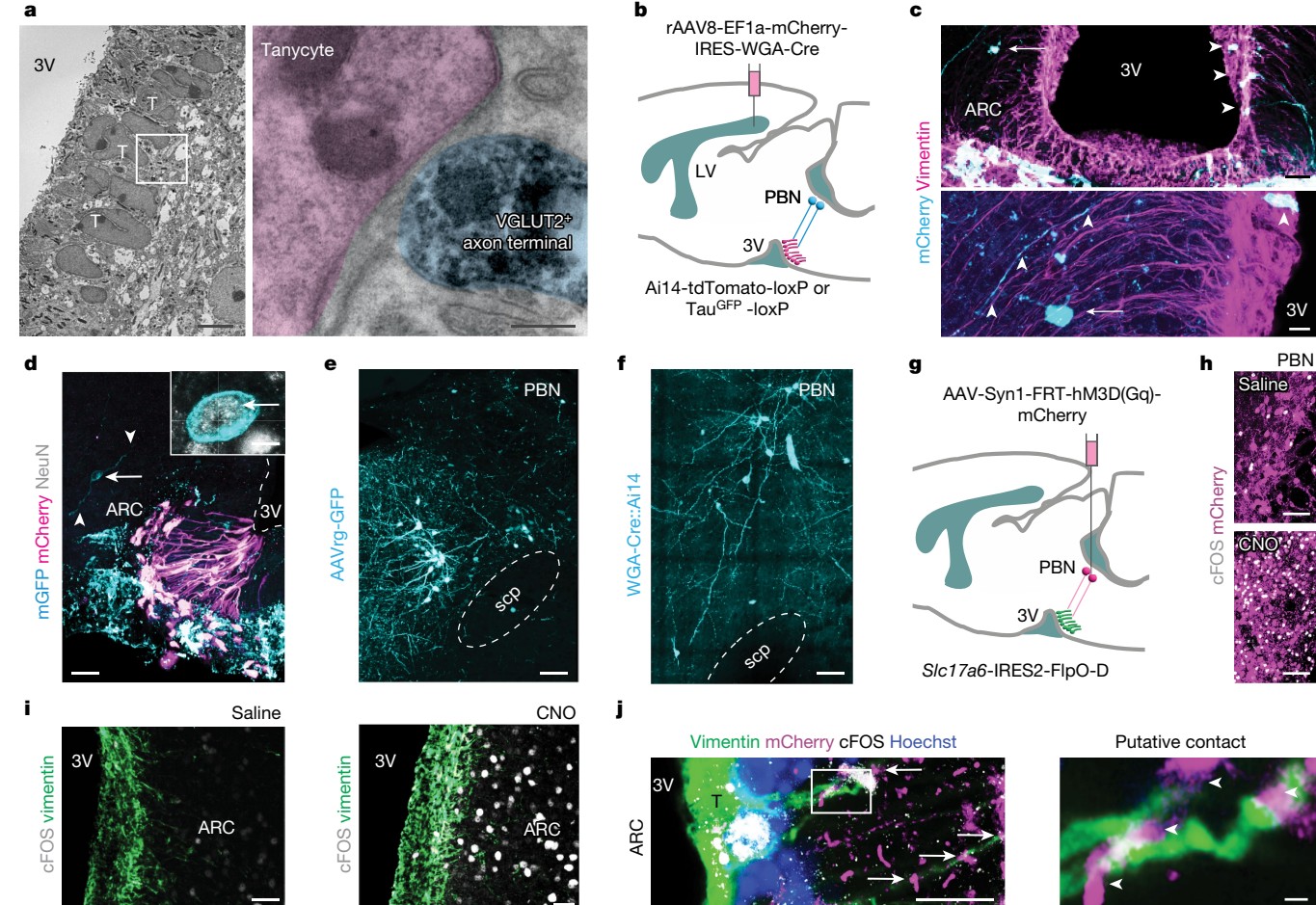

**Fig. 2 | Parabrachial neurons innervate α-tanycytes. a**, Left, VGLUT2⁺ presynapses (black) formed synapse-like contacts on tanycytes (T) along the third ventricle (3V). The outlined region is enlarged on the right. Right, VGLUT2⁺ presynapse (semi-transparent cyan) apposing the soma of a tanycyte (semi-transparent magenta). Scale bars: 2 μm (left), 200 nm (right). **b**, Schema of transsynaptic labelling. Tanycytes along the third ventricle served as the primary transduction site for WGA–Cre to mark synaptically connected neurons (cyan). LV, lateral ventricle. **c**, Top, transsynaptic labelling using rAAV8-EF1a-mCherry-IRES-WGA-Cre in Ai14-tdTomato-loxP mice. Vimentin⁺ tanycytes co-expressed mCherry (arrowheads) and were synaptically connected, among others, to local neurons (arrows). Bottom, putative contact between a vimentin⁺ tanycyte process (arrowheads) and an mCherry⁺ transsynaptically labelled neuron (arrow, cyan). Scale bars: 20 μm (top), 6 μm (bottom). **d**, Transsynaptic labelling by rAAV8-EF1a-mCherry-IRES-WGA-Cre in Tau^mGFP-loxP mice. mCherry⁺ tanycytes were innervated by local NeuN⁺ neurons (mGFP, colour-coded to cyan, arrows in both overview and inset). Arrowheads denote neuronal processes. Scale bars: 40 μm (main image), 2 μm (inset). **e**, GFP⁺ parabrachial neurons labelled with AAVrg-CAG-GFP injected at the level of ARC. Scale bar, 50 μm. **f**, Transsynaptic labelling by rAAV8-EF1a-mCherry-IRES-WGA-Cre in Ai14 mice. Parabrachial neurons (tdTomato⁺, cyan) synaptically connected to tanycytes (see also **c**) resided laterally to the scp. Scale bar, 50 μm. **g**, Cartoon of the chemogenetic manipulation of parabrachial glutamatergic neurons in *Slc17a6*-IRES-FlpO-D mice. **h**, Transduced neurons expressed mCherry. cFOS⁺ neurons are shown 1.5 h after either saline or CNO (5 mg kg⁻¹, intraperitoneal) injection. Scale bars, 50 μm. **i**, cFOS activation (grey) in α-tanycytes (green) 1.5 h after administration of saline or CNO. Scale bars, 10 μm. **j**, Left, mCherry⁺ parabrachial efferents (magenta) from glutamatergic neurons apposed (arrows) vimentin⁺Hoechst 33,421⁺ tanycytes (composite; outlined region is enlarged on the right) co-expressing cFOS. Right, mCherry⁺ parabrachial efferents (arrowheads) apposed vimentin⁺ processes. Scale bars: 8 μm (left), 1 μm (right).

us to specify tanycytes as cellular targets. In doing so, we found a subset of PBN neurons that were recombined after tanycyte-driven labelling by WGA–Cre (Fig. 2f and Extended Data Fig. 4b,c). These data identify tanycytes as a probable circuit motif for projection neurons of the PBN.

## Excitatory afferents activate tanycytes

The PBN encapsulates many subtypes of glutamatergic neurons, some of which relay thermal stimuli to the hypothalamus[1,11,14,37,39]. We explored whether tanycytes are activated when exciting the PBN. We thus injected viruses encoding excitatory designer receptors exclusively activated by designer drugs (DREADDs) in the PBN of *Slc17a6*-IRES2-FlpO-D mice to selectively drive the activity of glutamatergic PBN neurons (Fig. 2g), and used cFOS as a surrogate of neuronal as well as tanycyte activation in response to clozapine-*N*-oxide (CNO). CNO significantly upregulated cFOS expression in both *Slc17a6*⁺mCherry⁺ neurons in the PBN (Fig. 2h and Extended Data Fig. 4d) and in α-tanycytes (Fig. 2i and Extended Data Fig. 5a) embedded in mCherry⁺ projections along the wall of the third ventricle (Extended Data Fig. 4e). cFOS⁺ α-tanycytes were more numerous in the caudal ARC, a pattern reminiscent of our results after acute heat exposure (Fig. 2i), and were positioned proximal to mCherry⁺ presynapse-like boutons in *Slc17a6*-IRES2-FlpO-D mice (Fig. 2j), providing histochemical support to a PBN-to-tanycyte circuit arrangement. These results suggest that α-tanycytes could respond to excitatory inputs from the PBN.

## Tanycytes respond to synaptic excitation

Even if ependymocytes that line the dorsolateral portion of the third ventricle can respond to glutamatergic innervation[38], neither the density of glutamatergic inputs to ventral tanycytes nor the biophysical consequence of their excitation is known. Because WGA–Cre rAAV8 particles only produced sparse labelling of tanycytes (Fig. 2c,d), we used immunohistochemistry for VGLUT2 and vimentin to determine the density of VGLUT2+ inputs onto topographically subclassified α- and β-tanycytes[2]. We reconstructed the vimentin+ processes of tanycytes at rostrocaudal positions of −1.94 mm and −2.30 mm (relative to bregma; Extended Data Fig. 6a) together with the number of VGLUT2+ terminals within less than 0.5 μm distance of these structures (Extended Data Fig. 6b). More than 50% of all tanycytes−irrespective of their subtypes−received 2 or 3 VGLUT2+ terminals on average (Extended Data Fig. 6c,d). However, when α-tanycytes and β-tanycytes were separated and pooled, we found a significantly higher density of VGLUT2+ inputs onto α-tanycytes (Extended Data Fig. 6e). These data suggest that tanycytes are synaptically modulated.

Single-cell RNA sequencing showed the expression of genes encoding α-amino-3-hydroxy-5-methyl-4-isoxazolepropionic acid (AMPA) receptor (AMPAR) subunits[17,40,41], including *Grm1–Grm8*, *Grin1–Grin3a* and *Gria1–Gria4*[21] in tanycytes (see also Extended Data Fig. 8b). Here, we first histochemically detected GluA1 (encoded by *Gria1*) and GluA2 (encoded by *Gria2*) subunits, which positionally segregated along the wall of the third ventricle: GluA2 preferentially labelled α-tanycytes, whereas GluA1 marked both α- and β-tanycytes (Fig. 3a). Next, we used *Rax*-CreER[T2]::Ai14 mice 4–7 days after tamoxifen-induced recombination to show that VGLUT2+ presynapses juxtaposed GluA2+ stretches in the basal processes of α-tanycytes (Extended Data Fig. 6f).

Next, patch-clamp electrophysiology showed that both α- and β-tanycytes lacked active membrane properties upon step depolarization (Extended Data Fig. 6g), and the resting membrane potential, membrane resistance, capacitance and electrotonic current dissipation coefficient did not differ between their subtypes (Extended Data Fig. 6h–k). To functionally characterize glutamatergic inputs, we recorded spontaneous excitatory postsynaptic currents (sEPSCs). At a holding potential of −70 mV, both populations of tanycytes had sEPSCs (Fig. 3b), noting that the amplitude of sEPSCs was significantly higher in β-tanycytes and that the recorded tanycytes co-expressed GluA2 subunits. Moreover, s-AMPA (100 μM) evoked a negative tonic current that returned to baseline upon wash-out (Extended Data Fig. 6l). These data allow for the hypothesis that tanycytes could respond to synaptic stimulation.

We used ex vivo Ca²⁺ imaging to test whether direct neuronal stimulation could trigger Ca²⁺ signals in tanycytes, noting that *Adarb1* (also known as *Adar2*) was not expressed[40], thus *Gria2*-containing AMPARs in tanycytes are unlikely to be Q/R edited and remain Ca²⁺ permeable. To retain ex vivo brain slice integrity, we probed neurons positioned proximal to and innervating tanycytes (Fig. 2c,d), and performed paired recordings to determine whether tanycytes undergo action potential-dependent activation. When using *Rax*-CreER[T2]::PC-G5-tdTomato mice to record Ca²⁺ transients by monitoring the intensity of GCaMP5g, a genetically encoded Ca²⁺ indicator, in tdTomato+ tanycytes (Fig. 3c), we found that action potentials evoked in neurons triggered Ca²⁺ transients in tanycytes (Fig. 3d and Supplementary Video 1). Ca²⁺ waves initially propagated through a single filament and then spread to adjacent tanycytes within 300 ms after the last of 8 action potentials delivered at 25–30 Hz (Fig. 3d). Post hoc reconstruction of biocytin-filled neurons confirmed that their axon indeed innervated the tanycytes (Fig. 3e). We then superfused NBQX (20 μM), an AMPAR antagonist, which invariably occluded action potential-evoked postsynaptic Ca²⁺ transients in tanycytes (Extended Data Fig. 7a and Supplementary Videos 2 and 3). These data suggest that the glutamatergic innervation of tanycytes induces their

depolarization, which can even spread across tanycytes connected by gap junctions[16,42,43]. We have reinforced these observations by showing that AMPA (100 μM)-induced Ca²⁺ transients were abolished when NBQX (20 μM) was also present (Extended Data Fig. 7b and Supplementary Videos 4–6). Next, we applied picrotoxin (100 μM) to block type A γ-aminobutyric acid (GABA_A) receptors, thus also increasing Ca²⁺ transients. This response was sensitive to 5 μM tetrodotoxin (TTX), a drug that impedes action potential-dependent excitatory neurotransmission by blocking Na+ voltage-gated channels (Extended Data Fig. 7c and Supplementary Videos 7–9). KCl (50 mM), used to indiscriminately depolarize excitable cells both in vitro and in vivo[44,45], also evoked Ca²⁺ waves in tanycytes (Extended Data Fig. 7d and Supplementary Videos 10 and 11). These data suggest that tanycytes can respond to neuronal excitation.

## Monosynaptic PBN inputs onto tanycytes

We also tested whether tanycytes respond to monosynaptic inputs from glutamatergic neurons of the PBN. AAV1-CAG-FLEXFRT-ChR2(H134R)-mCherry particles were bilaterally delivered into the PBN of *Slc17a6*-IRES2-FlpO-D mice, enabling Flp-dependent expression of ChR2(H134R)-mCherry in VGLUT2+ neurons (Fig. 3f, left). Twenty-one days later, mCherry+ axons were found coursing in the vicinity of the wall of the third ventricle (Fig. 3f, middle), and even contacted tanycytes directly (Fig. 3f, right). Next, tanycytes were held at −70 mV in whole-cell configuration ex vivo, and superfused with 1 μM TTX and 100 μM 4-aminopyridine to maximize the Ca²⁺-dependent depolarization[46] of ChR2-containing presynapses. Subsequent exposure to 470-nm light pulses (50 ms each) induced excitatory postsynaptic currents (EPSCs) in tanycytes (Fig. 3g, left), which occurred 134.8 ± 24.4 ms after stimulus onset, with amplitudes of 7.0 ± 0.3 pA (Fig. 3g and Extended Data Fig. 6m), and a failure rate of 77.0 ± 2.3% (Extended Data Fig. 6m). These data corroborated both the temporal dynamics and the size of tanycyte responses measured by Ca²⁺ signalling and suggested that glutamatergic neurons of the PBN directly innervate tanycytes.

## VEGFA production upon thermal challenge

Subsequently, we aimed to identify the factors that tanycytes can produce when stimulated. First, we interrogated open-label single-cell RNA-sequencing data[40] with a focus on neuropeptides and signalling proteins that are preferentially expressed in tanycytes. We found that *Rax+Col23a1+* tanycytes expressed *Vegfa*, *Tgfb3*, *Tgfb2*, *Fgf10* and *Pdgfa* (Extended Data Fig. 8a). We focused on *Vegfa* because earlier studies showed that it increases capillary fenestration in the ARC when secreted from tanycytes upon fasting[6]. Similarly, hyperthermia[47–49] and exercise[50] can up-regulate *Vegfa* expression, but with unknown cellular foci. Fluorescence in situ hybridization (FISH) revealed that exposure to 40 °C triggered *Vegfa* mRNA expression in α-tanycytes (in both sexes; Fig. 4a,b and Extended Data Fig. 9a). These data were confirmed by quantitative PCR in microdissected tissues of the ventrolateral segment of the wall of the third ventricle (Extended Data Fig. 9a). Similarly, increased VEGFA protein content in tanycytes was found in mice acutely exposed to 40 °C (compared with 25 °C; Extended Data Fig. 9b), and after chemogenetic activation of excitatory neurons in the PBN of *Slc17a6*-IRES2-FlpO-D mice (Extended Data Fig. 9c). Thus, tanycytes could produce VEGFA in response to both acute heat and the selective activation of *Slc17a6*+ projection neurons of the PBN in mice.

## Unidirectional VEGFA release

On the basis of previous data[50], we hypothesized that VEGFA might be released into the CSF for volumetric transport to distant brain regions. To test this mode of action, we exposed adult rats to either

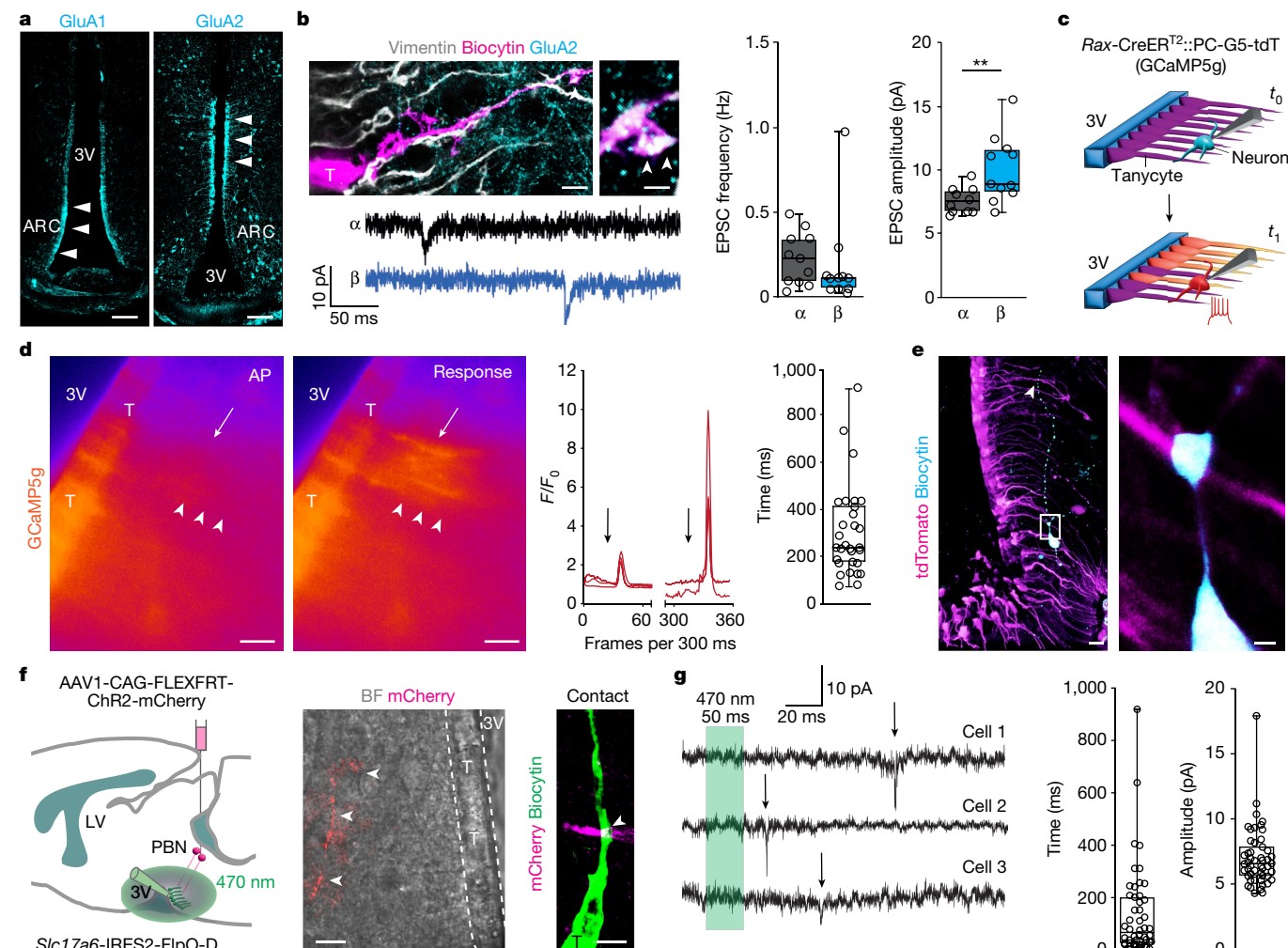

**Fig. 3 | Tanycyte responses to synaptic afferent modulation. a**, GluA1 and GluA2 in tanycytes. Scale bars, 100 μm. **b**, Top left, biocytin-filled, vimentin⁺ tanycyte (T) with GluA2 expression (main image; scale bar, 5 μm), with enlarged view showing GluA2 in a vimentin⁺ basal filament (arrowheads; scale bar, 800 nm). Frequency (middle) and amplitude (right) of sEPSCs recorded in α- and β-tanycytes (both $n = 11$ cells). Detailed statistics are presented in Methods. **c**, Schema of ex vivo experiments. Top, configuration at test ($t_0$). Bottom, action potentials in neurons evoked at $t_1$. **d**, Left, Ca²⁺ transients in tanycytes (arrowheads) in response to an evoked action potential (AP) of 30 pA per 100 ms (arrow) (Supplementary Video 1). Middle, relative fluorescence intensity for GCaMP5g ($F/F_0$) in tanycytes upon action potential induction in neurons (arrows). Right, time lag of GCaMP5g relative fluorescence after the last action potential (trains of 8 action potentials; $305.2 \pm 35.25$ ms, $n = 30$ tanycytes, $n = 6$ experiments). Scale bars, 20 μm. **e**, Left, a biocytin-filled neuron in the ARC of a *Rax*-CreER^T2^::PC-G5-tdT mouse. Right, intersection between a biocytin⁺ neuronal process (cyan) and a tanycyte (tdTomato⁺, magenta) in the outlined region in the left image. Scale bars: 20 μm (left); 2 μm (right). **f**, Left, cartoon showing tanycytes (green) tested for optogenetically induced EPSCs by stimulating PBN efferents (ChR2–mCherry, red) with 50-ms pulses of 470-nm light. Middle, bright-field (BF) view of tanycytes along the third ventricle overlaid on an mCherry⁺ afferent (scale bar, 10 μm). Right, a putative intersection between a tanycyte and afferent (scale bar, 2 μm). **g**, Left, optically induced EPSCs (arrows) in tanycytes. Time lag (middle; $256.5 \pm 34.66$ ms) and amplitude (right; $6.759 \pm 0.48$ pA; $n = 29$ EPSCs from $n = 7$ tanycytes, $n = 4$ independent experiments). **b**,**d**,**g**, In box plots, the centre line is the median, box edges delineate top and bottom quartiles, whiskers extend to minimum and maximum values and circles depict individual data points.

25 °C or 40 °C for 1 h, and aspirated CSF from their cerebellomedullar cistern immediately after thermal manipulation. Conspicuously, acute heat did not significantly increase the amount of VEGFA in the CSF (Extended Data Fig. 9d). These data seem incompatible with a role for VEGFA in ventricular volume transmission. Alternatively, VEGFA could be released along the basal process of tanycytes to modulate neuronal activity locally in the ARC. This arrangement could be reminiscent of VEGFA modulating neuronal plasticity in the hippocampus by reducing neuronal excitability[51] when hyperpolarizing the inactivation threshold of voltage-gated Na⁺ channels[52]. Light and electron microscopy documented tanycyte processes in close apposition (less than 1 μm) to both *Th*⁺ and *Agrp*⁺ neurons in the ARC (Fig. 4c and Extended Data Fig. 9e,f). Furthermore, both *Th*⁺ and *Agrp*⁺ neurons expressed *Flt1*, the primary VEGFA receptor, in the ARC (Fig. 4d and Extended Data Fig. 9g,h).

By contrast, *Flt1* expression was not detected in *Pomc*⁺ neurons (Extended Data Fig. 9i). These data suggest that VEGFA release could modulate orexigenic neurons in the ARC.

Next, we tested whether tanycyte-derived VEGFA could alter neuronal activity by temperature switching in brain slices ex vivo[53]. We took advantage of our mass spectrometry data showing that primary tanycytes expressed thermosensitive Ca²⁺-permeable TRPV2 channels (Extended Data Table 1; see also refs. 54,55). Thus, increasing the temperature of the superfusate from 25 °C to 38 °C could evoke heat-dependent VEGFA release from tanycytes. The voltage threshold to generate spontaneous action potentials by ARC neurons became significantly increased ($-33.66 \pm 0.70$ mV at 25 °C versus $-29.21 \pm 1.98$ mV at 38 °C; Fig. 4e). We then used axitinib (40 μM), a VEGF receptor antagonist[6], which occluded the change in action potential threshold

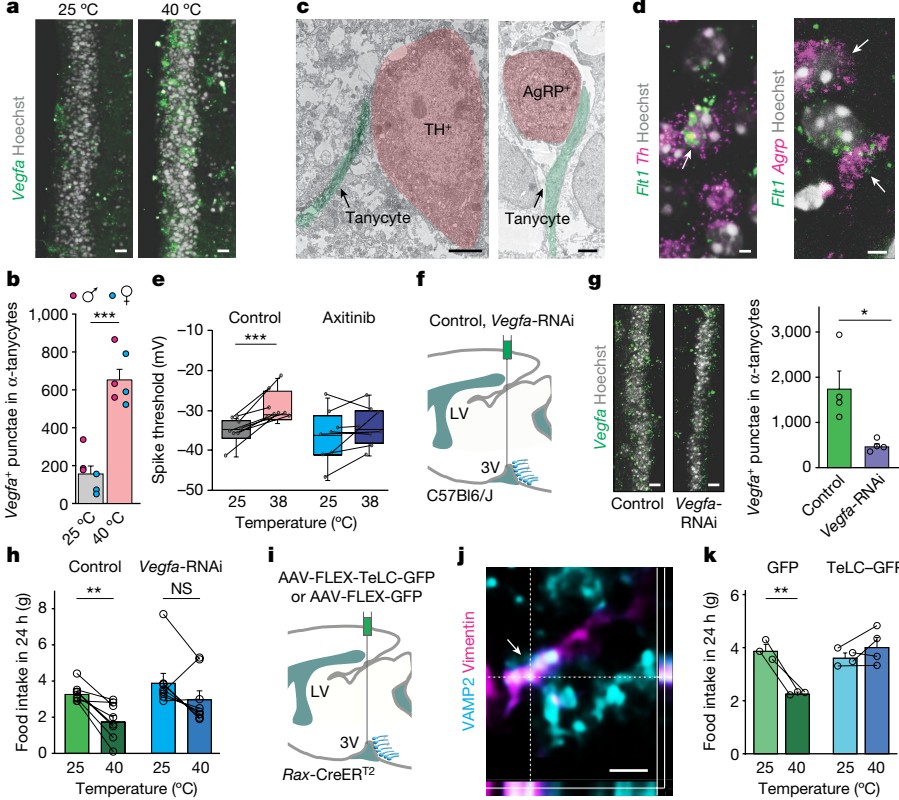

**Fig. 4 | Tanycytes produce VEGFA upon heat exposure of mice. a**, *Vegfa* mRNA (green precipitate) in α1-tanycytes of mice at 25 °C and 40 °C. Scale bars, 5 μm. **b**, Quantification of *Vegfa* punctae in α1-tanycytes. **c**, Basal processes of tanycytes (green overlay) apposed the perikarya of both TH⁺ (left) and AgRP⁺ neurons (right). Scale bars, 2 μm. **d**, Co-localization (arrows) of *Flt1* and either *Th* (left) or *Agrp* (right) in the ARC. Scale bars, 3 μm. **e**, Threshold of spontaneous action potentials in neurons sequentially recorded at 25 °C and 38 °C in artificial CSF (ACSF) alone (control) or ACSF supplemented with axitinib (40 μM; *n* = 8 pairs). In box plots, the centre line is the median, box edges indicate interquartile ranges, and whiskers extend to minimum and maximum values. **f**, Experimental design for infusion of scrambled RNAi (control) or *Vegfa*-RNAi in the third ventricle. **g**, Left, *Vegfa* mRNA (green precipitate) in α-tanycytes in control or after *Vegfa*-RNAi infusion. Right, quantification of *Vegfa* punctae in α-tanycytes (*n* = 4 mice per condition). Scale bars, 5 μm. **h**, Food intake during a 24-h period after acute thermal challenge (1 h) with pre-treatments as indicated (*n* = 8 mice per group). **i**, Schema of the experiment with AAV-FLEX-GFP and AAV-FLEX-TeLC-GFP virus particles infused in the third ventricle of *Rax*-CreER^T2 mice. **j**, Orthogonal image stack showing VAMP2 along a vimentin⁺ process. Scale bar, 500 nm. **k**, Temperature switching reduced food intake in control mice expressing only GFP in tanycytes, whereas TeLC–GFP expression in tanycytes abolished the temperature sensitivity of food intake (*n* = 4 mice per condition). **b,g,h,k**, Data are mean ± s.e.m. **b,e,h**, Individual data points are displayed as circles. Sections were routinely counterstained with Hoechst 33,421. **e,g,h,k**, Detailed statistics are presented in Methods. NS, not significant.

(−34.71 ± 2.81 mV at 25 °C versus −32.77 ± 1.42 mV at 38 °C; Fig. 4e). These results suggest that VEGFA might reduce the excitability of neurons proximal to tanycytes.

Nevertheless, the presence of TRPV2 channels in tanycytes allows the alternative hypothesis that these cells could be thermosensing per se, and themselves inhibit food intake (noting that TRPV2 channels are maximally activated by noxious heat (52 °C or more), with limited Ca²⁺ permeability in the warm temperature range (around 40 °C)). Therefore, we first exposed mice to 38 °C, which is typically below the activation threshold of TRPV family channels. Food intake was still significantly reduced (Extended Data Fig. 9j). Moreover, tranilast, a brain-permeant TRPV2 antagonist (at 20 mg kg⁻¹), did not affect food intake at either 25 °C or 38 °C (Extended Data Fig. 9j). These results suggest that tanycytes entrained by PBN neurons generate anorexigenic signalling in the hypothalamus upon thermal challenge in vivo.

## Inhibition of VEGFA rescues food intake

We knocked down expression of *Vegfa* mRNA in tanycytes by delivering small interfering RNA (siRNA) targeting *Vegfa* (*Vegfa*-RNAi) into the third ventricle of adult mice to test its effect on food intake (Fig. 4f). *Vegfa*-RNAi reduced the amount of *Vegfa* mRNA and VEGFA protein eight days after delivery (Fig. 4g and Extended Data Fig. 10a). This approach did not affect food intake or body weight for five days prior to thermal manipulation (Extended Data Fig. 10b,c). Next, we exposed mice to 25 °C for 1 h and then to 40 °C a day later. Acute heat significantly reduced food intake in mice that had received non-targeting siRNAs (controls) during 24 h post-induction (Fig. 4h). However, *Vegfa*-RNAi attenuated the heat-induced reduction in food intake (Fig. 4h). Neither siRNA affected either the body weight (Extended Data Fig. 10d) or locomotion (Extended Data Fig. 10e) following acute heat. We then injected Cre-dependent AAV2-FLEX-TeLC-GFP or AAV2-FLEX-GFP (control) viruses in the third ventricle of *Rax*-CreER^T2 mice to show that VEGFA released from tanycytes is required to suppress food intake upon acute thermal manipulation (Fig. 4i and Extended Data Fig. 10f). This approach took advantage of the expression of VAMP2 in cultured tanycytes (Extended Data Table 1) and in vivo (Fig. 4j and Extended Data Figs. 8b and 10g), and VAMP2 cleavage by tetanus toxin light chain (TeLC), thus preventing vesicular exocytosis[56,57]. When exposing mice expressing GFP alone to 25 °C and then 40 °C, acute heat significantly reduced food intake (Fig. 4k). By contrast, TeLC–GFP occluded the heat-induced suppression of food intake (Fig. 4k). These data implicate VEGFA release from tanycytes in inhibiting food intake after exposure to acute heat.

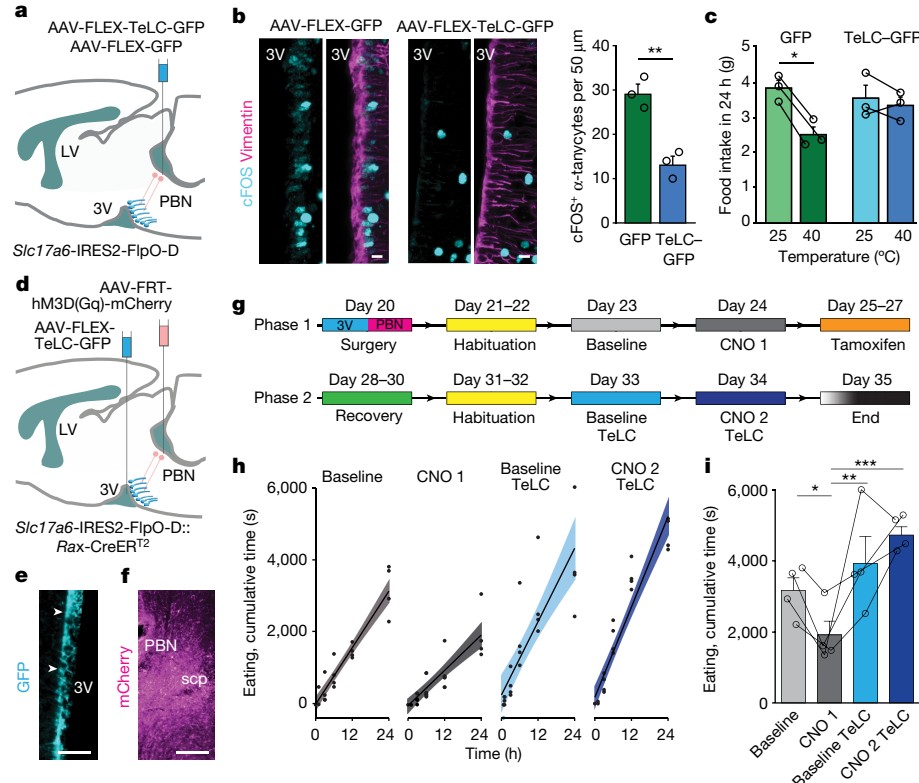

**Fig. 5 | Tanycytes link parabrachial activity to feeding. a**, Cartoon showing a TeLC-based strategy to inhibit glutamate release. **b**, Left, exposure to 40 °C (1 h) resulted in cFOS⁺ tanycytes in mice that had received control viruses (AAV-FLEX-GFP). By contrast, cFOS⁻ (inactive) tanycytes were mostly seen after thermal challenge of mice treated with AAV-FLEX-TeLC-GFP. Right, quantification of cFOS⁺ tanycytes. **c**, Inactivation of glutamatergic PBN neurons by TeLC attenuated the sensitivity of food intake to acute heat. **d**, Schema of the intersectional genetic approach used to simultaneously modulate VGLUT2⁺ PBN neurons and RAX⁺ tanycytes. **e**, TeLC–GFP-expressing hypothalamic tanycytes. **f**, hM3D(G_q)-mCherry-expressing *Slc17a6*⁺ neurons in the PBN.

**e,f**, Scale bars, 50 μm. **g**, Experimental timeline. **h**, Facet-wrap timeline plot showing the cumulative eating time during baseline (24 h), CNO exposure (3 mg kg⁻¹; CNO 1), baseline to TeLC (recombined with 4-hydroxytamoxifen), and AAV2-FLEX-TeLC-GFP recombined in tanycytes and injected with CNO (3 mg kg⁻¹; CNO 2 TeLC). Data are means ± s.e.m., solid circles show individual data points. **i**, Cumulative eating time in 24 h. Group designations and colouring correspond to those in **h**. **b,c,i**, Data are means ± 95% confidence interval of the s.d. (*n* = 3–4 per group), with dots representing individual data points. Detailed statistics are provided in Methods.

## Tanycytes link PBN to feeding

The activation of PBN neurons upon thermosensation could suppress feeding[11], with our data suggesting a PBN–tanycyte–orexigenic ARC neurocircuit as its hardwired underpinning. To reinforce this circuit arrangement, we first tested whether the PBN provides sufficient input for the activation of tanycytes when exposed to heat acutely. To this end, the PBN of *Slc17a6*-IRES2-FlpO-D mice was targeted by either AAV2-FlpON-GFP (control) or AAV2-FlpON-TeLC-GFP viruses to block VAMP2-mediated presynaptic neurotransmitter release by selectively expressing TeLC in *Slc17a6*⁺ PBN neurons (Fig. 5a). Acute heat induced quasi-equivalent cFOS expression in PBN neurons of *Slc17a6*-IRES2-FlpO-D mice injected with either AAV2-FlpON-GFP or AAV2-FlpON-TeLC-GFP viruses (655 ± 46 cFOS⁺ neurons (GFP) versus 698 ± 76 cFOS⁺ neurons (TeLC–GFP) in *n* = 3 mice per group, *P* = 0.660; Extended Data Fig. 10h), suggesting that the expression of either construct did not alter their ability to respond to upstream thermal inputs. Tanycytes in *Slc17a6*-IRES2-FlpO-D mice injected with AAV2-FlpON-GFP in the PBN also expressed cFOS (Fig. 5b). By contrast, acute heat provoked significantly less cFOS expression in α-tanycytes in *Slc17a6*-IRES2-FlpO-D mice that had received AAV2-FlpON-TeLC-GFP in the PBN (Fig. 5b). These data suggest the reliance of hypothalamic tanycytes on excitatory PBN afferents for activation upon acute heat exposure. Coincidentally, mice expressing only GFP, but not those expressing TeLC–GFP, had reduced food intake when exposed to 40 °C

for 1 h (Fig. 5c), suggesting that the activation of PBN efferents is necessary to reduce food intake.

Finally, we set out to test whether the activation of glutamatergic PBN input onto tanycytes is sufficient to trigger the tanycyte-dependent reduction in food intake. We generated *Slc17a6*-IRES2-FlpO-D::*Rax*-CreER^T2 mice for the independent targeting of excitatory PBN neurons (*Flp*) and tanycytes (*cre*) by the injection of AAV2-EF1a-FRT-hM3D(G_q)-mCherry (into PBN) and AAV2-FLEX-TeLC-GFP (into third ventricle), respectively (Fig. 5d). We expected this strategy to lead to the simultaneous activation of glutamatergic PBN neurons and the inactivation of tanycytes (Fig. 5e,f). We sequentially tested mice by CNO administration prior to and then after 4-hydroxytamoxifen-mediated Cre-dependent TeLC recombination (Fig. 5g), three weeks after viral delivery in mice kept at thermoneutrality[31] (29 °C), with ad libitum access to food and water. The body weight of the mice did not vary significantly throughout, although a partial and transient reduction was observed upon the chemogenetic activation of PBN neurons (Extended Data Fig. 10i). We found a statistically significant reduction in the time component the mice had spent eating following CNO-dependent activation of glutamatergic neurons in the PBN, whereas tanycyte functions were left intact (Fig. 5h,i). However, when TeLC recombination blocked the exocytosis of any substance from tanycytes, the reduction in the time the mice had spent eating upon CNO-induced PBN activation was no longer affected (Fig. 5h,i). TeLC recombination alone in tanycytes did not alter eating time. Drinking and locomotion upon PBN

activation were not dependent on tanycytes (Extended Data Fig. 10j,k). Overall, these results suggest that tanycytes suppress food intake acutely upon their feed-forward activation by glutamatergic neurons of the PBN.

## Discussion

Our findings suggest that tanycytes constitute a novel circuital node to mediate defensive metabolic responses upon acute heat exposure. In doing so, tanycytes are directly modulated by glutamatergic long-range projections from the PBN. Our analysis is conceptually novel because it links the direct activation of a subset of topologically defined glutamatergic synapses on tanycytes to the directional release of a bioactive molecule, VEGFA, to modulating orexigenic neurons and thus, reducing food intake.

Tanycyte processes concentrate in the medial-to-central extent of the ARC. This anatomical arrangement means that the release of VEGFA can effectively impinge upon both $Th^+$ and $Agrp^+$ neurons. In turn, the lack of $Flt1$ expression in $Pomc^+$ neurons provides selectivity to this process, regardless of this cell population being contacted by tanycytes[2], and explains the shift towards net anorexigenic output from the ARC. Although only about half of the tanycytes received excitatory inputs, efficacious amplification steps, particularly gap junction coupling[42,43], can propagate $Ca^{2+}$ signalling[42], and thus release events, in tanycyte clusters encompassing 1 to 60 cells[43]. Such spreading activation might explain why cFOS expression is a feature of only α-tanycytes, poised to act as 'starter cells', whereas pERK1/2 is broadly distributed in both α- and β-tanycytes. Thus, the number of excitatory synapses formed by glutamatergic neurons on tanycytes seems sufficient to activate a large enough cluster of tanycytes for body-wide effects to take place. It is noteworthy that only a subset of PBN neurons serves as a source of this innervation. As yet, the molecular identity of these thermosensory neurons remains largely obscure, beyond their co-expression of prodynorphin[1,11,14], among the many glutamatergic neuronal subtypes in the PBN. Even if we emphasize monosynaptic inputs of extrahypothalamic origin to tanycytes, our retrograde tracing suggests that intrahypothalamic neurons could function as relays to transduce PBN-derived activity. Thus, our data are compatible with earlier findings showing thermosensitive neurons in the POA, DMH and ventromedial hypothalamic nucleus[11,14,27], which could act as second-order 'amplifiers' or 'transducers' of PBN output onto tanycytes. In sum, and also considering that the effect of tanycytes is phase-locked, directional and local, multiple upstream neuronal modules are poised to safeguard the fidelity of this sensory-to-metabolic switch. This is conceptually important, since earlier studies extensively described neuronal determinants of thermodefensive mechanisms[9,11,14]. Thereby, our study on a fundamental physiological mechanism by ventral tanycytes forms a counterpart to the activity of dorsal ependymocytes and their responses to POA neurons when body temperature is increased in disease[9].

A central element of our analysis is that both acute heat and the chemogenetic and/or optogenetic activation of glutamatergic PBN neurons can prime tanycytes to release signalling molecules into the ARC. The secretion of bioactive molecules by tanycytes and/or ependymocytes has been demonstrated previously, including 2-arachydonoyl glycerol[17] (an endocannabinoid) and ciliary neurotrophic factor[38]. Nevertheless, these events were coupled to volume transmission to affect distant parvocellular hypothalamic neurons[17] or midbrain neurons[38]. VEGFA has been implicated in changing the permeability of the blood–brain barrier[6,58]. Nevertheless, a focal action of VEGFA in the hypothalamus has not been investigated. The combination of RNAi and TeLC overexpression strategies accommodate the hypothesis for directional VEGFA action upon its vesicular exocytosis on hypothalamic neurons. Once released, tanycyte-derived VEGFA could act on $Flt1$ expressed by both $Agrp^+$ and $Th^+$ neurons, the latter being an upstream reinforcer of orexigenic pressure[59]. We suggest that VEGFA can increase the action

potential threshold of—for example, $Th^+$ neurons—resulting in an increased action potential failure rate due to a rise in the hyperpolarization threshold of $Na^+$ channels[52]. Cumulatively, we suggest that the net output of the food intake circuit upon heat exposure shifts towards reducing food intake because VEGFA directly suppresses its $Agrp^+$ and $Th^+$ components and could even reduce the local dopaminergic inhibition of its $Flt1^-Pomc^+$ contingent[59].

Overall, our study identifies tanycytes as integrative cellular foci in the nervous system of mice linking a primary sensory modality to metabolic changes, thus prioritizing the need of the body to cope with brief thermal challenges, which is one of the most frequent environmental exposures.

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

# Methods

## Mice

Experimental procedures on mice conformed to the 2010/63/EU directive and were approved by the Austrian Ministry of Education, Science and Research (66.009/0145-WF/II/3b/2014 and 66.009/0277-WF/V3b/2017). All procedures were planned to reduce suffering, as well as mouse numbers. Mice were kept under standard housing conditions (12 h:12 h reverse light:dark cycle with light on at 22:00 and off at 10:00, 25 °C), with food and water available ad libitum. For acute thermal manipulations, ex vivo electrophysiology, neuroanatomy, and behavioural tests, C57Bl6/J mice were used. $Rax^{tm1.1(cre/ERT2)Sbls}$/J mice ($Rax$-CreER$^{T2}$; JAX 025521) were crossed with $B6.Cg$-$Gt(ROSA)26Sor^{tm14(CAG\text{-}tdTomato)Hze}$/J (referred to as Ai14; JAX 007914), $B6;129S6$-$Polr2a^{Tn(pb\text{-}CAG\text{-}GCaMP5g,\text{-}tdTomato)Turd}$/J (PC-G5-tdT; JAX 024477) or $B6;129S$-$Slc17a6^{tm1.1(flpo)Hze}$/J ($Slc17a6$-IRES2-FlpO-D; JAX 030212) at ages between postnatal days (P)60-90. $B6;129P2$-$Mapt^{tm2Arbr/J}$ (referred to as $Tau^{mGFP}$-loxP; JAX 021162) were used for transsynaptic labelling. $Agrp^+$ neurons were visualized by crossing $Agrp^{tm1(cre)Lowl}$/J (JAX 012899) and Ai14 reporter mice. Mice of both sexes were used for experiments, as indicated. For primary cultures of tanycytes, both male and female Wistar rats were used. To sample the CSF, male Wistar rats were cannulated, as approved by the Ethical Review Board of Semmelweis University (PE/EA/1234-3/2017, Hungary).

## Acute heat exposure

C57Bl6/J mice of both sexes aged P60–P100 were housed individually and habituated in an Aria BIO-C36 EVO incubator (Tecniplast) at 25 °C with a reverse 12 h:12 h light:dark cycle (light on at 22:00) with 42% humidity for 3 days. One day before acute thermal challenge, the temperature of an MIR-254 incubator (Sanyo) was set to the relevant target temperature. To maintain humidity in the incubator, a Becher glass filled with 1 l water was placed in the incubator. Humidity (~42%) and $CO_2$ levels (~396 ppm) were continuously measured with a CO100 $CO_2$ monitor (EXTECH Instruments). At 09:00 on day 4 (that is, 1 h before the beginning of the dark (off) phase of the light cycle), the mice were placed in new experimental cages without food and water, exposed to 25 °C for 1 h, and then returned to their home cages. On day 5 (09:00), mice were again placed in experimental cages without food and water, and then exposed to either 4 °C or 40 °C for 1 h. Subsequently, the mice were returned to their cages in an Aria BIO-C36 EVO incubator (Tecniplast) set at 25 °C.

To record skin temperature, C57Bl6/J male mice were singly housed, with their interscapular area above the main brown fat depot shaved 2–3 days prior to the experiments. Mice were then exposed to 40 °C for 1 h. Control mice were kept at 25 °C. Body temperature was recorded at both the interscapular area and the perianal region of each mouse using an infrared thermometer[60,61] (DET-306, Femometer). Baseline temperature was acquired 15 min prior to the thermal challenge, followed by switching them to a thermo-controlled chamber (Memmert, MEMM-OT3007S) set to 40 °C, and left undisturbed for 1 h. Temperature recordings resumed at intervals of 15 min for another 180 min after heat exposure ended, with the mice returned to their home cages.

## Measurement of food intake and body weight

Food pellets and mice were measured on an Entris II Essential line scale with 0.01 g accuracy (Sartorius, 1000059011) to determine food intake and body weight, respectively. Baseline parameters were determined 1 h prior to thermal manipulation. In select experiments, food pellets were weighed 2 h (12:00), 4 h (14:00) and 24 h (09:00) after thermal challenge.

In multiparametric experiments (Fig. 5h,i and Extended Data Fig. 10i–k), food and fluid intake, as well as horizontal movement were simultaneously recorded by using PhenoTyper cages (Noldus). Herein, food intake was approximated by recording the time spent to consume food when the infrared beam within the pellet dispenser was interrupted by the nose-pokes of the mice ($\Delta t$). The same technical setup was used to measure the time spent to drink. Data were analysed by Ethovision XT15 (Noldus).

## Immunohistochemistry

For immunofluorescence labelling, mice were anaesthetized with isoflurane and transcardially perfused with ice-cold phosphate buffer (PB) (0.1 M, pH 7.4) followed by ice-cold paraformaldehyde (4% in 0.1 M PB). Subsequently, the brains were removed and kept in the same fixative at 4 °C overnight. Next, the brains were washed with 0.1 M PB and stored with 0.025% NaN$_3$ as antifungal agent at 4 °C until processing. Fifty-micrometre-thick coronal sections spanning the ARC and PBN were cut on a vibratome (V1000S; Leica) in 0.02 M tris-buffered saline (TBS). Free-floating sections were stored in 0.02 M TBS supplemented with 0.025% NaN$_3$ at 4 °C. To produce 30-μm glass-mounted sections, brains were cryoprotected in 0.1 M PB containing 30% sucrose and 0.025% NaN$_3$. Then, brains were flash-frozen in liquid N$_2$, and embedded in optimal cutting temperature embedding matrix (OCT, Tissue-Tek). Coronal sections were cut on a cryostat microtome (CryoStar NX70; Thermo Scientific). Brain sections were washed in 0.02 M TBS, then blocked with a solution containing 5% normal donkey serum, 2% bovine serum albumin (BSA, Sigma Aldrich), 0.3% Triton X-100 in 0.02 M TBS at 22–24 °C for 2 h. Select combinations of primary antibodies were used as follows: guinea pig anti-cFOS (1:1,000; Synaptic Systems, 226005), rabbit anti-cFOS (1:2,000; Synaptic Systems, 226003), rabbit anti-DsRed (1:200; Clontech/Takara, 632496), rabbit anti-RFP (biotinylated, 1:1,000; Rockland, 600-406-379), chicken anti-RFP (1:500; Rockland, 600-901-379), goat anti-GFP (1:200; Abcam, ab6662), goat anti-mCherry (1:500; Antibodies Online, ABIN1440058), guinea pig anti-GluA1 (1:100; Alomone Labs, AGP-009), rabbit anti-GluA2 (1:100; Alomone Labs, AGC-005), chicken anti-NeuN (1:500; Millipore, ABN91), rabbit anti-p44/42 MAPK (pERK1/2$^{Thr202/Tyr204}$; 1:200; Cell Signaling Technology, 9101S), rabbit anti-TH (1:500; Millipore, AB152), goat anti-VEGFA (1:100; R&D Systems, AF-493-NA), guinea pig anti-VGLUT2 (1:200; Synaptic Systems, 135404), rabbit anti-VGLUT2 (1:500; Synaptic Systems, 135403), and chicken anti-vimentin (1:500; Synaptic Systems, 172006). Cocktails of the antibodies were incubated on an orbital shaker in 0.02 M TBS to which 2% normal donkey serum, 0.1% BSA, 0.3% Triton X-100 and 0.025% NaN$_3$ had been added at 4 °C for 3–4 days. Secondary antibodies included: Alexa Fluor 488 donkey anti-rabbit IgG (1:2,000; Invitrogen, AB21206), Alexa Fluor 488-conjugated AffiniPure donkey anti-guinea pig IgG (1:300; Jackson ImmunoResearch, 706-545-148), Alexa Fluor 488-conjugated AffiniPure donkey anti-mouse IgG (1:300; Jackson ImmunoResearch, 715-545-151), Alexa Fluor 647-conjugated AffiniPure donkey anti-rabbit IgG (1:300; Jackson ImmunoResearch, 711-605-152), Cy2-conjugated AffiniPure donkey anti-goat IgG (1:300; Jackson ImmunoResearch, 705-225-147), Cy2-conjugated AffiniPure donkey anti-rabbit IgG (1:300; Jackson ImmunoResearch, 711-225-152), Cy3-conjugated AffiniPure donkey anti-chicken IgG (1:300; Jackson ImmunoResearch, 703-165-155), Cy3-conjugated AffiniPure donkey anti-guinea pig IgG (1:300; Jackson ImmunoResearch, 706-165-148), Cy3-conjugated AffiniPure donkey anti-rabbit IgG (1:300; Jackson ImmunoResearch, 711-165-152), Cy5-conjugated AffiniPure donkey anti-chicken IgG (1:300; Jackson ImmunoResearch, 703-175-155), Cy5-conjugated AffiniPure donkey anti-guinea pig IgG (1:300; Jackson ImmunoResearch, 706-175-148) and Cy5-conjugated streptavidin (1:200; Jackson ImmunoResearch, 016-170-084). Secondary antibodies were applied in 0.02 M TBS containing 2% BSA, 0.3% Triton X-100, and Hoechst 33,342 (1:10,000; Sigma Aldrich, used as nuclear counterstain, B2261) on an orbital shaker at 22–24 °C for 2 h. After washing in 0.02 M TBS, sections were glass-mounted and coverslipped with an antifade solution consisting of 10% Mowiol (Sigma, 81381), 26% glycerol (Sigma, G7757), 0.2M Tris buffer (pH 8.0) and 2.5% Dabco (Sigma, D27802). Ex vivo brain slices (250–300 μm) after patch-clamp recordings were

cleared in an ascending series of glycerol (25%, 50%, 80% and 100% for 1 h each, and 100% overnight), and mounted with the same antifading solution as above.

## Chromogenic histochemistry and electron microscopy for VGLUT2

To localize VGLUT2 in the periventricular area, samples were prepared as previously published[38]. In brief, mice ($n = 4$) were transcardially perfused with ice-cold 0.1 M PB (20 ml), followed by 4% PFA and 0.1% glutaraldehyde (GA) in 0.1 M PB. Sections were washed three times in 0.1 M PB. Endogenous peroxidase activity was blocked by treating the sections with 1% $H_2O_2$ for 10 min. Next, sections were blocked (see 'Immunohistochemistry') and immunolabelled with a rabbit anti-VGLUT2 antibody (1:1,000; a gift from M. Watanabe)[62] and incubated at 4 °C for 2 days to reveal presynaptic terminals in apposition to tanycytes. Following repeated washes in 0.1 M PB, sections were exposed to biotinylated anti-rabbit secondary antibody (Vector Labs BA-1000) at 22–24 °C for 2 h. Next, sections were washed in 0.1 M PB and incubated with pre-formed avidin–biotin–peroxidase complexes (ABC Elite; Vector Laboratories) at 4 °C overnight. Thereafter, sections were osmificated, dehydrated, embedded in durcupan (Fluka, ACM), and cut at 60 nm on an Ultracut UCT microtome (Leica). Imaging was performed on a Transmission Electron Microscope FEI Tecnai 10 (100kV) equipped with a TEM side-mounted camera (EMSIS MegaView III G3).

## Electron microscopy for vimentin, TH and tdTomato

Male C57Bl6/N mice ($n = 3$) were used for vimentin plus TH immunostaining. Mice were perfused with a fixative containing 4% PFA, 15% picric acid (by volume) and 0.08% GA in 0.1 M PB. Tissue was post-fixed overnight in GA-free fixative, then washed in PB. Sections containing intact ARC were kept in 10% sucrose in 0.1 M PB for 30 min and 20% sucrose in 0.1 M PB for 1 h. The sections were rapidly freeze/thawed (3×), washed (3×) with 0.1 M PB, and double-stained with chicken anti-vimentin antibody (1:1,000; Sigma in goat blocking serum) and mouse anti-TH antibody (1:3,500 Sigma) on a shaker at 4 °C for 48 h. After repeated washes in PB, sections were incubated for 1.5 h in biotinylated goat anti-mouse and biotinylated goat anti-chicken IgG (1:200 each in goat blocking serum; Vector Labs) at 22–24 °C. Sections were then washed (3×) and incubated in ABC complex (1:100 in PB; ABC Elite kit, Vector Labs) at 22–24 °C for 1.5 h. The immunoreaction was visualized with 3,3-diaminobenzidine (DAB), then extensively washed. *Agrp*-Cre::Ai14 mice were perfused as above, and carried through the same procedures as above but the sections were incubated in chicken anti-RFP antibody (1:2,000; Rockland) at 4 °C for 48 h. This was followed by biotinylated goat anti-chicken IgG, then ABC (both for 1.5 h) to visualize tdTomato+ (*Agrp*-Cre) neurons. Following the DAB reaction, sections were osmificated (1% $OsO_4$ in 0.1 M PB) for 30 min, washed in PB followed by double-distilled $H_2O$, and 50% ethanol. Sections were kept in 1% uranyl acetate in 70% ethanol for 1 h, washed in 95% and 100% ethanol, washed (2×) in propylene oxide, and left in a solution of 50% propylene oxide and 50% durcupan for 3 h. Sections were left in pure durcupan overnight, flat-embedded on liquid release-coated slides, coverslipped with Aclar (Electron Microscopy Sciences), glued and trimmed. Sections were collected on Formvar-coated single slot copper grids and imaged using a Philips Tecnai T-12 Biotwin electron microscope.

## Fluorescence in situ hybridization

PFA-fixed 30-μm glass-mounted sections were used for FISH. We followed the HCR 3.0 protocol for 'generic sample on slide' per the manufacturer's recommendations (Molecular Instruments; https://files.molecularinstruments.com/MI-Protocol-RNAFISH-FrozenTissue-Rev2.pdf) with *Agrp*, *Flt1*, *Pomc*, *Th* and *Vegfa* probes. In brief, slides were defrosted and gradually dehydrated in an ascending ethanol gradient (50%, 70%, 100%) for 5 min each at 22–24 °C. Tissue samples were then hybridized by incubation with 1.2 μl of 1 μM stock of each probe (1.2 pmol) in a humid chamber at 37 °C overnight. Excess probe was washed with warm washing buffer (37 °C) mixed with 5× SSCT buffer (that is, sodium chloride/sodium citrate (5× SCC) and 0.1% Tween 20; Sigma Aldrich, 9005-64-5) at scaled composition (75% washing buffer/25% 5× SSCT; 50% washing buffer/50% 5× SSCT; 25% washing buffer/75% 5× SSCT; 100% 5× SSCT) for 15 min each at 37 °C. Next, 2 μl of amplifiers (hairpins) were diluted (from 3 μM stock) in 100 μl amplification buffer and applied to the samples in a humid chamber at 22–24 °C for 12 h. Thereafter, slides were washed in 5× SSCT buffer. Nuclei were counterstained with Hoechst 33,342 (1:10,000; Sigma Aldrich, B2261) diluted in 5× SSCT at 22–24 °C for 15 min. After another wash with 5× SSCT, the samples were coverslipped with an antifade solution made up of 10% Mowiol (Sigma, 81381), 26% glycerol (Sigma, G7757), 0.2 M Tris buffer (pH 8.0), and 2.5% Dabco (Sigma, D27802).

## Confocal and epifluorescence imaging

Confocal micrographs were acquired on Zeiss LSM710, LSM880/Airyscan or Zeiss LSM900/Airyscan 2 setups. We used a Zeiss AXIO Observer ApoTome.2 platform for epifluorescence microscopy. The number of VGLUT2+ presynapses contacting vimentin+ tanycytes were determined by using a Zeiss LSM880/Airyscan microscope equipped with a Plan-Apochromat 63×/1.4 NA oil objective (Zeiss). We separately acquired 2 × 2 tile scans covering each tanycyte subcategory in coronal brain sections at both −1.94 mm and −2.30 mm relative to bregma. Orthogonal z-stacks were acquired at a depth of 25 μm. Images to quantify the intensity of pERK1/2 were captured on an LSM880 microscope equipped with a Plan-Apochromat 25×/0.8 Imm Korr DIC M27 objective (Zeiss). Images showing complementary GluA2 and VGLUT2 signals within individual synapses were captured on a Zeiss LSM900/Airyscan 2 microscope equipped with a Plan-Apochromat 40×/1.4 NA oil objective.

## Image analysis

Confocal images were loaded in either Imaris 9.0.2 (Biplane) or Fiji 1.52e (https://imagej.net/Fiji).

**Mapping of VGLUT2+ presynaptic terminals in apposition to tanycytes.** α1-, α2-, β1- and β2-tanycytes (all vimentin+) were separately captured at −1.94 mm and −2.30 mm relative to bregma, and at a tissue depth of 25 μm (z-scan) on a Zeiss LSM880 microscope with their images loaded in Imaris x64 9.0.2 later (Bitplane). Tanycyte filaments were reconstructed along their vimentin signal using the built-in extension 'Filament tracer'. First, we determined the thickness of the basal process on x, y and z axes (~1 μm). Subsequently, we traced these basal processes by using the 'Autopath' method, and by setting the seeding point on the soma of each tanycyte separately. Next, tracing was centred, smoothed, and adjusted to a diameter of 1 μm. To quantify and to reconstruct the VGLUT2 signal in putative presynapses, we first set their diameter to <0.5 μm. Subsequently, we isolated any such VGLUT2 signal with the built-in 'Spots' extension to reconstruct spheres. We then used a 'find spots close to filaments' Imaris XTension to quantify the density and distribution of those VGLUT2+ presynapses (spots) that apposed vimentin+ tanycyte processes (filaments). The maximal accepted distance from the spot centre (VGLUT2+) to the filament edge (vimentin+) was set to <0.5 μm. Thus, the total number of spots within 0.5 μm was used for statistical analysis.

**cFOS in tanycytes and neurons.** To quantify the number of tanycytes activated by acute thermal manipulation in C57Bl6/J mice of both sexes or after chemogenetically activating glutamate inputs in *B6;129S-Slc17a6^{tm1.1(flpo)Hze}/J* mice bilaterally injected with either AAV-EF1a-FRT-hM3D($G_q$)-mCherry or AAV2/1-Syn-FRT-hM3D($G_q$)-mCherry virus particles, we counted the absolute number of cFOS+ nuclei both in vimentin+ tanycytes along the wall of the third ventricle, and in mCherry+ neurons in the PBN per section from confocal micrographs at a tissue depth of 25 μm (z-scans).

**Intensity analysis for pERK1/2 and VEGFA.** Five-by-three tiled confocal images over the cross-section of the third ventricle were acquired on a Zeiss LSM880 microscope at an image depth of 8 bit. Confocal micrographs were loaded in Fiji 1.52e, and their signal intensity for either pERK1/2 or VEGFA was quantified in pre-defined tanycyte subgroups in male mice kept at either 25 °C or 40 °C. Images were acquired at identical settings (including laser power output, digital gain/offset) to allow for comparisons be made on signal intensities between the experimental groups.

***Vegfa* expression and localization.** Confocal images of *Vegfa* mRNA (FISH) from brains of both control and heat-exposed C57Bl6/J male mice that had received scrambled RNAi or *Vegfa*-targeting RNAi cocktails in the third ventricle were acquired on a Zeiss LSM710 microscope as 2 × 5 image tiles. Thus, the entire length of the ventricular wall was imaged as a *z*-stack of ~25 μm. We reconstructed the wall of the third ventricle with the 'Surface' method (over a nuclear signal), thus limiting data collection to only the perikarya of tanycytes. To quantify the number of *Vegfa* mRNA precipitates in tanycytes, images were loaded in Imaris (Bitplane) with the *Vegfa* signal in the somata of tanycytes transformed into spots with a maximal diameter of <0.5 μm. Then, the number of spots (*Vegfa*) that had been in close apposition to the surface was determined by the 'Find spots close to surface' Imaris XTension (threshold set to 1 unit) and used for statistical analysis.

## Chemogenetic induction of PBN projections onto tanycytes

To test whether tanycytes are directly activated by long-range glutamatergic projections, the PBN of *B6;129S-Slc17a6^{tm1.1(flpo)Hze}/J* was bilaterally injected with AAV-EF1a-FRT-hM3D(G_q)-mCherry or AAV2-Syn1-FRT-hM3D(G_q)-mCherry particles. Twenty-one days after virus delivery, mice were moved to an incubator (Tecniplast, Aria BIO-C36 EVO) set at 25 °C with a reverse 12 h:12 h light:dark cycle for 24 h. The following day, mice were injected intraperitoneally with either sterile physiological saline or CNO (5 mg kg⁻¹; Tocris, 6329) dissolved in saline. After 1.5 h, mice were transcardially perfused with 0.1 M PB followed by ice-cold 4% PFA for histochemistry.

## RNA isolation from the wall of the third ventricle wall and quantitative PCR

Two groups of P60-P90 C57Bl6/J male mice (*n* = 4 per group) were acutely exposed to 40 °C for 1 h and compared to mice kept at 25 °C. Their brains were rapidly removed, and 1-mm coronal brain slices were cut by using a steel brain matrix (Stoelting, 51386). The wall of the third ventricle was manually dissected, flash-frozen in liquid N₂, and stored at −80 °C until processing. RNA was extracted with the RNeasy mini kit (Qiagen, 74536). To eliminate genomic DNA, samples were treated with DNase I. Thereafter, RNA was reverse transcribed to cDNA with the high-capacity cDNA reverse transcription kit (Applied Biosystems, 4368814). Quantitative real-time PCR was performed (CFX-connect, Bio-Rad) with primer pairs as follows: mouse *Vegfa* (forward: 5′-gaggggaggaagagaaggaa-3′, reverse: 5′-ctcctctcccttctggaacc-3′) and mouse glyceraldehyde-3-phosphate dehydrogenase (*Gapdh*; forward: 5′-aactttggcattgtggaagg-3′, reverse: 5′-acacattgggggtaggaaca-3′), which were designed with the NCBI Primer Blast software. Quantitative analysis of gene expression was performed with SYBR Green master mix (Life Technologies, 4364344). Expression levels were normalized to *Gapdh*, used as a housekeeping standard. Fold changes were determined with the Livak method[63].

## Primary cultures of tanycytes

Primary cultures of tanycytes were generated as described[64]. P10 Wistar rats (local breeding) were decapitated, and their brains were extracted and immersed in ice-cold sterile Hank's balanced salt solution (HBSS; Thermo Fisher). The median eminence was dissected under a stereomicroscope (Leica, M205) and crushed on 80-μm nylon meshes.

Dissociated cells were cultured in DMEM/F12-phenol red free medium (Thermo Fisher) supplemented with 10% fetal calf serum (Invitrogen). Primary cultures of tanycytes were kept in 5% CO₂ atmosphere at 37 °C. Media were half-refreshed every three days. Two days before protein extraction, primary cultures of tanycytes were split in 6-well plates and cultured in DMEM/F12-phenol red free medium supplemented with 5 μg ml⁻¹ insulin from bovine pancreas (Sigma) and 100 μM putrescine dihydrochloride (Sigma).

## Protein extraction from cultured tanycytes

Primary cultures of tanycytes were washed with ice-cold HBSS (Thermo Fisher), harvested, and pelleted at 1,000 rpm for 60 s. The supernatant was discarded. Pellets were resuspended in 300 mM NaCl, 50 mM HEPES (pH 8.0), 1% IGEPAL CA-630, 0.1% sodium deoxycholate, 1 mM DTT, 1 mM protease inhibitors (EDTA-free, Roche) and incubated on ice for 10 min. Cell lysates were flash-frozen in liquid N₂ and stored at −80 °C.

## Mass spectrometry

Bands on SDS gels (*n* = 3 biological replicates) were cut into three pieces each and the corresponding proteins were extracted. The proteins of each band were collected as fractions (three for each sample) and subjected to tryptic digest and post-digest purification.

Approximately 1 μg of tryptic peptides (4.5 μl injection volume) from each fraction (in total three) were separated by an online reversed-phase (RP) HPLC (Dionex Ultimate 3000 RSLCnano LC system, Thermo Scientific) connected to a benchtop Quadrupole Orbitrap (Q-Exactive Plus) mass spectrometer (Thermo Fisher Scientific). Online separation was performed on analytical (nanoViper Acclaim PepMap RSLC C18, 2 μm, 100 Å, 75 μm internal diameter × 50 cm, Thermo Fisher Scientific) and trap (Acclaim PepMap100 C18, 3 μm, 100 Å, 75 μm internal diameter × 2 cm, Thermo Fisher Scientific) columns. The flow rate for the gradient was set to 300 μl min⁻¹, with an applied maximum pressure at 750 mbar. The liquid chromatography method was a 175-min run and the exponential gradient was set at 5–32% buffer B (v/v%; 80% acetonitrile, 0.1% formic acid, 19.9% ultra-high purity LC-MS water) over -118 min (7 curves). This was followed by a 30-min gradient of 50% buffer B (6 curves) and then increased to 90% of buffer B for another 5 min (5 curves). The liquid chromatography eluent was introduced into the mass spectrometer through an integrated electrospray metal emitter (Thermo Electron). The emitter was operated at 2.1 kV and coupled with a nano-ESI source. Mass spectra were measured in positive ion mode applying top ten data-dependent acquisition (DDA). A full mass spectrum was set to 70,000 resolution at *m/z* 200 (Automatic Gain Control (AGC) target at 3 × 10⁶, maximum injection time of 30 ms and a scan range of 350–1,800 (*m/z*)). The MS scan was followed by a MS/MS scan at 17,500 resolution at *m/z* 200 (AGC target at 1 × 10⁵, 1.8 *m/z* isolation window and maximum injection time of 70 ms). For MS/MS fragmentation, normalized collision energy for higher energy collisional dissociation was set to 30%. Dynamic exclusion was at 30 s. Unassigned and +1, +8 and > +8 charged precursors were excluded. The minimum AGC target was set to 1.00e³ with an intensity threshold of 1.4e⁴. Isotopes were excluded. Targets were accepted if more than two peptide fragments covered each and listed in Extended Data Table 1.

## CSF extraction and VEGF ELISA

Wistar rats of ~P60 of age (all male, *n* = 3 for 25 °C and *n* = 4 for 40 °C) were allowed to habituate to the experimental setting in an incubator (Tecniplast, Aria BIO-C36 EVO) at 25 °C with a reverse 12 h 12 h light:dark cycle for 3 days. Next, rats were acutely exposed to either 25 °C or 40 °C for 1 h, anaesthetized intramuscularly with a mixture of ketamine (50 mg kg⁻¹) and xylazine (4 mg kg⁻¹), and their heads were mounted in a stereotaxic frame (RWD). For CSF sampling, the fourth ventricle was approached. For this, the skin was incised, nuchal muscles were retracted to the sides, and partially removed. The dorsal wall of the ventricle formed by the *lamina epithelialis* was identified as a silvery

membrane caudal to the cerebellum between the rim of the *foramen magnum* and first cervical vertebra. The membrane was pierced with a 26G syringe and 15 µl CSF was removed from the fourth ventricle using a standard 20-µl laboratory pipette (Eppendorf). Samples were flash-frozen in liquid $N_2$ and stored at −80 °C. To test the VEGF content of the CSF, we used a rat VEGF ELISA Kit (Sigma Aldrich; RAB0511) as per the manufacturer's instructions. An ELISA plate reader set at 450 nm (Glomax Multi[+], Promega) was used to read out VEGF levels in 20-µl sample volumes. VEGF concentrations were expressed in pg ml$^{-1}$.

## Electrophysiology, Ca$^{2+}$ imaging, optogenetics and analysis

Acute coronal slices comprising, in the rostrocaudal axis, the medial-caudal portion of the third ventricle were obtained from P60-P90 male C57Bl6/J, *Rax$^{tm1.1(cre/ERT2)Sbls}$/J::B6;129S6-Polr2a$^{Tn(pb-CAG-GCaMP5g,-tdTomato)Turd}$/J* and *B6;129S-Slc17a6$^{tm1.1(flpo)Hze/J}$* mice. Mice were anaesthetized with isoflurane (5%, 1 l min$^{-1}$ flow rate) prior to decapitation, and their brains were rapidly dissected out. Two hundred fifty-µm-thick coronal slices were cut on a vibratome (VT1200S, Leica) in ice-cold cutting solution (pH 7.3) containing (in mM): 135 *N*-methyl-D-glucamine, 1 KCl, 1.2 KH$_2$PO$_4$, 10 glucose, 20 choline bicarbonate, 1.5 MgCl$_2$, and 0.5 CaCl$_2$ and continuously oxygenated with 95% O$_2$/5% CO$_2$. Acute slices of the caudal portion of the hypothalamic third ventricle/ARC were incubated at 32 °C for 1 h and allowed to cool to 25 °C in oxygenated ACSF (pH 7.3) containing (in mM): 124 NaCl, 3 KCl, 1.25 KH$_2$PO$_4$, 2 MgCl$_2$, 2 CaCl$_2$, 26 NaHCO$_3$, and 10 mM glucose. For recordings, brain slices were transferred to a recording chamber (Examiner.D1, Zeiss) and superfused with ACSF (25 °C) at a rate of 3 ml min$^{-1}$ with a peristaltic pump (PPS5, Multichannel Systems). Tanycytes and neurons were recorded through patch pipettes (3–5 MΩ) made from borosilicate glass capillaries pulled on a P100 glass puller (Sutter Instruments). Patch pipettes were filled with an intracellular solution containing (pH 7.3, 300 mOsm; in mM): 125 K-gluconate, 20 KCl, 0.1 EGTA, 2 MgCl$_2$, 10 HEPES, 2 Na-ATP, 0.4 Na-GTP, 10 phosphocreatine and 0.5% biocytin (Tocris, 3349).

**Electrophysiology.** To record glutamatergic inputs onto tanycytes, both sEPSCs and tonic currents were recorded at −70 mV using a Multiclamp 700B amplifier (Molecular Devices), sampled at 10 KHz, and filtered at 2 KHz. EPSCs were analysed using the Mini Analysis Program (Synaptosoft). Both the amplitude and frequency of sEPSCs were statistically tested in both α- and β-tanycytes. s-AMPA (100 µM; Tocris, 0254) was superfused to test for tonic currents. To define voltage responses to currents ramps, tanycytes were recorded in current-clamp mode with the holding current set at 0 pA. Current injections were applied for 1 s with consecutive steps of current of 5 pA for 20 sweeps. To determine the effect of the threshold for neuronal spiking on VEGFA release, acute slices were either superfused with ACSF (control) or with axitinib (40 µM; LC Laboratories A-1107), a selective inhibitor of VEGF receptors. To define their action potential thresholds, patch-clamped neurons were recorded in current-clamp mode with the holding current set at 0 pA. Patch-clamped neurons in the ARC and apposing α-tanycytes were recorded in repeated measures first at 25 °C and after increasing the temperature of the recording chamber to 38 °C by using a temperature controller (Warner Instruments, TC-324C). The voltage value corresponding to the exponential rise of the action potential was used for statistical analysis (Clampfit, Molecular Devices).

**Ca$^{2+}$ imaging.** We recorded neuronal input-dependent Ca$^{2+}$ transients in tanycytes from acute slices from *Rax$^{tm1.1(cre/ERT2)Sbls}$/J* crossed with *B6;129S6-Polr2a$^{Tn(pb-CAG-GCaMP5g,-tdTomato)Turd}$/J* mice (*n* = 8, males). We used an AxioExaminer.D1 microscope (Zeiss) and visualized Ca$^{2+}$ transients with a water-immersion W40×/1.0 DIC VIS-IR Plan-Apochromat objective (Zeiss) and a CoolSnap HQ$^2$ camera (Photometrics). We first proceeded to patch-clamp neurons proximal to the wall of the third ventricle. To induce action potentials in patch-clamped neurons, we injected steps of currents ranging between 10 pA and 30 pA for 500 ms. Simultaneously,

a VisiChrome monochromator (Visitron Systems) was used to visualize GCaMP5g in tanycytes. To demonstrate the AMPA receptor (AMPAR) dependence of Ca$^{2+}$ transients, tanycytes were imaged while ACSF was supplemented with 2,3-dioxo-6-nitro-7-sulfamoyl-benzo[f]quinoxaline (NBQX, 20 µM, Tocris, 1044). In recordings where neuronal activity was pharmacologically manipulated, acute slices from *Rax$^{tm1.1(cre/ERT2)Sbls}$/J* crossed with *B6;129S6-Polr2a$^{Tn(pb-CAG-GCaMP5g,-tdTomato)Turd}$/J* mice were placed on µ-Dish 35 mm high chamber for cell culture imaging (Ibidi) mounted on an inverted LSM880 confocal microscope (Zeiss), and visualized with a Plan-Apochromat 20×/0.8 M27 objective (Zeiss). ACSF, 100 µM picrotoxin (Tocris, 1128), 5 µM TTX (Tocris, 1069), 100 µM s-AMPA (Tocris, 0254), 20 µM NBQX (Tocris, 1044) and KCl 50 mM were superfused at a rate of 1.5 ml min$^{-1}$ with a peristaltic pump (PPS5, Multichannel Systems). Single-plane images of the GCaMP5g signal were captured upon excitation with a 488-nm laser at 5.5% of total efficient power output to avoid phototoxicity. A frame dimension of 512 × 512 pixels at 8 bit with a rate of 600 ms was used with the pinhole set at 447 µm. To analyse Ca$^{2+}$ transients, image series were loaded in Fiji and the intensity of GCaMP5g transients was calculated from manually drawn regions of interest over tanycyte somata and basal processes proximal to the third ventricle. The GCaMP5g signal was normalized to the difference between the signal intensity in tanycytes during their period of inactivity and background.

**ChR2-assisted circuit mapping.** Ex vivo coronal brain slices (300 µm) encompassing the medial-caudal portion of the third ventricle were cut from *B6;129S-Slc17a6$^{tm1.1(flpo)Hze/J}$* mice bilaterally injected with AAV1-CAG-FLEXFRT-ChR2(H134R)-mCherry in the PBN to test possible monosynaptic inputs onto tanycytes. Brain slices were superfused with oxygenated ACSF containing 1 µM TTX (Tocris, 1069) and 100 µM 4-aminopyridine (Sigma Aldrich, 275875) with a peristaltic pump (Multichannel systems, PPS2) at a flow rate of 3 ml min$^{-1}$ at 25 °C throughout. A BX51WI microscope (Olympus) equipped with a DIC prism (Olympus, WI-DICHTRA2), and LUMPlanFI/IR 60X/0.90W and Plan N4×/0.10 objectives (Olympus) was used. channelrhodopsin-2(ChR2)-mCherry$^+$ axons in close apposition to the third ventricle were excited with a CoolLED (pE-100) light source at 535 nm and imaged on an ORCA-Fusion digital camera (Hamamatsu, C14440). Tanycytes were clamped at a holding potential of −70 mV, and data acquired on an EPC10 USB Quadro patch-clamp amplifier (HEKA) were sampled at 20 KHz, and filtered at 2 KHz. ChR2−mCherry$^+$ terminals were excited with 50-ms light pulses at 470 nm (CoolLED, pE-100) synchronized to the recording of possible optically induced EPSCs in tanycytes. The time response (in ms) and amplitude (in pA) of EPSCs were analysed in PatchMaster Next (HEKA).

## Effects of TRPV2 inhibition and 38 °C on food intake

We injected tranilast (20 mg kg$^{-1}$, intraperitoneally T0318-10MG; Sigma Aldrich) in C57Bl6/N mice (*n* = 4) and compared its effect with naive controls (*n* = 4) and mice injected with DMSO used as a vehicle (D2650; Sigma Aldrich). Mice were injected with either tranilast or DMSO 10 min before being exposed to 25 °C and then to 38 °C for 1 h on consecutive days. The tranilast concentration was chosen based on dose conversion from human to mouse (considering the body surface area according to US Food and Drug Administration guidelines: http://www.fda.gov/downloads/Drugs/Guidances/UCM078932.pdf). An equivalent mg kg$^{-1}$ dose for tranilast in mice was calculated by multiplying its human dose (100 mg per 60 kg, equivalent to 1.6 mg kg$^{-1}$ for human) by the body surface area conversion factor in mice (12.3), resulting in a dose of 19.68 mg kg$^{-1}$ in mouse.

## Stereotaxic surgery for viral injections

All mice undergoing stereotaxic delivery of AAV viral particles were processed 21 days after virus delivery. Anaesthesia was induced with isoflurane (5%; 0.6 l min$^{-1}$ flow rate). The mice were then mounted in

a stereotaxic frame (RWD) with anaesthesia maintained with isoflurane (1.5%; 0.6 l min$^{-1}$ flow rate) through a snout mask. Viral particles were delivered with a micropipette (Drummond) mounted on either a Quintessential Stereotaxic Injector (Stoelting) or an R-480 nanolitre microinjection pump (RWD) at a speed of 100 nl min$^{-1}$. The pipette was slowly withdrawn 10 min after AAV delivery.

*B6.Cg-Gt(ROSA)26Sor*$^{tm14(CAG-tdTomato)Hze/J}$ and *B6;129P2-Mapt*$^{tm2Arbr/J}$ mice used for the transsynaptic mapping of neuronal afferents to tanycytes were unilaterally injected (lateral ventricle) with rAAV8-EF1a-mCherry-IRES-WGA-Cre particles (UNC Vector Core; 1.0 µl) at the following coordinates (all relative to bregma): anterior–posterior (AP): −0.1 mm, lateral (L): 0.9 mm, dorsoventral (DV): −2.3 mm.

To perform long-range axonal tracing to the third ventricle, C57Bl6/J mice were unilaterally injected (in the ARC) with AAVrg-CAG-GFP particles (70 nl, Addgene, 37825) as above at the following coordinates (all relative to bregma): AP: −1.94 mm; L: 0.25 mm; DV: −5.86 mm.

For ChR2-assisted-circuit mapping to assess monosynaptic inputs from the PBN to tanycytes, *B6;129S-Slc17a6*$^{tm1.1(flpo)Hze/J}$ mice were bilaterally injected with AAV1-CAG-FLEXFRT-ChR2(H134R)-mCherry particles (250 nl, Addgene, 75470-AAV1) as above at the following coordinates (all relative to bregma): AP: −5.2 mm; L: ±1.25 mm; DV: −2.8 mm.

To test tanycyte activation following chemogenetic manipulation of PBN projections and also in behavioural tests, *B6;129S-Slc17a6*$^{tm1.1(flpo)Hze/J}$ and *Rax*$^{tm1.1(cre/ERT2)Sbls/J}$ mice were crossed to obtain *Slc17a6*-FlpO::*Rax*-CreER$^{T2}$ mice that were bilaterally injected with either AAV2/1-Syn-FRT-hM3D(G$_q$)-mCherry particles (Viral Vector Core Facility, Canadian Neurophotonics Platform; RRID:SCR_016477) or AAV-EF1a-FRT-hM3D(G$_q$)-mCherry particles (Molecular Biology Services, Institute of Science and Technology Austria) at volumes of 250 nl each at the following coordinates (all relative to bregma): AP: −5.2 mm; L: ±1.25 mm; DV: −2.8 mm.

To block tanycyte-dependent VAMP2-mediated exocytosis in behavioural experiments, *Rax*-CreER$^{T2}$ mice or *Slc17a6*-FlpO::*Rax*-CreER$^{T2}$ mice were medially injected in the third ventricle with AAV-TeLC-FLEX-GFP[56] or AAV2-FLEX-GFP (control) viruses (1.0 µl; coordinates relative to bregma: AP: −1.70 mm; L: ±0.0 mm; DV: −5.85 mm).

To block VAMP2-mediated exocytosis in PBN neurons, *Flp*-dependent AAV2-FlpON-TeLC-GFP or AAV2-FlpON-GFP (control) viruses were injected in the PBN of *Slc17a6*-IRES2-FlpO-D-mice (250 nl) at the coordinates: AP: −5.2 mm; L: ±1.25 mm; DV: −2.8 mm (all relative to bregma).

## Tamoxifen injection

*Rax*$^{tm1.1(cre/ERT2)Sbls}$ mice used for histochemical analysis, as well as *Rax*$^{tm1.1(cre/ERT2)Sbls}$/J mice crossed with *B6;129S6-Polr2a*$^{Tn(pb-CAG-GCaMP5g,-tdTomato)Turd}$/J mice for Ca$^{2+}$ imaging were injected intraperitoneally for 3 consecutive days with 150 mg kg$^{-1}$ tamoxifen (Sigma, T5648), and processed 3 days following the last injection. For behavioural tests, *Rax*$^{tm1.1(cre/ERT2)Sbls}$ mice crossed with *B6;129S-Slc17a6*$^{tm1.1(flpo)Hze/J}$ mice were injected intraperitoneally for 3 consecutive days with 50 mg kg$^{-1}$ 4-hydroxytamoxifen (Sigma, H6278) to ensure maximal recombination of the AAV-TeLC-FLEX-GFP construct in tanycytes.

## Behavioural tests and controls

To test the effect of acute heat exposure on food intake, P60 C57Bl6/J mice were habituated to the experimental room set to 25 °C for 24 h. Next, mice were transferred to thermo-controlled cabinets (Sanyo Incubator, MIR-254) preset to either 25 °C (control) or 40 °C for 1 h. Following heat exposure, mice were single housed in PhenoTypers (Noldus) placed into incubators (Memmert, MEMM-OT3007S and Tecniplast, Aria BIO-C36 EVO) set to 25 °C with a reversed 12 h:12 h light:dark cycle for another 24 h. Food and fluid intake, as well as mobility were monitored over 24 h after acute thermal manipulation by weighing the food pellet, measuring the volume of water consumed, or scoring the frequency of eating bouts and general

mobility (both in EthoVision XT15; Noldus). Behavioural tests were designed such that each mouse served as its own control (baseline versus post-heat exposure data), allowing statistical analysis through repeated-measures analysis of variance (ANOVA).

To test if neuronal activity-induced VEGFA release from tanycytes affected food intake, P60–P70 male C57Bl6/J mice were intracerebroventricularly infused with 1 nmol/1.5 µl of either Accell mouse *Vegfa* siRNA (*Vegfa*-RNAi; Dharmacon, E-040812-00-0020) or Accell non-targeting siRNA (control; Dharmacon, D-001950-01-20) in the third ventricle (AP: −1.70 mm; L: ±0.0 mm; DV: −5.85 mm relative to bregma). First, we tested the knockdown efficiency of *Vegfa*-RNAi by infusing P60–P70 male C57Bl6/J mice ($n = 4$ per group) with either scrambled RNAi or *Vegfa*-RNAi (stereotaxic surgery was identical as described above). Eight days after RNAi infusion, mice were perfused with ice-cold 4% PFA and the brains processed for FISH. Next, to test the impact of reduced VEGFA release on food intake upon heat exposure, P60–P70 male C57Bl6/J mice were intracerebroventricularly infused with either scrambled RNAi (control) or *Vegfa*-RNAi ($n = 8$ per group). Mice were single housed in PhenoTypers (Noldus) placed into incubators (Tecniplast, Aria BIO-C36 EVO) at 25 °C with a reversed 12 h:12 h light:dark cycle, and allowed to recover for 8 days. From day 3 to 8 post-surgery, we monitored both food intake and body mass by weighting the food pellets and mice, respectively. On days 9 and 10, mice were subjected to thermal challenge (40 °C, 1 h) in incubators (Sanyo, MIR-254). This was followed by measuring foor intake and body mass for 24 h as above (PhenoTypers, Noldus).

To test the effect of the chemogenetic activation of PBN projections onto tanycytes, male *Rax*$^{tm1.1(cre/ERT2)Sbls}$::*B6;129S-Slc17a6*$^{tm1.1(flpo)Hze}$/J mice were stereotaxically injected with AAV-TeLC-FLEX-GFP (third ventricle) and AAV-FRT-hM3D(G$_q$)-mCherry (PBN, bilaterally) to simultaneously manipulate tanycytes and glutamatergic output form the PBN. All tests were performed in a self-controlled design to use the same mice before and after blocking VAMP2-mediated exocytosis from tanycytes, by the temporally controlled recombination of the AAV-TeLC-FLeX-GFP construct that encodes TeLC (Fig. 5g). Twenty-one days after virus delivery, mice were placed individually in PhenoTypers (Noldus) mounted in ventilated and temperature-controlled (29 °C) cabinets (Memmert, MEMM-OT3007S) with a reversed 12 h:12 h light:dark cycle. Food intake, locomotion, and drinking were monitored with EthoVision XT15 (Noldus). Mice were allowed to habituate for 2 days to the experimental setup (days 21,22). Next, baseline activity was recorded for 24 h (day 23). On day 24, mice were treated with 3 mg kg$^{-1}$ CNO (Tocris, 6329) by both intraperitoneal delivery and in the drinking water, together with 5 mM saccharine (Sigma), to test the effect of chemogenetically activating PBN projections on feeding, drinking, and locomotor activity, whilst leaving VAMP2-mediated exocytosis from tanycytes unaffected. Thereafter, mice were placed individually in home cages for Cre-dependent recombination of the TeLC construct to take place into *Rax*-expressing tanycytes by injecting 50 mg kg$^{-1}$ 4-hydroxytamoxifen (Sigma) for 3 days (days 25–27). Mice were then allowed to recover for another 3 days (days 28–30). On day 31, we returned the mice to the PhenoTypers and allowed them to habituate for another 48 h (days 31 and 32). Thereafter, we recorded (for 24 h, day 33) their baseline activity following the TeLC-dependent block of VAMP2 in tanycytes. The next day (day 34), we triggered neuronal activity in the PBN by injecting CNO (3 mg kg$^{-1}$) and using it as an additive to the drinking water together with saccharine (5 mM), and tested feeding, drinking, and locomotor activity again. On the last day (day 35), mice were transcardially perfused with ice-cold 4% PFA. Their brains were routinely processed to verify the accuracy of virus delivery. No mouse was excluded from the analysis.

To test if blocking VAMP2-mediated exocytosis in PBN neurons projecting to tanycytes affected food intake following acute heat exposure, *Slc17a6*$^{tm1.1(flpo)Hze}$/J mice were bilaterally injected in the PBN with either AAV2-FlpON-GFP (control) or AAV2-FlpON-TeLC-GFP. Twenty-one days after virus delivery, mice were sequentially exposed to either

25 °C (control) or 40 °C for 1 h (on consecutive days). Food intake was determined by measuring the weight of food pellets. To test if blocking VAMP2-mediated exocytosis in tanycytes could modify food intake following acute heat exposure, $Rax^{tm1.1(cre/ERT2)Sbls}/J$ mice were medially injected in the third ventricle with either AAV2-FLeX-GFP (control) or AAV2-FLeX-TeLC-GFP. To induce Cre-dependent recombination, mice were injected with tamoxifen (150 mg kg$^{-1}$) for 3 consecutive days, starting 2 days after surgery. Twenty-one days after virus delivery, mice were sequentially exposed to either 25 °C (control) or 40 °C for 1 h (on consecutive days). In both experiments, food intake was determined by measuring the weight of food pellets.

## Statistics and reproducibility

Data were analysed using GraphPad Prism 8.0.2 (GraphPad). Two sets of independent samples were compared using two-tailed Student's $t$-test. Repeated measures of pair-wise comparisons were analysed by paired two-tailed Student's $t$-test. Multiple sets of measurements involving one independent variable were analysed by one-way ANOVA and further justified by Bonferroni's post hoc comparison. Repeated-measures two-way ANOVA and three-way ANOVA were used to evaluate between and within factors, with Bonferroni's post hoc test applied throughout. The Kolmogorov–Smirnov test was used to analyse cumulative distribution. Data were expressed as means ± s.e.m. throughout, except in box-and-whisker plots that show median ± interquartile ranges, and minimum and maximum values. Statistical significance was indicated as $*P < 0.05$, $**P < 0.01$ or $***P < 0.001$. For neuroanatomy, a minimal desired cohort size of $n = 3$ mice was chosen, with higher mouse numbers specified in the relevant figure legends.

## Statistical output for main figures

Figure 1b: two-way repeated-measures ANOVA: interaction (sex versus temperature): $F = 0.005$, $P = 0.942$; sex: $F = 7.969$, $P = 0.013$; temperature: $F = 32.240$, $P < 0.001$. Bonferroni's multiple comparison: $t = 4.067$, $**P = 0.002$ (males at 25 °C versus 40 °C); $t = 3.963$, $**P = 0.003$ (females at 25 °C versus 40 °C).

Figure 1c: repeated-measures ANOVA: $F = 18.030$, $p < 0.001$.

Figure 1f: two-way ANOVA: interaction (sex versus temperature): $F = 1.497$, $P = 0.249$; sex: $F = 3.589$, $P = 0.087$; temperature: $F = 81.700$, $P < 0.0001$. Bonferroni's multiple comparison: $t = 6.788$, $***P < 0.001$ (males 25 °C versus 40 °C); $t = 5.969$; $***P < 0.001$ (females 25 °C versus 40 °C).

Figure 3b, middle: frequency: Student's $t$-test (two-sided), $t = 0.476$, $P = 0.639$; α- versus β-tanycytes.

Figure 3b, right: amplitude: Student's $t$-test (two-sided), $t = 3.006$, $**P = 0.007$; α- versus β-tanycytes.

Figure 4b: Student's $t$-test (two-sided), $t = 7.120$, $***P < 0.001$.

Figure 4e: repeated-measures ANOVA: interaction: $F = 3.974$, $P = 0.066$; treatment (ACSF versus axitinib): $F = 1.947$, $P = 0.185$; temperature: $F = 23.880$, $P < 0.001$; subject, $F = 6.723$; $P < 0.001$. Bonferroni's multiple comparison: ACSF (25 °C versus 38 °C), $t = 4.865$; $***P < 0.001$; axitinib (25 °C versus 38 °C) $t = 2.046$; $P = 0.1201$.

Figure 4g: Student's $t$-test (two-sided), $t = 3.143$, $*P = 0.020$.

Figure 4h: repeated-measures ANOVA: interaction (treatment versus temperature), $F = 1.081$, $P = 0.316$; temperature: $F = 17.310$, $P = 0.001$; treatment: $F = 3.089$, $P = 0.094$. Bonferroni's multiple comparison: temperature: control, $t = 3.677$, $**p = 0.005$; $Vegfa$-RNAi, $t = 2.207$, $P = 0.089$ (not significant).

Figure 4k: Student's $t$-test (two-sided), $**P < 0.01$.

Figure 5b, right: Student's $t$-test (two-sided), $**P < 0.01$; $n = 3$ mice per group.

Figure 5c: two-way ANOVA: interaction (TeLC versus temperature): $F = 8.682$, $P = 0.042$; GFP versus TeLC: $F = 0.683$, $P = 0.455$; temperature: $F = 16.34$, $P = 0.0156$. Bonferroni's multiple comparison: $P = 0.016$ (GFP; 25 °C versus 40 °C); $P = 0.964$ (TeLC; 25 °C versus 40 °C).

Figure 5i: two-way repeated-measures ANOVA: time: $F = 6.202$, $P = 0.026$; treatment: $F = 6.839$, $P = 0.048$; interaction (time versus treatment): $F = 1.944$, $P = 0.208$.

## Reporting summary

Further information on research design is available in the Nature Portfolio Reporting Summary linked to this article.

## Data availability

Source data are provided with this paper.

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

**Acknowledgements** The authors thank A. Reinthaler, P. Rebernik, E. Borok, D. Miloradovic, M. Shanabrough and A. Kristant for their expert laboratory assistance; S. Siegert and A. Venturino for providing B6;129S6-$Polr2a^{Tn(pb-CAG-GCaMP5g,-tdTomato)Tvrd}$/J mice and assistance with super-resolution microscopy; and X. Tan, S. Franzmeyer and S. Sideromenos for help with morphometry. This work was supported by the Austrian Science Fund (I 4854 and P 34281; D.D.P.), the National Brain Research Program of Hungary (2017-1.2.1-NKP-2017-00002 and NAP2022-I-1/2022; A.A.), the Excellence Program for Higher Education of Hungary (TKP-EGA-25; A.A.), the Klarman Family Foundation (T.L.H.), NIH (DK115933, DK126447, DK120891, AG067329 and AG051459; T.L.H.), the Swedish Research Council (2023-03058; T.H.); Novo Nordisk Foundation (NNF23OC0084476; T.H.); Hjärnfonden (FO2022-300; T.H.), European Research Council (ERC-2015-AdG-695136 and ERC-2020-AdG-101021016; T.H.) and intramural funds of the Medical University of Vienna (T.H.).

**Author contributions** T.H. and M.B. conceived the project. T.H. and M.B. designed experiments. A.A., D.D.P., T.L.H. and T.H. procured funding. M.B., S.R., L.A., J.H., Z.H., A.G. and A.A. performed experiments and analysed data. K.B., D.D.P., G.L., P.W., V.P. and T.L.H. provided unique reagents, infrastructure and mouse models. M.B. and T.H. wrote the manuscript with input from all co-authors.

**Funding** Open access funding provided by Karolinska Institute.

**Competing interests** The authors declare no competing interests.

**Additional information**
**Correspondence and requests for materials** should be addressed to Tibor Harkany.

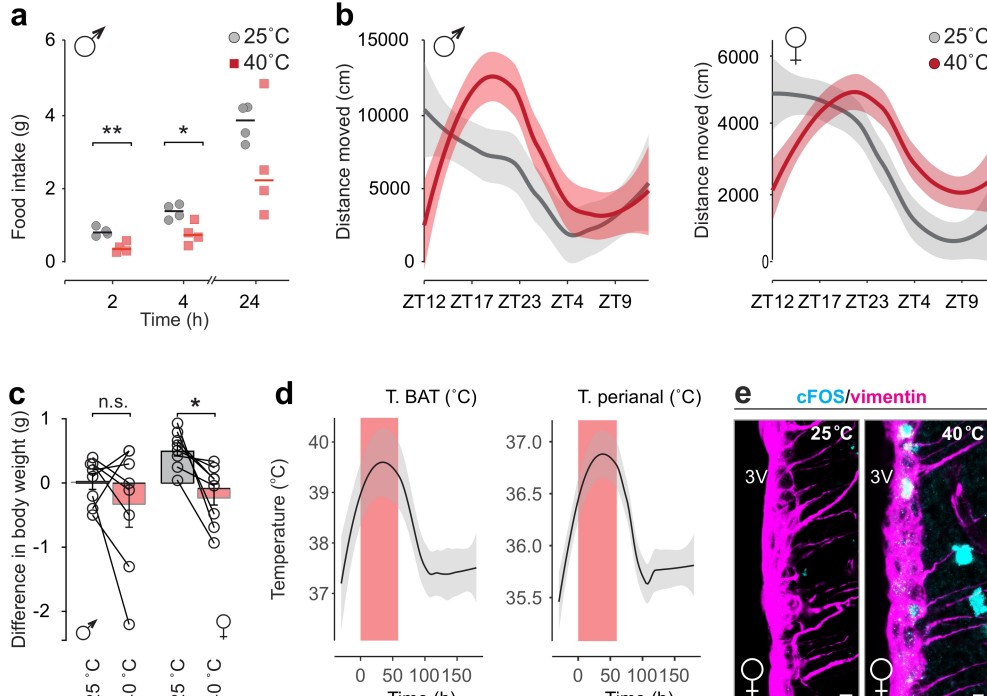

**Extended Data Fig. 1 | Acute heat restricts food intake and activates α tanycytes. a**. Time-resolved measurement of food intake after exposure to 40 °C for 1h in male mice. Control measurements (25 °C, 1h) were performed in the same mice 24h earlier. Data are from $n = 4$ mice, **$p < 0.01$, *$p < 0.05$, Student's $t$-test. **b**. Diurnal activity of male ($\male$, *left*) and female ($\female$, right) mice, expressed as the distance moved (cm) in a period of 20h following exposure to 25 °C (grey line and shading) and then to 40 °C (red line and shading) for 1h. Time intervals were expressed as *zeitgeber* time points (ZT), wherein ZT12 and ZT0 coincided with the onset of the dark and the light phases, respectively. Data were presented as facet-wraps with measurements at 1h intervals for 20 h; $n = 8$ mice/sex, repeated-measures ANOVA: <u>males</u> ($\male$), interaction (time *vs*. temperature): $F = 2.223$; $p = 0.002$; time: $F = 5.047$; $p < 0.001$; temperature: $F = 0.017$; $p = 0.898$ and <u>females</u> ($\female$): interaction (time *vs*. temperature): $F = 2.700$; $p < 0.0001$; time: $F = 11.190$; $p < 0.0001$; temperature: $F = 6.536$; $p < 0.001$. **c**. Changes in body weight (g) for each mouse 24h after exposure to 25 °C (grey) and then to 40 °C (pink bars) for 1h. Data from both males ($\male$) and females ($\female$)

are shown. Data in bar graphs were expressed as means ± s.e.m., while individual changes are shown by interconnected circles; $n = 8$/sex; repeated-measures ANOVA: interaction (sex *vs*. temperature): $F = 0.796$; $p = 0.3871$; sex: $F = 2.266$; $p = 0.1545$; temperature: $F = 8.713$; $p = 0.0105$. Bonferroni's multiple comparisons for temperature in <u>males</u>: $t = 1.456$; $p = 0.334$; <u>females</u>: $t = 2.718$; $p = 0.033$. **d**. Temperature changes in the intrascapular region (BAT, *left*) and the perianal region (*right*), expressed in Celsius (°C). Data represents facet-wraps with measurements at 15-min intervals before, during, and after exposing mice to acute heat for 1h (40 °C, pink shading); $n = 8$ mice/group, repeated measures ANOVA: BAT: $F = 11.260$; $p < 0.001$; perianal: $F = 36.050$; $p < 0.001$. **e**. Confocal micrographs showing cFOS immunoreactivity (in cyan) in vimentin⁺ α-tanycytes (in magenta; at −2.30 mm relative to bregma) at the temperatures indicated in female mice ($\female$; $n = 4$/group). Data in facet-wrap plots were expressed as means ± 95% confidence interval of the s.d. throughout. *Abbreviation*: 3V, 3rd ventricle. *Scale bar* = 5 μm.

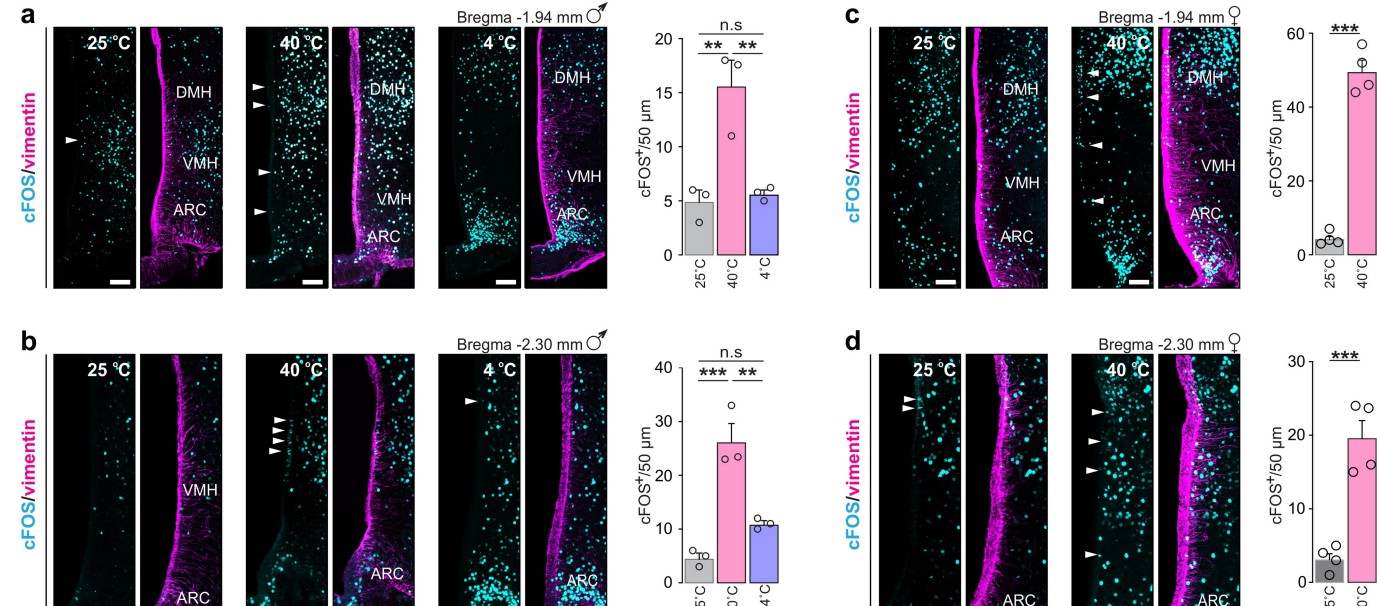

**Extended Data Fig. 2 | Acute heat exposure induces cFOS expression in tanycytes. a**. *Left*: Confocal micrographs showing cFOS immunoreactivity (in cyan) at −1.94 mm (relative to bregma) after 1h exposure to either 40 °C or 4 °C *vs.* 25 °C, used as control, in <u>male</u> (♂) mice. The dorsomedial hypothalamic nucleus (DMH), the ventromedial hypothalamic nucleus (VMH) and the arcuate nucleus (ARC) were indicated. Arrowheads point to cFOS⁺/vimentin⁺ (the latter marker in magenta) tanycytes. *Scale bar* = 70 μm. *Right*: Quantitative analysis of cFOS⁺ α-tanycytes. Data in bar graphs represent means ± s.e.m. with individual data points shown as circles; $n = 3$/condition; one-way ANOVA: $F = 16.34$; $p = 0.004$. Bonferroni's multiple comparison: 25 °C *vs.* 40 °C: $t = 5.102$; $p = 0.007$; 25 °C *vs.* 4 °C: $t = 0.3189$; $p \sim 1.0$ (n.s., non-significant); 40 °C *vs.* 4 °C: $t = 4.783$; $p = 0.009$. **b**. *Left:* Confocal micrographs showing cFOS immunoreactivity (in cyan) at −2.30 mm (relative to bregma) after 1h exposure to 40 °C or 4 °C *vs.* 25 °C, used as control, in <u>male</u> (♂) mice. Arrowheads point to cFOS⁺/vimentin⁺ (the latter marker in magenta) α-tanycytes. *Scale bar* = 70 μm. *Right*: Quantitative analysis of cFOS⁺ α-tanycytes. Data in bar graphs show means ± s.e.m. with individual

data points depicted as circles; $n = 3$/condition; one-way ANOVA: $F = 30.460$; $p < 0.001$. Bonferroni's multiple comparison: 25 °C *vs.* 40 °C: $t = 7.590$; $p < 0.001$; 25 °C *vs.* 4 °C: $t = 2.219$; $p = 0.205$; 40 °C *vs.* 4 °C: $t = 5.372$; $p = 0.005$. **c**. *Right*: Confocal micrographs of cFOS⁺ α-tanycytes (in cyan) at −1.94 mm (relative to bregma) after 1h exposure to 40 °C *vs.* 25 °C in <u>female</u> (♀) mice. *Arrowheads* denote cFOS⁺/vimentin⁺ tanycytes (in magenta). *Scale bar* = 100 μm. *Right:* cFOS⁺ α-tanycytes at −1.94 mm (relative to bregma) at the temperatures indicated. Data in bar graphs represent means ± s.e.m. with individual data points shown as circles; $n = 4$/condition; $t = 14.700$; $p < 0.0001$; Student's *t*-test. **d**. *Left:* Confocal micrographs with cFOS immunoreactivity (in cyan) at −2.30 mm (relative to bregma) following the temperature manipulations as indicated in <u>females</u> (♀). Arrowheads point to cFOS⁺/vimentin⁺ α-tanycytes (in magenta). *Scale bar* = 50 μm. *Right:* cFOS⁺ α-tanycytes at −2.30 mm. Data in bar graphs are means ± s.e.m. with individual data points shown as circles; $n = 4$/condition; $t = 6.331$; $p < 0.001$; Student's *t*-test.

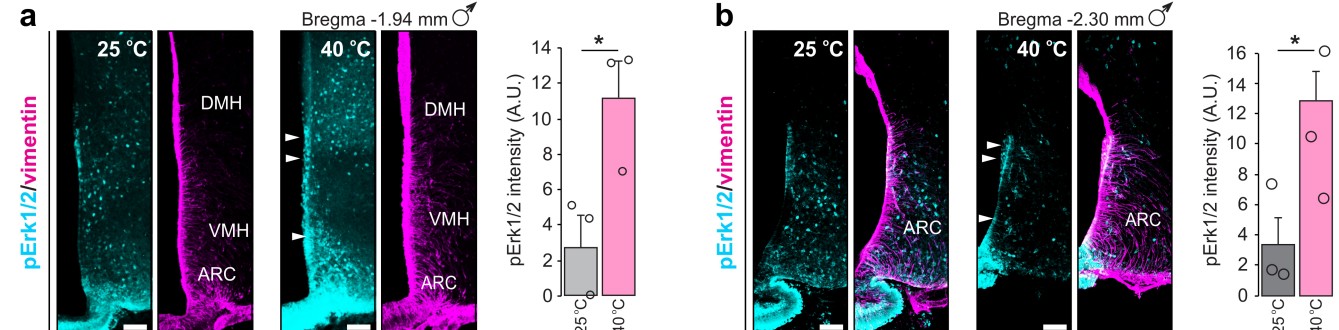

**Extended Data Fig. 3 | Acute heat induces pErk1/2 in tanycytes. a**. *Left*: Photomicrographs showing pErk1/2 immunoreactivity (in cyan) in tanycytes co-labeled with vimentin (in magenta) at −1.94 mm (relative to bregma) after 1h exposure to 40 °C *vs*. mice kept at 25 °C. The dorsomedial hypothalamic nucleus (DMH), the ventromedial hypothalamic nucleus (VMH) and the arcuate nucleus (ARC) were indicated. *Scale bar* = 70 μm. *Right*: pERK1/2 immunoreactivity in α-tanycytes in control (25 °C, in grey) and after exposure to 40 °C for 1 h (in pink). Data in bar graphs represent means ± s.e.m. with individual data points shown

as circles; *n* = 3/condition; *t* = 3.002; *p* = 0.039; Student's *t*-test. **b**. *Left*: Photomicrographs showing pErk1/2 immunoreactivity (in cyan) in vimentin⁺ α-tanycytes (in magenta) residing at −2.30 mm (relative to bregma) after 1 h thermal manipulation, relative to mice kept at 25 °C. *Scale bar* = 70 μm. *Right*: pErk1/2 immunoreactivity in α-tanycytes at −2.30 mm (relative to bregma) as indicated. Data in bar graphs are means ± s.e.m. Individual data points appear as circles; *n* = 3/condition; *t* = 3.565; *p* = 0.023; Student's *t*-test.

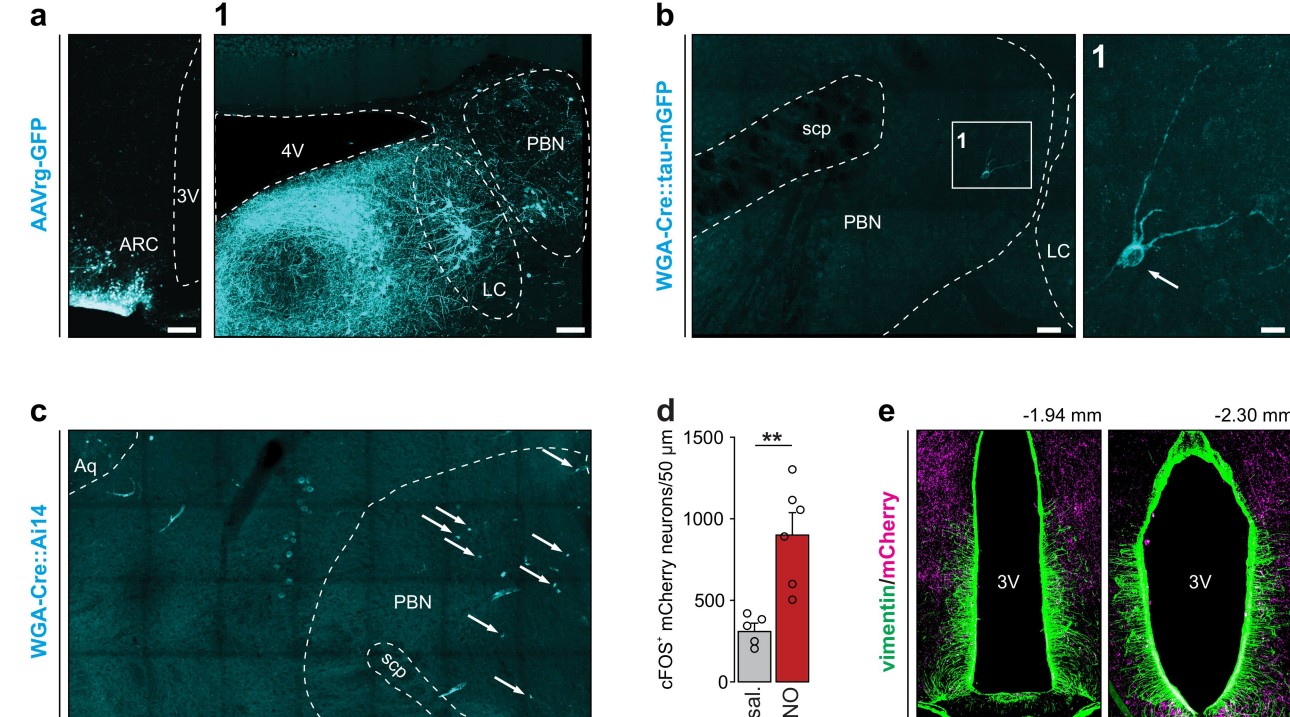

**Extended Data Fig. 4 | Parabrachial neurons innervate tanycytes. a.** Confocal image of GFP⁺ neurons within the arcuate nucleus (ARC; in cyan) transduced with an AAVrg-CAG-GFP vector and positioned ventrolateral to the 3$^{rd}$ ventricle (3V). *Scale bar* = 100 µm. **'1'.** Panoramic image from Fig. 2e, showing GFP⁺ neurons (in cyan) in the locus coeruleus (LC) and parabrachial nucleus (PBN) retrogradely labelled with an AAVrg-CAG-GFP construct. *Scale bar* = 100 µm. **b.** Panoramic confocal image of a GFP⁺ neuron (in cyan). Open rectangle denotes the position of '1' in the PBN trans-synaptically labelled by rAAV8-EF1a-mCherry-IRES-WGA-Cre in *Tau$^{mGFP}$-loxP* mice. *Scale bar* = 50 µm (b), 8 µm (1). **c.** Tiled image survey of Fig. 2f showing GFP⁺ neurons (in cyan, *arrows*) in the PBN

trans-synaptically labelled by a rAAV8-EF1a-mCherry-IRES-WGA-Cre virus in Ai14 mice. *Scale bar* = 50 µm. *Abbreviations*: 4V, 4$^{th}$ ventricle; Aq, aqueduct; scp, superior cerebellar peduncle. **d.** Number of cFOS⁺ mCherry-transduced neurons in the PBN. Data in bar graphs represent means ± s.e.m. for saline (grey, *n* = 5) and CNO (5 mg/kg in red, *n* = 6) groups. Circles correspond to individual data points; *t* = 4.094; *p* = 0.003; Student's *t*-test. **e.** Confocal images of the 3$^{rd}$ ventricle (3V) showing periventricular PBN efferents (mCherry, in magenta) with vimentin⁺ tanycytes in green. *Left* and *right* images are from coordinates −1.94 mm and −2.30 mm (relative to bregma), respectively. *Scale bars* = 70 µm.

**a**

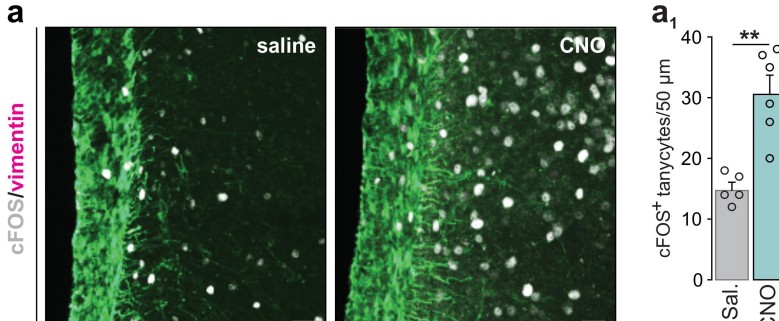

**Extended Data Fig. 5 | Chemogenetic induction of PBN neurons activates tanycytes. a**. Confocal images along the wall of the 3rd ventricle (−1.94 mm, relative to bregma) showed cFOS activation (in grey) both in vimentin+ α-tanycytes (in green) and neurons following an *i.p.* injection of either saline or CNO (5 mg/kg; 1.5 h) in *Slc17a6*-IRES2-FlpO-D mice in which AAV2-Syn1-FRT-hM3D(Gq)-mCherry particles were stereotaxically delivered to the PBN 21 days earlier. *Scale bar* = 15 μm. **a₁**. The number of cFOS+ α-tanycytes (−2.30 mm, relative to bregma) was expressed along 50 μm ventricular wall segments. Data in bar graphs are means ± s.e.m.; for saline (grey, $n$ = 5) and CNO-injected mice (5 mg/kg in red, $n$ = 6). Open circles correspond to individual data points; $t$ = 4.732; $p$ = 0.001; Student's *t*-test.

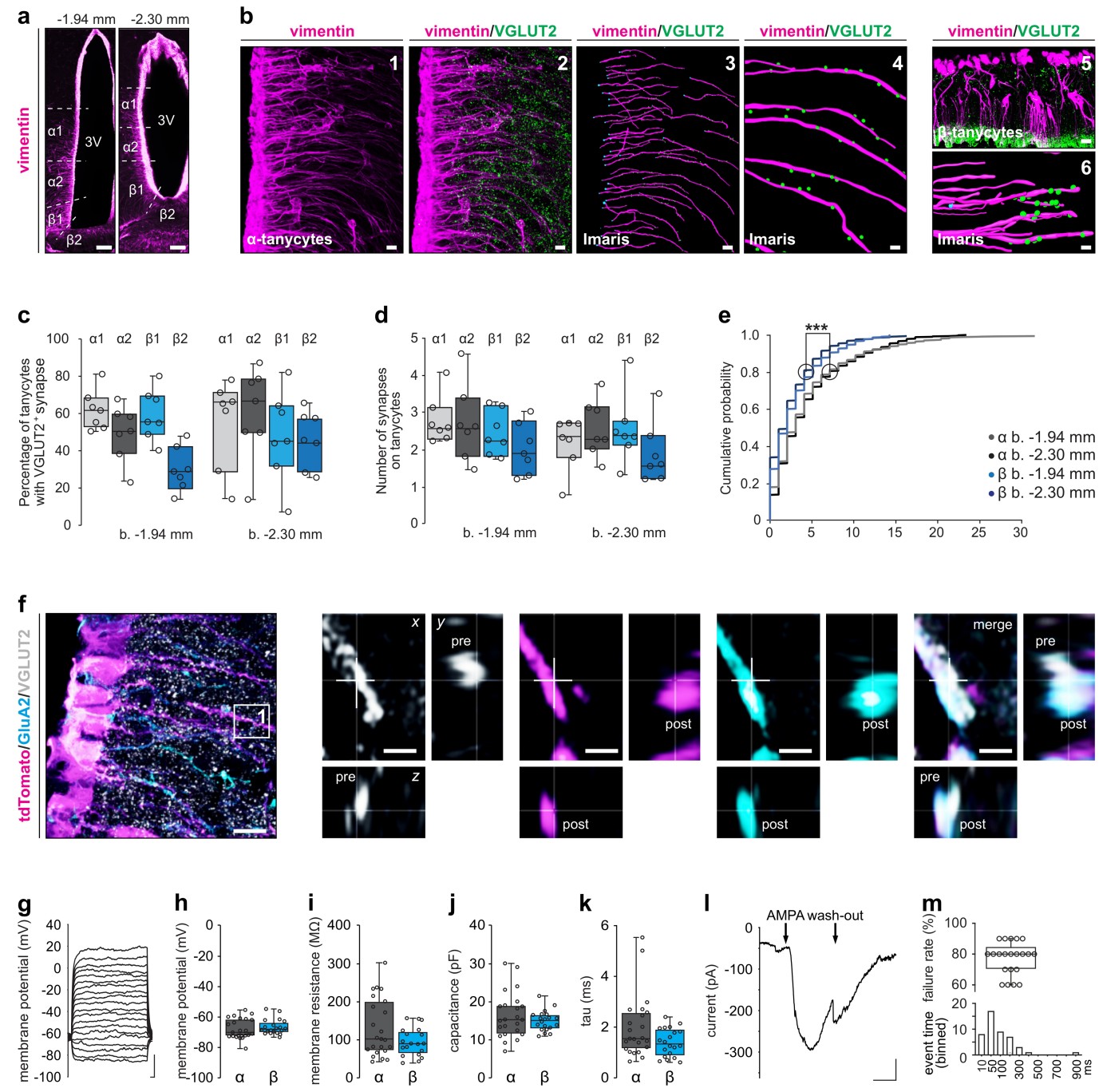

**Extended Data Fig. 6** | See next page for caption.

**Extended Data Fig. 6 | Synaptic activation of tanycytes. a**. Image survey of the third ventricle (3V) used to determine the density and distribution of synaptic inputs to vimentin⁺ tanycytes (in magenta). Dorsoventral subsetting of the ventricle wall (dashed lines) helped to identify the positions of α1, α2, β1, and β2 tanycytes. *Scale bars* = 100 μm. **b**. Image series to illustrate the image analysis pipeline used to quantify synaptic input onto tanycytes. **1**,**2**: Confocal images captured at 63x primary magnification show vimentin⁺ α-tanycytes (in magenta) and VGLUT2⁺ putative presynapses (in green). *Scale bar* = 10 μm. **3**: *'Filament'* and *'spots'* rendering algorithms were superimposed on vimentin (in magenta) and VGLUT2 (in green), respectively. *Scale bar* = 10 μm. The image shows how VGLUT2⁺ terminals were reconstructed as *'spots'* and accepted as relevant only within a distance of < 0.5 μm from tanycyte filaments by using the built-in MatLab *'find-spots-close-to-filaments'* function in Imaris. **4**: Magnified view of a subset of VGLUT2⁺ spheres (in green) that were < 0.5 μm to the vimentin signal (in magenta). *Scale bar* = 2 μm. **5**: VGLUT2⁺ inputs contacting β2-tanycytes. *Scale bar* = 20 μm. **6**: Reconstruction of VGLUT2⁺ inputs contacting β2-tanycytes. *Scale bar* = 5 μm. **c**. Boxplots showing the probability of α1, α2, β1, and β2 tanycytes receiving VGLUT2⁺ inputs at -1.94 mm (relative to bregma, *left*) and −2.30 mm (relative to bregma, *right*). None of the tanycyte subtypes was significantly different; $n$ = 7 mice, $n$ = 50 processes reconstructed/subtype/animal, $n$ = 200 processes/animal in total. **d**. Boxplots of the number of VGLUT2⁺ terminals on tanycytes at the rostro-caudal locations as above. None of the tanycyte subtypes was significantly different in $n$ = 7 mice, $n$ = 50 processes reconstructed/subtype/animal, $n$ = 200 processes/animal in total. **e**. Quantitative analysis of VGLUT2⁺ punctae in a distance of < 0.5 μm to reconstructed filaments from both rostral and caudal regions.

Data were shown as cumulative plots of VGLUT2⁺ terminals juxtaposing α- and β-tanycytes (pooled) at both rostral and caudal positions relative to bregma (b.) as above; $n$ = 500 filaments were reconstructed/tanycyte subtype from $n$ = 7 male mice (P60-90). Kruskal-Wallis test: KW = 96.040; $p < 0.001$ for α (caudal) *vs*. β (caudal); for α (rostral) *vs*. α (caudal); for β (caudal) *vs*. β (rostral). **f**. *Left*: Topographic view of tanycytes expressing GluA2 in apposition to VGLUT2⁺ terminals in *Rax*-CreER^T2::Ai14 mice. Open rectangle ('1') shows the location of the image series to the right. *Scale bar* = 5 μm. *Right*: Image series in left-to-tight order: high-resolution images of VGLUT2⁺ presynapses ('pre'; in grey), tdTomato⁺ tanycytes (in magenta), GluA2 subunits ('post', in cyan), and their overlay. *Scale bar* = 500 nm. **g**. Current-clamp trace showing the membrane potential (mV) in tanycytes upon current steps. Vertical and horizontal scales are 20 mV and 50 ms, respectively. **h**. Boxplots define the resting membrane potential (mV) in α- ($n$ = 23) and β-tanycytes ($n$ = 20); $t$ = 0.624; $p$ = 0.536. **i**. Boxplots showing the membrane resistance (MΩ) in α ($n$ = 23) and β tanycytes ($n$ = 20); $t$ = 1.980; $p$ = 0.054. **j**. Boxplots depict the membrane capacitance (pF) in α- ($n$ = 23) and β-tanycytes ($n$ = 20); $t$ = 0.960; $p$ = 0.343. **k**. Boxplots for the time constant (τ) to dissipate current (ms) in α- ($n$ = 23) and β-tanycytes ($n$ = 20); $t$ = 1.974; $p$ = 0.0552. Open circles in **h-k** are individual data points; data were analyzed by Student's *t*-test throughout. **l**. Representative trace showing the inward deflection of a tonic current in response to s-AMPA (100 μM) and its reversal upon wash-out. Vertical and horizontal scales are 50 pA and 100 s, respectively. **m**. *top:* Failure rate of optogenetically-induced EPSCs in tanycytes. Boxplot depicts means ± s.e.m., with individual data points shown as open circles. *Bottom*: Distribution of the lag-time of EPSCs (ms) relative to the onset of optical stimuli. Data were binned as indicated.

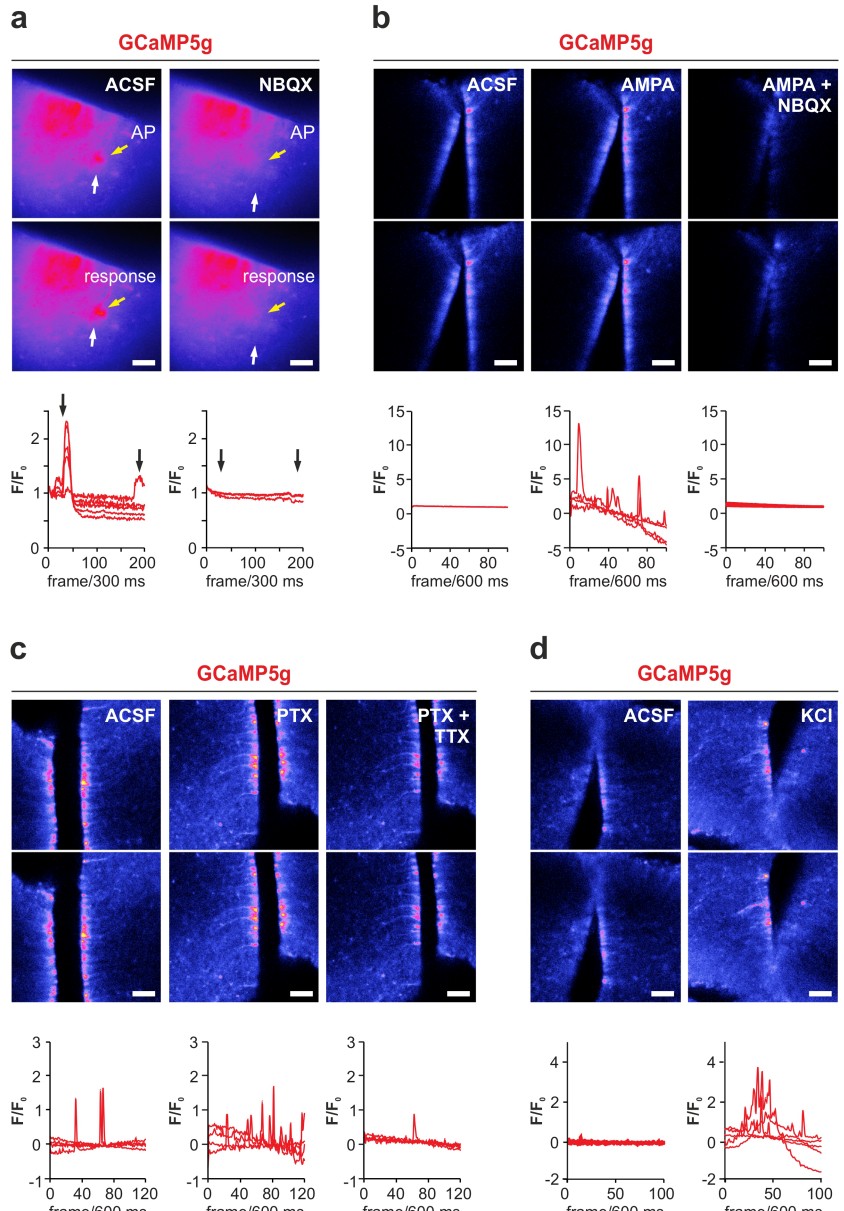

**Extended Data Fig. 7 | Neuronal activity triggers Ca²⁺ transients in tanycytes.**
**a**. *Top left*: GCaMP5g transients in tanycytes after evoking a single action potential (AP) in a synaptically-connected neuron in brain slices superfused with ACSF. Images show Ca²⁺ transients in tanycytes (frame response, orange-fire, *arrow*) subsequent to injecting current in a patch-clamped neuron (frame AP, neuron is marked by *arrow*; see also Supplementary Videos 1, 2). *Bottom left*: Representative traces of GCaMP5g relative fluorescence intensity transients (F/F₀) in tanycytes following an evoked AP (*arrows*). *Top right*: NBQX (20 μM) blocked GCaMP5g-encoded Ca²⁺ transients in tanycytes (frame response, orange-fire, *arrow*) following AP induction in the same neuron (frame AP; see also Supplementary Video 3). *Bottom right*: Representative traces of GCaMP5g F/F₀ transients in tanycytes following evoked AP (*arrows*). Time was expressed as frames with inter-frame interval of 300 ms. **b**. AMPA (100 μM)-induced GCaMP5g-encoded Ca²⁺ transients in tanycytes. Confocal time series (20x) from 300-μm coronal hypothalamic slices from *Rax*-CreER^T2::PC-G5-tdT mice. *Left:* Representative frames of GCaMP5g basal activity as F/F₀ transients in tanycytes superfused with ACSF. *Middle*: AMPA (100 μM)-induced increase in GCaMP5g transients. *Right*: The effect of AMPA was abolished by NBQX (20 μM; see also Supplementary Videos 4–6). Time was expressed as frames with inter-frame interval of 600 ms. *Scale bar* = 20 μm. **c**. PTX (100 μM)-induced GCaMP5g-encoded Ca²⁺ transients in tanycytes. Confocal time series (20x) from *Rax*-CreER^T2::PC-G5-tdT mice as above. *Left:* Sequential frames of basal GCaMP5g activity (F/F₀) when tanycytes were superfused with ACSF. *Middle*: PTX (100 μM)-induced increase in GCaMP5g transients, which were substantially reduced when co-applying TTX (5 μM, *right*; see also Supplementary Videos 7–9). *Scale bars* = 20 μm. Time was expressed as frames with an inter-frame interval of 600 ms. **d**. *Top*: KCl (50 mM)-induced tanycyte activation. Confocal time series (20x) from 300-μm coronal hypothalamic slices from *Rax*-CreER^T2::PC-G5-tdT mice. *Left*: Representative frames of GCaMP5g basal activity (F/F₀) in tanycytes superfused with ACSF. *Right*: KCl (50 mM)-induced increase in GCaMP5g-encoded Ca²⁺ events in tanycytes (F/F₀, trace; see also Supplementary Videos 10, 11). *Scale bars* = 20 μm. Time was expressed as frames with an inter-frame interval of 600 ms.

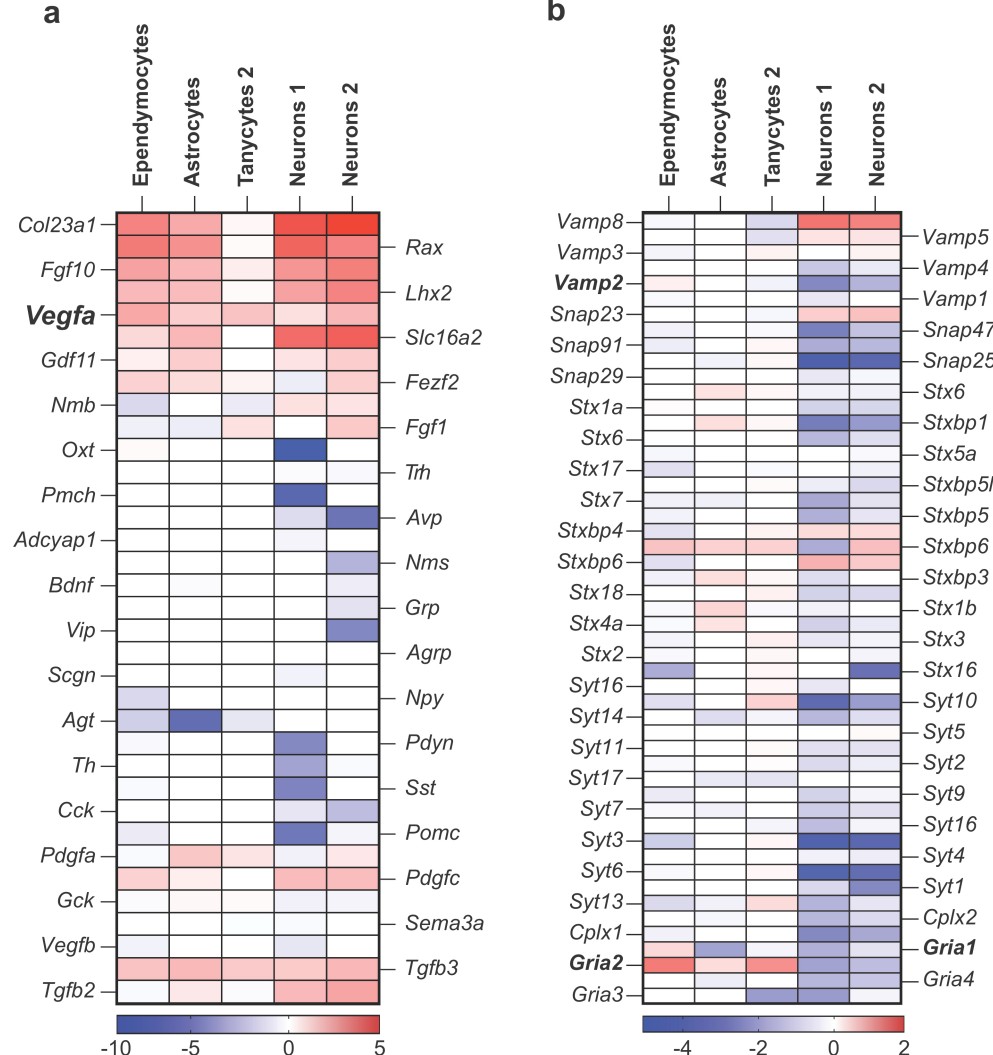

**Extended Data Fig. 8 | Neuropeptides, SNARE proteins, and AMPA receptor subunits in tanycytes. a**. Fold changes for differentially-expressed neuropeptides and signaling molecules in tanycytes (group '1') relative to ependymocytes, astrocytes, tanycytes '2', and two clusters of neurons after re-processing open-label single-cell RNA-seq data from ref. 40. Scaled fold changes with a false discovery rate of > 25% were included. *Expression scale*: 5 (red), 0 (white), −10 (blue). **b**. Fold changes for differentially-expressed SNARE complex-related genes and AMPA receptor subunits in tanycytes '1' relative to ependymocytes, astrocytes, tanycytes '2', and two neuronal clusters, with data re-processed from ref. 40. Scaled fold changes with a false discovery rate of > 25% were included. *Expression scale*: 2 (red), 0 (white), −2 and −4 (blue).

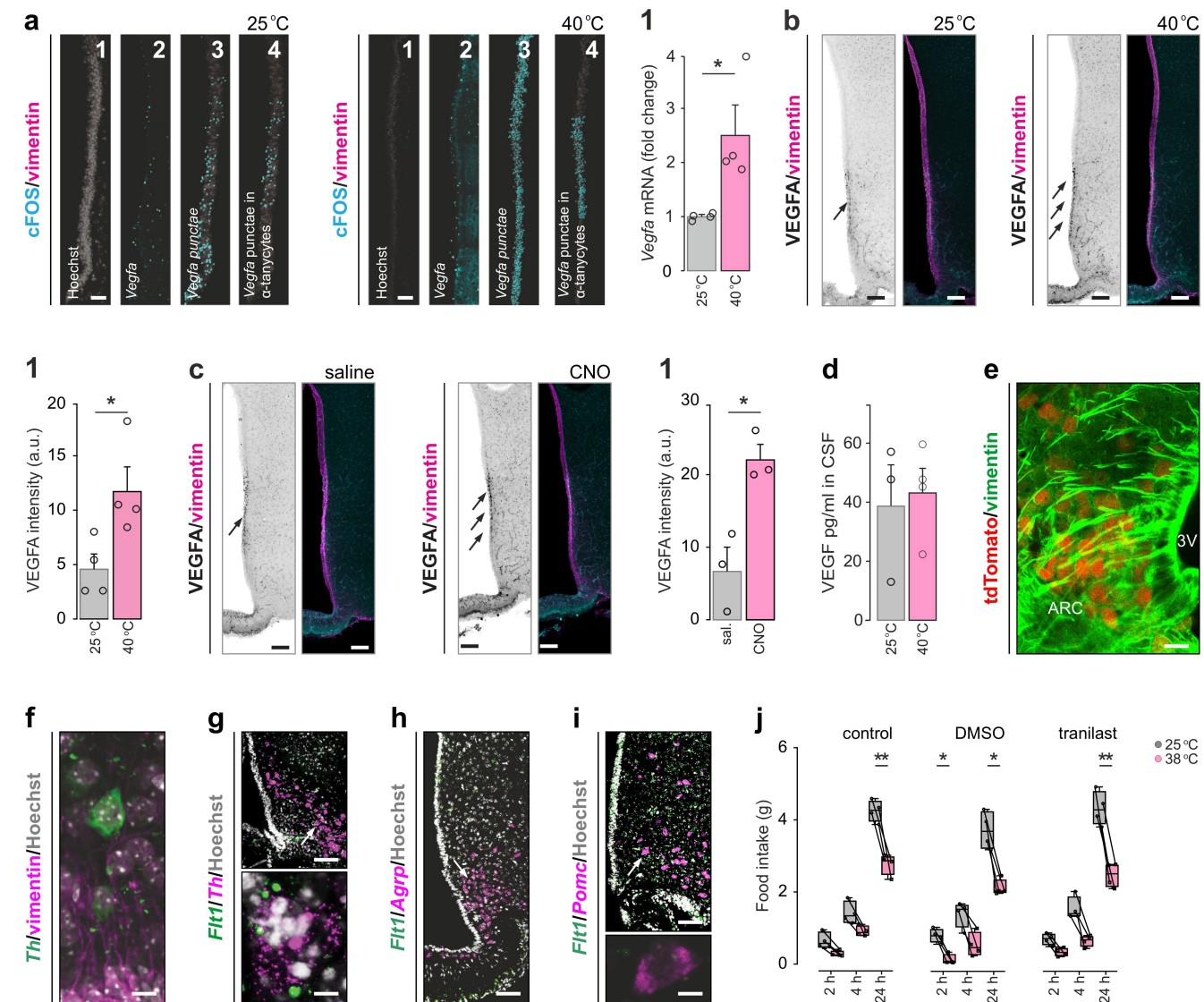

**Extended Data Fig. 9 | Tanycytes produce VEGFA upon heat exposure.**
**a**. Image series showing the analysis pipeline used to quantify the number of
*Vegfa* mRNA in tanycytes at both 25 °C and at 40 °C (1 h) *in vivo*. Images show
the surface reconstruction of tanycytes along the wall of the third ventricle
(counterstained by Hoechst 33,342, in grey) and the *Vegfa* mRNA probe (in
cyan). To determine the number of *Vegfa* mRNAs in apposition to the surface of
tanycytes, we used the *'find-spots-close-to-surface'* Imaris function. **1:** Confocal
image of tanycytes labelled with Hoechst 33,342 (in grey). **2:** *Vegfa* mRNA signal
(in cyan). **3:** *Vegfa 'spots'* (spheres, in cyan) rendered for reconstruction by
assigning each sphere to 0.4 µm x 0.4 µm in diameter over the *Vegfa* mRNA
signal along the Hoechst 33,342-labelled ventricular surface. **4:** Raster shows
the number of *Vegfa 'spots'* (in cyan) in α-tanycytes. *Scale bar* = 20 µm. **'1'**. Fold
change of *Vegfa* mRNA expression in the wall of the 3rd ventricle microdissected
from mouse hypothalami at 25 °C *vs*. 40 °C. Data in bar graphs correspond to
means ± s.e.m. with individual data points shown as circles; *n* = 4 mice/group;
*t* = 2.751; *p* = 0.033; Student's *t*-test. **b**. VEGFA immunoreactivity in tanycytes
lining the third ventricle (*arrows*), and its change after thermal manipulation.
*Left:* Grey-scaled VEGFA immunoreactivity. *Right*: Overlay of VEGFA (in cyan)
and vimentin (in pink). *Scale bars* = 70 µm. **'1'**. Quantitative data on VEGFA
immunoreactivity in tanycytes (as in panel **'b'**) from *n* = 4 mice/group; *p* < 0.05
(Student's *t*-test). **c**. VEGFA immunoreactivity in tanycytes (*arrows*), and its
change after chemogenetic activation by CNO. Saline was used as vehicle

control. *Left:* Grey-scaled VEGFA immunoreactivity. *Right*: Overlay of VEGFA (in
cyan) and vimentin (in pink). *Scale bars* = 70 µm. **'1'**. Quantitative data on VEGFA
immunoreactivity in tanycytes (as in panel **'c'**) from *n* = 4 mice/group; *p* < 0.05
(Student's *t*-test). **d**. VEGFA levels (pg/ml) in the CSF of rats exposed for 1h to
25 °C (in grey, *n* = 3) or 40 °C (in red, *n* = 4). Data in bar graphs show means ±
s.e.m. with open circles being individual data points; *t* = 0.306, *p* = 0.772;
Student's *t*-test. **e**. Coincident detection of tdTomato (corresponding to *Agrp*⁺
neurons; in red) and vimentin⁺ tanycytes (in green) in the arcuate nucleus
(ARC). *Scale bar* = 20 µm. **f**. Anatomical arrangement between vimentin⁺
tanycytes and *Th*⁺ neurons in the ARC. *Scale bar* = 5 µm. **g-i**. Confocal surveys of
the ARC for the distribution of *Th*⁺, *Agrp*⁺, and *Pomc*⁺ neurons (all in magenta),
and their co-expression of *Flt1* (or not, *see for Pomc*⁺ neurons). Hoechst 33,342
was used as nuclear counterstain (grey). *Scale bars* = 50 µm (*overviews*),
2 µm (i, *inset*), 1 µm (g, *inset*). **j**. *Left:* Changes in food intake upon exposure
to 38 °C *vs*. 25 °C. *Middle*: Food intake in mice acutely injected with DMSO
(0.001%; used as vehicle) prior to exposure to 38 °C. *Right*: Pre-treatment
with tranilast (20 mg/kg) prior to exposure to 40 °C. Note that tranilast did not
alter food intake at the concentration tested. Individual data points are shown
from *n* = 4 male mice/group. **p* < 0.05, ***p* < 0.01 (Student's *t*-test, pair-wise
comparisons). Data in bar graphs (**a'1'**,**b'1'**,**c'1'**,**d**) were expressed as means ±
s.e.m. throughout. Results in box-and-whisker plots (**j**) show medians ±
interquartile ranges.

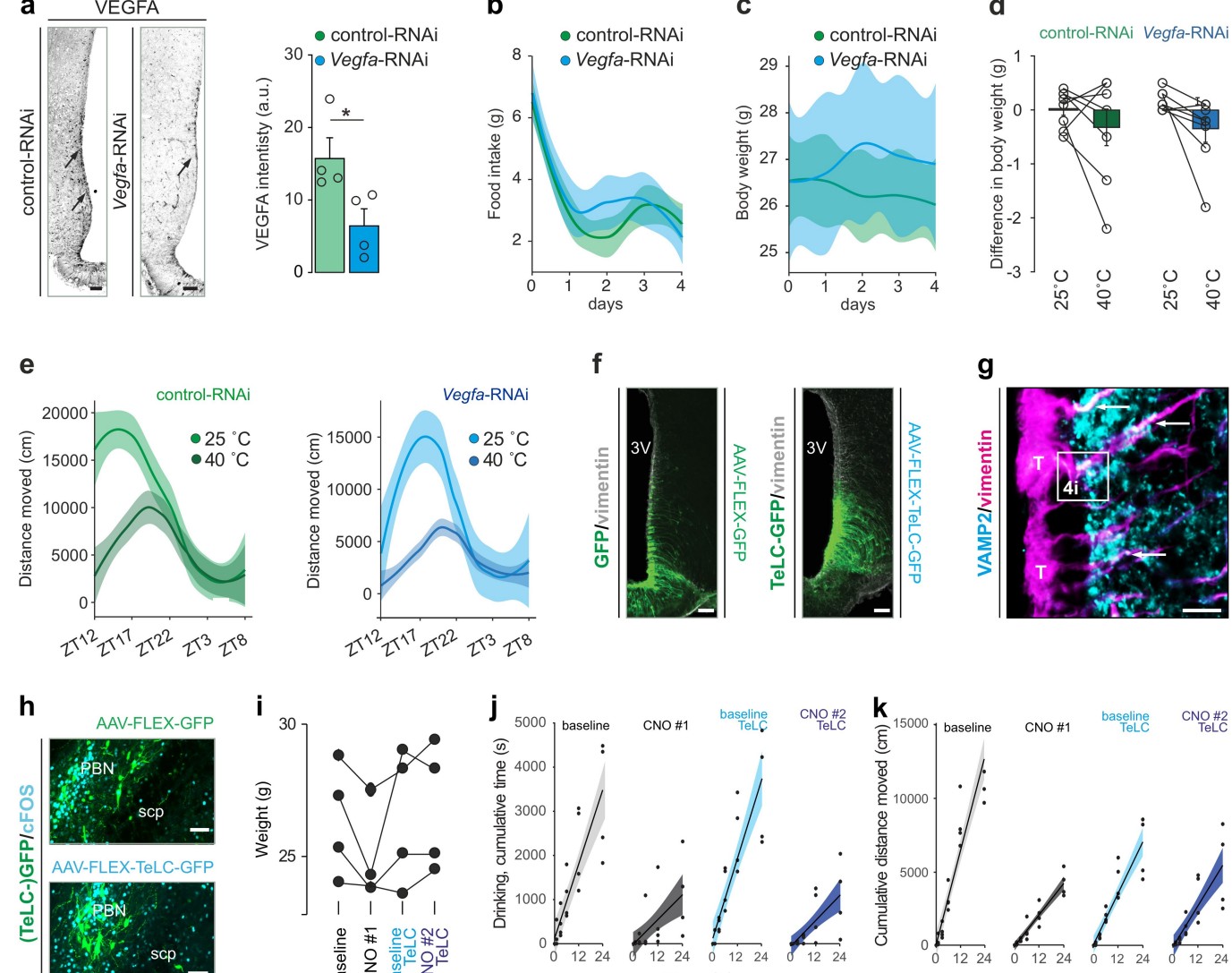

**Extended Data Fig. 10 | Tanycytes link the parabrachial nucleus to feeding.**
**a**. Left: Immunohistochemical detection of VEGFA (*arrows*) under control
conditions and after *Vegfa*-specific RNAi infusion in the 3rd ventricle, with
fluorescence intensity quantified in *n* = 4 mice/group (*right*). *$p < 0.05$
(Student's *t*-test). *Scale bars* = 20 μm. **b,c**. Timeline analysis (facet-wrap) of food
intake (**b**; *n* = 4/group) and body weight (**c**, *n* = 8/group) expressed in grams (g)
over 5 days prior acute thermal manipulation of mice injected with control or
*Vegfa*-RNAi. Repeated-measures ANOVA for underline{food intake}: interaction (RNAi *vs.*
days): $F = 1.457$; $p = 0.246$; days: $F = 57.150$; $p < 0.001$; RNAi: $F = 1.033$; $p = 0.349$,
and for underline{body weight}: interaction (RNAi *vs.* days): $F = 3.010$; $p = 0.004$; days:
$F = 1.427$; $p = 0.244$; RNAi: $F = 0.505$; $p = 0.489$. **d**. Difference in body weight
expressed in g after 24h at 25 °C and after being exposed to 40 °C in control *vs.*
*Vegfa*-RNAi. Data in bar graphs were expressed as means ± s.e.m. with
interconnected circles showing changes per animal (*n* = 8/group). Repeated-
measures ANOVA: interaction (RNAi *vs.* temperature), $F = 0.278$; $p = 0.614$;
temperature: $F = 3.783$, $p = 0.093$; RNAi: $F = 0.095$; $p = 0.767$. **e**. Diurnal activity
expressed as the mean distance moved (cm) 20h after exposure to 25 °C or 40 °C.
Time intervals were expressed as *zeitgeber* (ZT), with ZT12 and ZT0 coinciding
with the onset of the dark and light phases, respectively (*n* = 8/group). Repeated-
measures ANOVA: interaction (RNAi *vs.* temperature): $F = 0.049$; $p = 0.8249$;
interaction (RNAi *vs.* time): $F = 6.927$; $p < 0.001$; interaction (time *vs.*
temperature): $F = 2.510$; $p < 0.001$; time: $F = 15.210$; $p < 0.0001$; RNAi: $F = 12.620$;
$p = 0.001$; temperature: $F = 3.908$; $p = 0.058$. **f**. Benchmarking the specificity
of GFP (control) and TeLC-GFP expression in tanycytes after injection of AAV

particles in the third ventricle (3V). The vimentin immunosignal was color-
coded in light grey to enhance visual clarity, and to highlight somatic GFP
fluorescence in only the outermost cell layer of the ventricular wall. *Scale bars* =
20 μm. **g**. Vimentin+ tanycytes (in magenta) expressed VAMP2 (in cyan). Open
rectangle denotes the location of the orthogonal image stack shown in Fig. 4i.
Arrows point to tanycyte (T) processes with VAMP2 signal. *Scale bar* = 2 μm. **h**.
Histochemical controls of the successful transduction of PBN neurons with
AAVs expressing either GFP only or a TeLC-GFP fusion construct. *Abbreviation*:
scp, superior cerebellar peduncle. *Scale bar* = 70 μm. **i**. Individual bodyweights
across the phases of behavioral testing (*n* = 4; $F = 3.504$; $p = 0.148$; repeated-
measures one-way ANOVA). **j**. Facet-wrap timeline plot of cumulative drinking
time (s) during 24-h of baseline, in mice exposed to CNO (3 mg kg⁻¹; CNO 1),
repeated baseline after TeLC delivery and recombination by 4-hydroxytamoxifen
(baseline TeLC), and in mice with AAV2-FLEX-TeLC-GFP recombined in tanycytes
and injected with CNO (3 mg/g; 'CNO #2 TeLC', in blue), *n* = 4; repeated-measures
two-way ANOVA: interaction: $F = 5.281$; $p < 0.0001$; time (24 h): $F = 58.510$;
$p < 0.0001$; treatment: $F = 6.153$; $p = 0.009$; subject: $F = 4.936$; $p < 0.0001$.
**k**. Facet-wrap timeline plot of the cumulative distance moved (cm). Groups
were identical to (**i** and **j**); *n* = 4; repeated-measures two-way ANOVA: interaction:
$F = 11.130$; $p < 0.0001$; time (24 h), $F = 48.930$; $p < 0.0001$; treatment: $F = 47.570$;
$p < 0.0001$; subject: $F = 3.381$; $p = 0.0003$. Data in facet-wrap plots were
expressed as means ± 95% confidence interval of the s.d. throughout. Solid
circles represent individual data points at time and treatment.

**Extended Data Table 1 | Proteins detected by mass spectrometry in primary cultures of tanycytes**

| Protein ID | Protein names | Gene names | Peptide counts (all) |
|---|---|:---:|:---:|
| P63014 | Retinal homeobox protein Rx | *Rax* | 8 |
| P31000 | Vimentin | *Vim* | 46 |
| P21263 | Nestin | *Nes* | 40 |
| **Q9WUD2** | **Transient receptor potential cation channel subfamily V member 2** | ***Trpv2*** | **5** |
| P63045 | Vesicle-associated membrane protein 2 | *Vamp2* | 3 |
| Q9JHW5 | Vesicle-associated membrane protein 7 | *Vamp7* | 7 |
| P63025 | Vesicle-associated membrane protein 3 | *Vamp3* | 4 |
| Q9WUF4 | Vesicle-associated membrane protein 8 | *Vamp8* | 4 |

# Reporting Summary

## Statistics

For all statistical analyses, confirm that the following items are present in the figure legend, table legend, main text, or Methods section.

| n/a | Confirmed | |
|---|---|---|
| ☐ | ☒ | The exact sample size (*n*) for each experimental group/condition, given as a discrete number and unit of measurement |
| ☐ | ☒ | A statement on whether measurements were taken from distinct samples or whether the same sample was measured repeatedly |
| ☐ | ☒ | The statistical test(s) used AND whether they are one- or two-sided *Only common tests should be described solely by name; describe more complex techniques in the Methods section.* |
| ☒ | ☐ | A description of all covariates tested |
| ☐ | ☒ | A description of any assumptions or corrections, such as tests of normality and adjustment for multiple comparisons |
| ☐ | ☒ | A full description of the statistical parameters including central tendency (e.g. means) or other basic estimates (e.g. regression coefficient) AND variation (e.g. standard deviation) or associated estimates of uncertainty (e.g. confidence intervals) |
| ☐ | ☒ | For null hypothesis testing, the test statistic (e.g. *F*, *t*, *r*) with confidence intervals, effect sizes, degrees of freedom and *P* value noted *Give P values as exact values whenever suitable.* |
| ☒ | ☐ | For Bayesian analysis, information on the choice of priors and Markov chain Monte Carlo settings |
| ☒ | ☐ | For hierarchical and complex designs, identification of the appropriate level for tests and full reporting of outcomes |
| ☒ | ☐ | Estimates of effect sizes (e.g. Cohen's *d*, Pearson's *r*), indicating how they were calculated |

*Our web collection on statistics for biologists contains articles on many of the points above.*

## Software and code

Policy information about availability of computer code

| Data collection | 1) Electrophysiological data were collected using Clampex 10.7 (Molecular Devices); Clampfit 10.7, and PatchMaster Next (Heka). 2) Confocal image acquisition was done using the Zen Black Edition (Zeiss) software package. 3) Calcium imaging data were collected using the Visiview Software (version 3.0.3.0; Visitron Systems). 4) qPCR data were collected by the Bio-Rad CFX Manager software (version 3.1, Bio-Rad). 5) Behavioral data were captured with EthoVision XT14 (Noldus). 6) Primers were designed using Primer3web, version 4.1.0. |
|---|---|
| Data analysis | Images were analyzed using Imaris x64 9.0.2 (Bitplane) and Fiji 1.52e (GNU General Public Licence, https://imagej.net/Fiji). Electrophysiology data (EPSCs) were analyzed by the Mini Analisys Program (version 6.0; Synaptosoft). Data were processed in Microsot Excel (version 16.79.2), and statistically analyzed by GraphPad Prism 8.0.2 (GraphPad Software Inc.). |

For manuscripts utilizing custom algorithms or software that are central to the research but not yet described in published literature, software must be made available to editors and reviewers. We strongly encourage code deposition in a community repository (e.g. GitHub). See the Nature Portfolio guidelines for submitting code & software for further information.

## Data

Policy information about availability of data

All manuscripts must include a data availability statement. This statement should provide the following information, where applicable:
- Accession codes, unique identifiers, or web links for publicly available datasets
- A description of any restrictions on data availability
- For clinical datasets or third party data, please ensure that the statement adheres to our policy

> No data were included that shall be placed in a public repository. Single-cell RNA-seq data were reprocessed from public libraries as identified. All individual data were presented as single points, with the corresponding "raw data" and statistical analyses made available in the Source Data File that is part of this submission.

## Human research participants

Policy information about studies involving human research participants and Sex and Gender in Research.

| | |
|---|---|
| Reporting on sex and gender | n/a |
| Population characteristics | n/a |
| Recruitment | n/a |
| Ethics oversight | n/a |

Note that full information on the approval of the study protocol must also be provided in the manuscript.

# Field-specific reporting

Please select the one below that is the best fit for your research. If you are not sure, read the appropriate sections before making your selection.

☒ Life sciences        ☐ Behavioural & social sciences        ☐ Ecological, evolutionary & environmental sciences

For a reference copy of the document with all sections, see nature.com/documents/nr-reporting-summary-flat.pdf

# Life sciences study design

All studies must disclose on these points even when the disclosure is negative.

| | |
|---|---|
| Sample size | The sample size has been chosen as specified in previous publications PMID: 24121436; PMID: 32917598. |
| Data exclusions | No data were excluded from the analysis. |
| Replication | The experiments reported here were minimally performed in duplicates (biological repeats in two (or more) independent experimental settings). All attempts of replication were successful. |
| Randomization | Experimental animals used in this study were not randomized. This is mainly because of their complex genetic features and since representatives of all experimental groups were tested in parallel to minimize bias (e.g., in multi-chamber phenotyper systems). |
| Blinding | Experimenters were not blinded to the experimental conditions, because experimental manipulations were measured in automated systems and run as consecutive experiments with the re-use of the same animals (i.e. baseline at start, experimental conditions thereafter. Moreover, the complex genetics of the animals and the relatively low numbers of the experimental subjects necessitated clear group assignments. |

# Reporting for specific materials, systems and methods

We require information from authors about some types of materials, experimental systems and methods used in many studies. Here, indicate whether each material, system or method listed is relevant to your study. If you are not sure if a list item applies to your research, read the appropriate section before selecting a response.

## Materials & experimental systems

| n/a | Involved in the study |
|---|---|
| ☐ | ☒ Antibodies |
| ☒ | ☐ Eukaryotic cell lines |
| ☒ | ☐ Palaeontology and archaeology |
| ☐ | ☒ Animals and other organisms |
| ☒ | ☐ Clinical data |
| ☒ | ☐ Dual use research of concern |

## Methods

| n/a | Involved in the study |
|---|---|
| ☒ | ☐ ChIP-seq |
| ☒ | ☐ Flow cytometry |
| ☒ | ☐ MRI-based neuroimaging |

## Antibodies

| | |
|---|---|
| Antibodies used | Primary antibodies:<br>guinea-pig anti-cFOS (1:1,000; Synaptic Systems, #226005)<br>rabbit anti-cFOS (1:2,000; Synaptic Systems, #226003)<br>rabbit anti-DsRed (1:200; Clontech/Takara, #632496)<br>rabbit anti-RFP, biotin-conjugated (1:1000; Rockland, #600-406-379)<br>chicken anti-RFP (1:500; Rockland, #600-901-379)<br>goat anti-GFP (1:200; Abcam, #ab6662)<br>goat anti-mCherry (1:500; Antibodies Online, #ABIN1440058)<br>guinea-pig anti-GluA1 (1:100; Alomone labs, #AGP-009)<br>rabbit anti-GluA2 (1:100; Alomone labs, #AGC-005)<br>mouse anti-nestin (clone rat-401; 1:500; Millipore, #MAB353)<br>chicken anti-NeuN (1:500; Millipore, #ABN91)<br>rabbit anti-p44/42 MAPK (pERK1/2) (Thr202/Tyr204; 1:200; Cell Signaling Technology, #9101S)<br>rabbit anti-tyrosine hydroxylase (1:500; Millipore, #AB152)<br>rabbit anti-VGLUT2 (1:200; Synaptic Systems, #135403) or as gift of M. Watanabe (see Ref. 59 for quality controls)<br>chicken anti-vimentin (1:500; Synaptic Systems, #172006)<br>goat anti-VEGFA (1:100; R&D Systems, #AF-493-NA)<br><br>Secondary antibodies:<br>Alexa Fluor 488 donkey anti-rabbit IgG (1:2,000; Invitrogen, #AB21206)<br>Alexa Fluor 488-conjugated AffiniPure donkey anti-guinea pig IgG (1:300; Jackson ImmunoResearch, #706-545-148)<br>Alexa Fluor 488-conjugated AffiniPure donkey anti-mouse IgG (1:300; Jackson Immu-noResearch, #715-545-151)<br>Alexa Fluor 647-conjugated AffiniPure donkey anti-rabbit IgG (1:300; Jackson ImmunoResearch, #711-605-152)<br>Cy2-conjugated AffiniPure donkey anti-goat IgG (1:300; Jackson ImmunoResearch, #705-225-147)<br>Cy2-conjugated AffiniPure donkey anti-rabbit IgG (1:300; Jackson ImmunoResearch, #711-225-152)<br>Cy3-conjugated AffiniPure don-key anti-chicken IgG (1:300; Jackson ImmunoResearch, #703-165-155)<br>Cy3-conjugated Affin-iPure donkey anti-guinea pig IgG (1:300; Jackson ImmunoResearch, #706-165-148)<br>Cy3-conjugated AffiniPure donkey anti-rabbit IgG (1:300; Jackson ImmunoResearch, #711-165-152)<br>Cy5-conjugated AffiniPure donkey anti-chicken IgG (1:300; Jackson ImmunoResearch, #703-175-155)<br>Cy5-conjugated AffiniPure donkey anti-guinea pig IgG (1:300; Jackson Immu-noResearch, #706-175-148)<br>Cy5-conjugated streptavidin (1:200; Jackson ImmunoResearch, #016-170-084)<br><br>Hoechst 33,342 (1:10,000; Sigma Aldrich) was used as nuclear counterstain. |
| Validation | We refer to the RRID portal for antibodies (https://scicrunch.org/resources/) for validation of the immunoreagents described here. For each antibody, the RRID repository number provides exhaustive data on the host species, titers, links to manufacturer websites, and relevant citations.<br><br>Primary antibodies, as specified in the manuscript:<br>guinea-pig anti-cFOS (Synaptic Systems Cat# 226 005, RRID:AB_2800522)<br>rabbit anti-cFOS (Synaptic Systems Cat# 226 003, RRID:AB_2231974)<br>rabbit anti-DsRed (Takara Bio Cat# 632496, RRID:AB_10013483)<br>rabbit anti-RFP. biotin-conjugated (Rockland, #600-406-379; RRID:AB_828390)<br>chicken anti-RFP (Rockland, #600-901-379, RRID:AB_10704808)<br>goat anti-mCherry (Antibodies Online, #ABIN1440058. Used as recommeded by the manufacturer: https://www.antibodies-online.com/antibody/1440058/anti-mCherry+Fluorescent+Protein+antibody/. Recently used in  https://doi.org/10.1101/2023.02.03.527010).<br>goat anti-GFP (Abcam Cat# ab6662, RRID:AB_305635)<br>guinea-pig anti-GluA1 (Alomone Labs Cat# AGP-009, RRID:AB_2340961)<br>rabbit anti-GluA2 (Alomone Labs Cat# AGC-005, RRID:AB_2039881)<br>mouse anti-nestin (Millipore Cat# MAB353, RRID:AB_94911)<br>chicken anti-NeuN (Millipore Cat# ABN91, RRID:AB_11205760)<br>rabbit anti-p44/42 MAPK (pERK1/2), used as recommended by the manufacturer in:https://www.cellsignal.de/products/primary-antibodies/phospho-p44-42-mapk-erk1-2-thr202-tyr204-antibody/9101?gclid=Cj0KCQjwuuKXBhCRARIsAC-gM0iy9SRAUFwQ4SrB439mSrmAl1yb6WrECKWoYxjDx5mjlumgoKvFk3saAh_dEALw_wcB&gclsrc=aw.ds. In here, 10 different publications are cited that have validated the antibody.<br>rabbit anti-tyrosine hydroxylase (Millipore Cat# AB152, RRID:AB_390204)<br>rabbit anti-VGLUT2 (Synaptic Systems Cat# 135 403, RRID:AB_887883) or as gift of M. Watanabe as described in: https://doi.org/10.1046/j.1460-9568.2003.02698.x |

chicken anti-vimentin (Synaptic Systems Cat# 172 006, RRID:AB_2800525)
goat anti-VEGFA (1:100; R&D Systems, #AF-493-NA) Used as recommended by manufacturer in https://www.rndsystems.com/products/mouse-vegf-164-antibody_af-493-na?utm_source=antibodypedia&utm_medium=referral&utm_campaign=product&utm_term=primaryantibodies

# Animals and other research organisms

Policy information about studies involving animals; ARRIVE guidelines recommended for reporting animal research, and Sex and Gender in Research

| | |
|---|---|
| Laboratory animals | For behavioral, electrophysiological, and biochemical experiments only male mice aged P60-P100 days were used. The following mice (wild-type and transgenic) and their crosses as specified were included in our study:<br>C57Bl6/J (as wild-type, local breeding)<br>C57Bl6/N (as wild-type, local breeding)<br>Raxtm1.1(cre/ERT2)Sbls/J (The Jackson Laboratory #025521)<br>B6.Cg-Gt(ROSA)26Sortm14(CAG-tdTomato)Hze/J (The Jackson Laboratory #007914)<br>B6;129P2-Mapttm2Arbr/J (The Jackson Laboratory #021162)<br>B6;129S6-Polr2aTn(pb-CAG-GCaMP5g,-tdTomato)Tvrd/J (The Jackson Laboratory #024477)<br>B6;129S-Slc17a6tm1.1(flpo)Hze/J (The Jackson Laboratory #030212)<br><br>Animals were group housed at standard laboratory conditions unless specified otherwise. Behavioral experiments were performed after sufficiently-long periods of acclimatization to new environments and single housing, as specified in the Methods section.<br><br>For primary cultures of tanycytes, both male and female Wistar rats (at P10) were used.<br>To extract cerebrospinal fluid, male Wistar rats at P60 (Janvier) were used. |
| Wild animals | No wild animals were used in the study. |
| Reporting on sex | Males and females were used in this study. Sex was specified in each relevant figure/figure panel. |
| Field-collected samples | No field collected samples were used in the study |
| Ethics oversight | Experimental procedures on mice conformed to the 2010/63/EU directive and were approved by the Austrian Ministry of Education, Science and Research (66.009/0145-WF/II/3b/2014 and 66.009/0277-WF/V3b/2017). All procedures were planned to reduce suffering, as well as animal numbers. |

Note that full information on the approval of the study protocol must also be provided in the manuscript.

