## [Peer Review File · Nature]

Manuscript Title: A brainstem-hypothalamus neuronal circuit reduces feeding up-on heat exposure

Reviewer Comments & Author Rebuttals

Reviewer Reports on the Initial Version:

Referees' comments:

Referee #1 (Remarks to the Author):

Harkany and colleagues here report studies investigating the neural mechanism connecting the sensing of environmental temperature change to the control of food intake. They show that tanycytes are “activated” upon acute heat and are necessary to change food intake. Data are also included which indicate that thermosensing glutamatergic neurons of the PBN innervate tanycytes either directly or through second-order hypothalamic neurons. In addition the authors report that vascular growth factor-A (VEGFA) may be involved in the molecular underpinnings of this process. VEGFA is released directly in the arcuate nucleus to elevate the spike threshold of Flt1+ dopamine and Agouti-related peptide-containing neurons, thereby priming net anorexigenic output. Acute heat exposure and chemogenetic activation of glutamatergic PBN neurons at thermoneutrality are shown to reduce food intake for hours, dependant on tanycyte exocytosis processes. The authors propose that tanycytes can convert neuronal activity into a metabolic code to produce antagonistic associations between sensory and metabolic processes. This process may be facilitating the survival of mammals (in this case mice) under adverse conditions.

This elegant set of studies and novel observations is adding up to a really intriguing overall story. The authors have used an impressive array of techniques to examine mechanistic processes such as the tanycyte release of VEGFA in the ARC or the activation of tanycyte calcium signaling induced by PBN neurons. The EM images showing the morphology of the tanycytes are impressive and are adding depth. There are no major concerns, however there are a number of minor questions, the authors may want to take into consideration for a potential revision of the manuscript:

This reviewer may have missed it, but while the authors show independently that tanycytes release VEGFA and that PBN neurons activate tanycytes, they didn't seem to show the tanycytes release VEGFA after PBN neuron activation? Is there a reason for omitting this aspect?

Are the authors certain that the lower number of feeding bouts immediately following an acute thermal stress is important and specific? It seems entirely possible that all activity is simply reduced by the thermal stress, including feeding activity.

The link to climate change link feels somewhat forced/overstated. If that aspect was indeed the main motivation for these studies, then chronic heat exposure might be much more relevant than acute exposure. Moreover, if consequences of chronic heat exposure on mammalian health is being examined as a health threat, this reviewer is not convinced decreases food intake is ranking particularly high on that list? Potentially these aspects should be reframed – at least they are easy to misunderstand.

Are the authors sure that an animal that lives its whole life at 22 degrees is a good model for the response to a temperature stress at 40 degrees? If this is interpreted as being at all relevant for human pathophysiology another model might be more useful where lifelong experience of and response to environmental temperature changes (inside/outside, winter/summer, day/night) plays a role.

The title of the manuscript has to include the species exclusively used. It should say: "...in mice". The authors are repeatedly referring to food seeking. Has that actually been studied here? Or just time spent feeding? That is not the same behavioral endpoint. Is the impact of heat exposure on food intake dependent on the type of food?

Why has the experimental setting been limited to just one temperature? Shouldn't there several temperatures be studied? Why just 40 degrees Celsius? Does cold exposure do the opposite? Or does that have comparable effects?

Referee #2 (Remarks to the Author):

This study's aim was to dissect the mechanism/s by which acute heat stress induces temporary anorexia. In effect, how is thermosensing by neuronal clusters in the Parabrachial nucleus transduced and translated into temporary suppression of appetite. A range of in vivo and in vitro studies were followed to establish whether Tanycytes constitute a node in this pathway by asking whether tanycytes are innervated by PBN neurons and connected to local hypothalamic neurons, and whether experimental heat stress either via ambient temperature alternations, or mimicked by chemogenetic manipulations can induce anorexia. In due course, a biased approach was followed to evidence that Vascular Endothelial Growth Factor (VEGF) release from tanycytes, to act on local hypothalamic neurons, may be a critical mediator of this pathway.

The idea is original as hypothalamus sits at the core of energy uptake neural circuits. The parenchyma of medio-basal hypothalamus into which tanycytes project is a very crowded interface, with the involvement of a multitude of neuronal subtypes, inputs from extra-hypothalamic regions (PBN included), glia cells and microcapillaries (which are the best documented targets of tanycyte end-feed). Therefore, it would not be surprising that tanycyte process come within micrometers of neurons. The study has built a strong case, but has not proven that tanycytes actually synapse with neurons (and vice versa). This doesn't negate the findings but is a critical premise. I.e. Non-synaptic communications could also play a critical role and this has not been explored in depth.

The experimental paradigms and approaches are interesting but uneven in quality. (see detailed comments below).

The manuscript is also uneven in quality and there is little reflection or exploration of concepts such as adaptation to acute heat stress. The impact of the work other than climate-change associated

heat stress is not explored. In the context of environmental heat stress, the elderly are the most vulnerable to heat stress-induced anorexia and an ageing component is missing here. Similarly, previously-described pathways linking transient anorexia to illness (explored by David Anderson's group), also involving PBN, is not discussed.

The study has merit but, in several places lacks depth and rigour.

Suggestions and comments

Title and throughout text:

As a subset of tanycytes are implicated in the described mechanisms, the title and text should consistently stress that as a heterogeneous population, only a subset (approximately 50% according to figure 3 data) are implicated.

Figure 1 (and related data).

It was surprising that conclusions were based on n= 3-4 mice per treatment. Typically, these sorts of behaviour tests involve at least n=6 per group and in such a critical paradigm both sexes would be analysed. Not indicated whether the mice were litter mates? Where they weighed before experiments to show equivalence etc. Can general reduced 'activity' be ruled out downstream of heat stress?

1C & 1D – Vimentin expression/label intensity also appears to be elevated under heat stress (40 C compared to 25 C). Is this a real effect or artefact?

Figure 2 (and related data)

2C - What was the time course between ventricular viral injection and end point analysis. Could the mcherry+ve neurons be derived from tanycytes, and not necessarily connected to them as the figure concludes? There is no information on this in methods.

2c, d, j. The figure and experimental approach is already complicated and so false colouring should be uniform to save further confusion. Perhaps Mcherry consistently in red, Vimentin in green (in J)

Extended data 3

- Please show examples of beta2 tanycytes association with Vglut 2 terminals.
- GluA1 is not restricted to beta tanycytes and tapers off to include some alpha 2 tanycyte domain as well.
- The quantifications could also include proximal versus distal tanycyte processes as the major glut terminal in Figure 1 was noted as close to soma.

Figure 4d

Is there any evidence that tanycytes have a 1:1 or 1:many relationship with neurons? No evidence has been produced to this effect. It should be clarified or noted, as stoichiometry is likely to be a critical factor in eliciting a signal in any signal transduction pathway.

Extended data 5

Statement in line 237/238 is incorrect. A glance Ex/data Figure 5 show that the three cited factors are in fact also expressed at similar levels by some neuronal populations, as well as tanycytes. Text should also mention that in addition to vegf, Tgfb3 and Tgb2, Fgf1 and Pdfga are clearly the other potential strong candidates for signal transduction (at least according the heat map).

Methods

Details of In situ hybridization studies missing.

Referee #3 (Remarks to the Author):

In this study, Benevento et al. propose a role for tanycytes in linking heat sensation to food intake suppression. The authors first show that acute heat exposure reduces food consumption while activating tanycytes. They then demonstrate anatomical and functional connections between neurons in the parabrachial nucleus and tanycytes using electron microscopy imaging, transsynaptic tracing, and chemogenetics. Patch-clamp electrophysiology and ex vivo calcium imaging experiments further revealed that tanycytes can respond to synaptic excitation from glutamatergic neurons, though the role of glutamatergic parabrachial projection here remains unknown. They next show when activated, tanycytes likely release VEGFA to the arcuate nucleus and increase the threshold current of arcuate nucleus neurons. Finally, they found that tanycytes are required for parabrachial nucleus neuron stimulation to suppress appetite.

The interaction of neurons and non-neuronal cells is a topic of major interest. The authors propose that neurons may directly regulate tanycytic activity via synapse-like structures, and that tanycytes may also modulate neuronal activity. Furthermore, while recent papers demonstrated the neural mechanisms underlying cold-induced food seeking behavior (Yang et al., 2021; Deem et al., 2020), the neural basis underlying heat-induced feeding suppression remains unknown; thus, this discovery—if properly supported—is timely and fills a knowledge gap. The current study could have resulted in a significant advancement, but it appears that there are significant limitations with regard to the study design and data interpretation, as follows.

Major points:

1. According to the authors, the proposed circuit, which includes PBN neurons, tanycytes, and arcuate nucleus neurons, serves as a neural substrate that converts heat sensation into appetite suppression. However, it is unclear whether the proposed circuit is required for heat-induced food intake reduction. Are tanycytes necessary for this phenomenon? Demonstrating the necessity of this circuit is necessary for their claims. This can be accomplished by measuring food intake after a thermal challenge during acute inhibition and/or chronic tanycyte ablation.
2. The authors also suggested that thermosensory signals are relayed to tanycytes by parabrachial glutamatergic neurons (e.g., "thermosensing glutamatergic neurons of the PBN innervate tanycytes"). They did show that thermal challenge increases Fos and pErk1/2 levels in tanycytes, but it is unclear whether parabrachial neurons are required for tanycytic activation after heat exposure. Indeed, the authors did not demonstrate that tanycytes in the PBN are innervated by "thermosensing" glutamatergic neurons. Given that primary tanycytes express thermosensitive TRPV

channels, it appears necessary to show that the parabrachial nucleus is essential for tancytic activation in response to thermal challenge.

3. The authors argue that monosynaptic PBN input to tancytes mediates the heat-induced orexigenic effects and mention the possibility of second-order relay by intrahypothalamic neurons to tancyte in the discussion section. However, evidence suggesting that tancytes are directly modulated by PBN projection is insufficient to underline the importance of the monosynaptic PBN-tancyte connections. It would be better to directly demonstrate modulation of tancyte by direct PBN projection, such as ChR2-assisted circuit mapping (CRACM) experiment (that expresses ChR2 virus in the PBN and records from tancyte while activating the PBN terminal) or the author should tone down their arguments.

4. In Fig. 3, the authors illustrate the differences between α -tancytes and β -tancytes in a variety of ways (e.g. VGLUT2+ input density, AMPAR subunit expression, and EPSC features). However, there is no discussion about the significance of tancyte type distinctions in reduced food seeking by heat perception. Also, the authors should explain why they focused on α -tancytes (despite the fact that EPSC intensity is higher in β -tancytes) and how their data that tancytes in the caudal subdivision preferentially accumulate pErk1/2 after 40°C exposure can be reconciled (Fig. 1d).

5. The authors attempt to emphasize the importance of this study in the Abstract and Introduction sections by mentioning climate change and global warming, which represent an increase in environmental temperature over longer timescales. This is misleading, especially given that mice are only exposed to hot environments for a few hours in their experiments. They should revise the Abstract and Introduction sections in order to properly summarize and introduce the research (i.e. the function of tancytes during acute heat exposure).

Minor points:

1. In Extended Data Fig. 1a, the authors found no significant change in skin temperature after a 3-h exposure to 40°C. However, this observation is inconsistent with ref 16. As shown in Fig. 1a of ref 16, the skin temperature of mice increases as the ambient temperature rises. Please explain this discrepancy.

2. The authors use the phrase "food seeking" throughout the manuscript. This refers to the appetitive phase of feeding rather than the consummatory phase. What was measured, however, was the amount of time spent eating, which is indicative of the consummatory phase of feeding behavior. Please change the phrase "food seeking" to something more appropriate, such as "food intake" or "food consumption."

3. PBN neurons that project to the arcuate nucleus (Fig. 2e) and tancytes (Fig. 2f) appear to have distinct anatomical locations. It would be useful to have images that could identify the LPB subregion.

4. Please add quantitative information to the data that is currently only presented as images (e.g. Fig. 2h, i).

5. Few casual words such as "liquor". in Line 40, may be better rephrased.

6. Lines 164 and 811. Fig. 2l should be renamed Fig. 2j.

7. Lines 242 and 243. Fig. 5a,b and 5c should be renamed Fig. 4a,b and 4c.

8. Line 154. "Effernets" should be spelled "efferents".

9. Line 272. "VGEFA" should be spelled "VEGFA".

Referee #4 (Remarks to the Author):

In this study, Benevento et al describe a neural mechanism that translates environmental heat into reduced appetite. In particular, using a combination of imaging, electrophysiology and AAV-based circuit mapping, the authors claim that thermosensing glutamatergic neurons of the PBN send efferents to tanycytes, which in turn release VEGFA into the arcuate nucleus of the hypothalamus. This VEGFA influences dopamine positive and AgRP neurons thus modulating food intake. Conceptually, the message of the study is very interesting and provides a new perspective on the functions of tanycytes as integrators of environmental signals (i.e. nutrients and now thermal responses). However, the study as it is has a number of important flaws that reduces the enthusiasm of this reviewer. As a general criticism, it is worth mentioning that the manuscript is full of overstatements and conclusions that are not actually supported by the data. Most of the data is associative (not causal) and lack specificity (see below). Another important limitation is regarding the images. Many of them are not representative, they are taken at different magnification, different bregma positions and are not convincing. Below is a list of concerns and suggestions that may improve the study:

- Fig.1A. Total food intake, rather than number of bouts should be shown.
- Fig 1C. It would be nice to see a picture taken at reduced magnification to appreciate the entire (or larger portion) wall of the 3rd ventricle.
- Ext Fig 1B. These images actually do not support the text. In fact, it seems that 40°C decreases FOS in the 3V wall of the ARC and also in the ARC region.
- Ext Fig 1D. There is also increased FOS in both alpha and beta tanycytes (text only states alpha tanycytes).
- Fig 2. In general, images here should be improved by adding higher magnification insets (C-D), showing homologous areas (E-F), spanning larger areas (H-I) or providing additional images (J).
- The authors do not provide evidence that the VGLUT2 synapses are coming from PBN neurons. They should perform experiments injecting Cre-dependent synaptophysin AAVs into the PBN and count the number of synapses in tanycytes.
- Lines 147-149 overstate their findings (as VGLUT2 nerve endings could be projected from many other regions than the PBN).
- Line 152- 153 is not demonstrated by the results.
- Ext FIG2B is wrongly placed.
- Ext FIG2D. PDYN staining is not exclusive of PBN neurons and could actually come from other neurons.
- FIG 3D. Images should be at the same magnification. It does not seem that GluA2 preferably stains alpha tanycytes.
- It is difficult to understand why the authors included the electrophysiology data shown in FIG 3. These results do not demonstrate that tanycytes are excited via glutamatergic PBN neurons. A more direct way to show that tanycytes respond to glutamate would be to perfuse actual glutamate in the electrophysiology setting (rather than receptor agonists). Statement in lines 202-203 is false.
- Ext FIG 3A-C. Again, authors do not demonstrate that VGLUT2 synapses are coming from PBN neurons.
- Line 237 (Ext FIG 5A) is not true. According to the heat map, VegfA is also expressed in other cell types at similar levels as tanycytes.
- Lines 242-245 the authors referred to FIG 5 when they meant FIG 4.

- Line 253. KCl depolarizes all cell types, not only tanycytes. This approach is too drastic and inespecific, and therefore VEGF in CSF may not necessarily come from tanycytes.
- Images Ext Fig 5 D-E are misleading. Larger fields with more POMC neurons should be shown (in E) and provide separate images for dyes.
- The authors should take advantage of publicly available sequencing data, as done in other sections of the manuscript, to provide evidence of the expression of Flt1 in diverse subsets of neurons. ISH data for AgRP should be shown as it seems it is an important mediator of the appetite effects of tanycytes.
- Lines 282-283 is an overinterpretation of the data.
- FIG 4G. Authors should also provide ultrastructural data showing TH cells in close apposition to tanycytes.
- In the model suggested by the authors, it is not clear how VEGFA released by tanycytes leads to reduced food intake. Is VEGFA inhibiting TH and AgRP neurons thus enhancing the anorexigenic output? The effects of VEGFA on the activity of TH and AGRP neurons should be assessed. Chemogenetic/optogenetic studies could be also performed in TH and/or AgRP neurons to demonstrate that they mediate the feeding effects of this circuit.
- To prove that is the VEGFA derived from tanycytes the link between thermo-sensitive PBN neurons and feeding, the authors should repeat the experiment shown in FIG 5 using mice lacking VEGFA in tanycytes.
- Do all thermo-sensitive PBN neurons send projections to tanycytes? What is the molecular identity of glutamate PBN neurons that interact with tanycytes?

Author Rebuttals to Initial Comments:

Point-by-point responses to the Referees' concerns

(Nature-2022-08-13268)

Referee #1:

Thank you for your supportive view and helpful comments on our manuscript. We much appreciated learning that *'There are no major concerns, however there are a number of questions, the authors may want to take into consideration for a potential revision of the uscript'*. We have considered your queries carefully and addressed them exhaustively. Find out specific responses as follows:

Q1: 'This Referee may have missed it, but while the authors show independently that tanycytes release VEGFA and that PBN neurons activate tanycytes, they didn't seem to show the tanycytes release VEGFA after PBN neuron activation? Is there a reason for omitting this aspect?'

Thank you for this great question, which indeed was missing from the original manuscript. We did so, at least in part, because previous studies provided evidence that cFos mediates the transcriptional regulation of *Vegfa* (Catar R *et al.*, The proto-oncogene c-Fos transcriptionally regulates VEGF production during peritoneal inflammation. *Kidney Int.* (2013) 84:1119-28). Therefore, we have taken cFos expression in tanycytes as a surrogate (minimally) or even being causal to (maximally) priming *Vegfa* expression. We did so because acute heat and chemogenetic activation of PBN neurons phenocopied one another. **We have experimentally addressed your question by interfering with *Vegfa* expression in tanycytes through siRNAs. siRNA-mediated *Vegfa* loss-of-function was sufficient to occlude the reduction of food intake normally observed in mice following acute heat exposure. Data were shown in: Fig. 4h, ED9i-l.**

Q2: 'Are the authors certain that the lower number of feeding bouts immediately following an acute thermal stress is important and specific? It seems entirely possible that all activity is simply reduced by the thermal stress, including feeding activity.'

We appreciate your point because it queries the essential physiological significance of the parameter we have reported on (feeding bouts), particularly if it is a dependent variable related to general activity. To avoid any confusion, we switched to consistently and directly showing food intake, that is the difference in food consumed before and during a 24-h period after acute heat exposure. Throughout, and **in a manner independent of the sex of the experimental subjects, we observed that food intake was significantly reduced after thermal manipulation.** Moreover, we demonstrate that a change in food intake is independent of general activity because a reduction in locomotion persists for ~3h and then is gradually restored. **Data were shown in: Fig. ED1a,b and throughout.**

Q3: 'The link to climate change link feels somewhat forced/overstated. If that aspect was indeed the main motivation for these studies, then chronic heat exposure might be much more relevant than acute exposure. Moreover, if consequences of chronic heat exposure on mammalian health is being examined as a health threat, this Referee is not convinced decreases food intake is ranking particularly high on that list? Potentially these aspects should be re-framed – at least they are easy to misunderstand.'

We appreciate your concern. **We have revised the text and removed any potentially contentious reference** to climate changes and other existential threats.

Q4: 'Are the authors sure that an animal that lives its whole life at 22 degrees is a good model for the response to a temperature stress at 40 degrees? If this is interpreted as being at all relevant for human pathophysiology another model might be more useful where lifelong experience of and response to environmental temperature changes (inside/outside, winter/summer, day/night) plays a role.'

The Referee brought up an excellent point. Indeed, laboratory mice spend most of their lives at a well-regulated temperature of around 22 °C. As such, exposure to 40 °C constitutes a rather immediate, short-term temperature stress (*see data in Fig. ED1c*). This is precisely the reason why we chose this model as it accurately mimics a situation relevant to humans: in developed countries, people spend most of their time in a temperature-controlled (either heated or cooled) environment inside buildings with defined thermal regulation. Only when people step outside they are exposed to high temperatures of around 40 °C. Alternatively, they can choose to do so (as we have discussed) in a sauna (ultimately considering a dry sauna). Hence, we consider our murine model to be accurate for the mechanism we are describing, particularly since the change in core body temperature is brief, while its effect on food intake is prolonged. We are also confident in the accuracy of our data because a recent paper in the journal, with a focus on *pathomechanisms* instead of *circuit physiology* (Osterhout *et al.*, Nature 2022, 606(7916):937-944), showed that ependymocytes situated in the ventricular wall near the ventral medial preoptic area could contribute to thermodefensive behaviors. Tanycyte activation was observed as an auxiliary event upon fever induction, even though the authors of that study did neither discuss this cellular locus nor made an attempt to place it in a *physiological* circuit context.

To address the question of exposure to other temperatures, we have also conducted experiments in *cold* (+4 °C for 1h; reminiscent to a potential antagonism between summer vs. winter, as you have mentioned). **However, tanycyte activation was not observed upon cold exposure. Data were shown in: Fig. ED2 for both sexes.**

Q5: 'The title of the manuscript has to include the species exclusively used. It should say:"...in mice".'

Thank you for proposing this, **we have now implemented this change.**

Q6: 'The authors are repeatedly referring to food seeking. Has that actually been studied here? Or just time spent feeding? That is not the same behavioral endpoint. Is the impact of heat exposure on food intake dependent on the type of food?'

Given the limited number of metabolic variables we have initially tested, and consequently the low number of metabolic endpoints, we were most cautious to not over-interpret our data. This is why we opted for 'food seeking' as terminology. **In view of our additional data, we suggest that it is indeed food intake that has changed (not only the frequency of feeding but also the amount of food consumed and, consequently, caloric intake).** We have revised the terminology throughout the manuscript and report food intake as the actual change in the amount of food consumed before and 24 h after the acute heat challenge (expressed as Δg). Considering that the time spent eating and the amount consumed correlate, we concluded that acute heat indeed suppressed food consumption. **Data were shown in: Fig. 1b, 4h, ED1b, 9i-k.**

Q7: 'Why has the experimental setting been limited to just one temperature? Shouldn't there several temperatures be studied? Why just 40 degrees Celsius? Does cold exposure do the opposite? Or does that have comparable effects?'

Thank you for raising this important point. In our revised manuscript, we included cFos expression in tanycytes in mice exposed to 4 °C. This was comparable to controls kept at 25 °C (**Extended Data Fig. 1.d,e**). This finding justified our focus on acute heat.

Referee #2:

Thank you for your insightful and critical appraisal of our work. We were glad you deemed our study as having 'The idea is original as hypothalamus sits at the core of energy uptake neural circuits.' Please find our specific responses to your queries as follows.

Q1: 'The study has built a strong case, but has not proven that tanycytes actually synapse with neurons (and vice versa). This doesn't negate the findings but is a critical premise. I.e. Non-synaptic communications could also play a critical role and this has not been explored in depth.'

We appreciate that it is indeed critical to establish direct innervation and go beyond the anatomical premise on the existence of 'synaptoid contacts' between neurons and tanycytes (we refer to the manuscript for the specific studies cited). During our revisions, we used an optogenetic approach to show that PBN neurons form monosynaptic contacts on tanycytes. Therefore, we have injected AAV1-CAG-FLEXFRT-ChR2(H134R)-mCherry in the PBN of *Slc17a6-IREG2-FlpO-D* mice, and patch-clamped tanycytes *ex vivo*. Thus, we show that **excitation by light induces postsynaptic currents in tanycytes**. At the same time, our data precludes synaptic neurotransmission conversely from tanycytes onto neurons, because tanycytes do not display active membrane properties upon current injection. This supports our hypothesis that tanycytes modulate both neuronal firing and food intake through the paracrine secretion of VEGFA. To justify this arrangement, we used an siRNA approach for *Vegfa* loss-of-function in tanycytes, which inhibited the heat-induced reduction in food intake. Overall, **these data show unidirectional PBN-to-tanycyte synaptic signaling. Data were reported in: Fig. 3e-g₁, 4h, ED9i-l.**

Q2: 'In the context of environmental heat stress, the elderly are the most vulnerable to heat stress-induced anorexia and an ageing component is missing here. Similarly, previously-described pathways linking transient anorexia to illness (explored by David Anderson's group), also involving PBN, is not discussed.'

We would wish to emphasize that we did not study a pathomechanism that could be exacerbated by aging, which is in itself an interesting, yet distinct question. Instead, we have explored the **neurocircuit design of a physiological metabolic response**. Therefore, we **expanded the discussion of our paper, as requested**, and included some seminal papers from the Anderson laboratory (e.g., Cai *et al.*, *Nat Neurosci* 2014; 17(9):1240-8). These studies are indeed helpful to **contextualize behavioral aspects** of food intake.

Q3: 'Title and throughout text: As a subset of tanycytes are implicated in the described mechanisms, the title and text should consistently stress that as a heterogenous population, only a subset (approximately 50% according to figure 3 data) are implicated.'

We appreciate this level of attention to detail. We explicitly state the involvement of **α -tanycytes**.

Q4: 'Figure 1 (and related data). It was surprising that conclusions were based on n= 3-4 mice per treatment. Typically, these sorts of behaviour tests involve at least n=6 per group and in such a critical paradigm both sexes would be analysed. Not indicated whether the mice were litter mates? Where they weighed before experiments to show equivalence etc. Can general reduced 'activity' be ruled out downstream of heat stress?'

We agree that some of the experiments benefitted from increased cohort sizes. Therefore, we have repeated some of the experiments (e.g., in original Fig. 1), and **increased the number of observations to $n = 8$ /group. Moreover, we have reproduced the phenotype in both sexes.** Mice were not littermates to increase biological robustness (and exclude any bias due to 'litter effects/bias', as we have performed in all our published studies during the past two decades (e.g., Tortoriello *et al.*, EMBOJ (2014); Romanov *et al.*, EMBOJ (2015); Romanov *et al.*, Nat. Neurosci (2017); Cinquina *et al.*, Mol Psychiatry (2020); Romanov *et al.*, Nature (2020); Korchynska *et al.*, Nat. Commun. (2022)). Please also note that some of our genetic experiments simply excluded the use of littermates given the 1-in-4 or 1-in-8 breeding designs. **Yet, we think the inherent biological variability is a clear strength of our analyses.**

Q5: '1C & 1D – Vimentin expression/label intensity also appears to be elevated under heat stress (40 C compared to 25 C). Is this a real effect or artefact?'

We would identify this change in signal intensity as an artefact. Therefore, **at no point has this been analyzed in our study** (data acquisition was not set for intensity recording or transformation). Instead, we analyzed *cFos* expression as a *bona fide* marker of cellular activation throughout.

Q6: 'Figure 2 (and related data): 2C - What was the time course between ventricular viral injection and end point analysis. Could the mcherry+ve neurons be derived from tanycytes, and not necessarily connected to them as the figure concludes? There is no information on this in methods.'

Thank you for this interesting interpretation. Since we use AAVs, there is a 3-week lag-time between virus injection and histochemical analysis. The primary site of viral transduction is mCherry positive. Once the WGA-Cre AAV 'jumps', synaptically connected neurons will be green. **In case the neurons were tanycyte-derived progeny then they should express mCherry and not GFP** (see, e.g., Fig. 2d as illustration). **We have not seen any such neuron.** Moreover, and even if there is a low probability of proliferation events in the postnatal hypothalamus, the likelihood that a tanycyte-derived neuron would migrate from the hypothalamus to the PBN is infinitesimally low (there is no such data in the literature as far as we know). Likewise, **we exclusively see GFP-tagged neurons in the PBN. Thus, we suggest that the neurons are presynaptic to tanycytes.**

Q7: '2c, d, j. The figure and experimental approach is already complicated and so false colouring should be uniform to save further confusion. Perhaps Mcherry consistently in red, Vimentin in green (in J)

Thank you for highlighting this point. **We acquired the images using specifically this color coding**, which is an option available with Zeiss LSM confocal microscopes to avoid difficulties for color-blind colleagues. Therefore, and respectfully, we would prefer keeping the original color code, which we consistently used throughout the figures, and is precisely described in their legends.

Q8: 'Extended data 3 - Please show examples of beta2 tanycytes association with Vglut2 terminals.'

We have done as requested. **Data are shown in: Fig. ED6.**

Q9: '- GluA1 is not restricted to beta tanycytes and tapers off to include some alpha 2 tanycyte domain as well.'

This statement is correct ,yet does not impact our observations/analysis in any specific way. We **revised the relevant sentence** to avoid the view that tanycyte subtypes are inherently different (alike a binary 'yes/no' code) at the molecular level.

Q10: '- The quantifications could also include proximal versus distal tanycyte processes as the major glut terminal in Figure 1 was noted as close to soma.'

Indeed, quantitative anatomical studies exclusively focusing on tanycytes have applied such anatomical distinction. However, we see an inherent contradiction between Q9 vs. Q10: we have at no point in the manuscript alluded to that PBN-specific inputs would be on either proximal or distal stretches of tanycyte processes. Therefore, we think this distinction would be arbitrary and without significance for the circuit design we have described. For this reason, **we would suggest not to include an subjective morphological variable** whose physiological significance remains (at best) ambiguous.

Q11: 'Figure 4d: Is there any evidence that tanycytes have a 1:1 or 1:many relationship with neurons? No evidence has been produced to this effect. It should be clarified or noted, as stoichiometry is likely to be a critical factor in eliciting a signal in any signal transduction pathway.'

This is a very complex question, which we noted in the manuscript: 1) tanycytes are connected by gap junctions and function as 'small-world clusters' with, e.g., Ca^{2+} signaling being synchronized across close-sitting cells. 2) ~50% of tanycytes receive VGLUT2⁺ synapses. Thus, we suggest that it **is the size of the tanycyte cluster vs. the cumulative number of synapses along their parenchymal processes that produces a threshold for population excitation**.

Q12: 'Extended data 5: Statement in line 237/238 is incorrect. A glance Ex/data Figure 5 show that the three cited factors are in fact also expressed at similar levels by some neuronal populations, as well as tanycytes. Text should also mention that in addition to vegf, Tgfb3 and Tgb2, Fgf1 and Pdfga are clearly the other potential strong candidates for signal transduction (at least according the heat map).'

Thank you for noting this. **We revised the description of the single-cell RNA-seq data accordingly**.

Q13: 'Methods: Details of In situ hybridization studies missing.'

We prduced an extended description of the protocol we followed (Molecular Instruments: <https://files.molecularinstruments.com/MI-Protocol-RNAFISH-FrozenTissue-Rev2.pdf>)

Referee #3:

Thank you for your appreciative words on our manuscript (*'The interaction of neurons and non-neuronal cells is a topic of major interest. ... Furthermore, while recent papers demonstrated the neural mechanisms underlying cold-induced food seeking behavior (Yang et al., 2021; Deem et al., 2020), the neural basis underlying heat-induced feeding suppression remains unknown; thus, this discovery—if properly supported—is timely and fills a knowledge gap.'*) Herewith, we have addressed your queries in detail.

Q1: 'According to the authors, the proposed circuit, which includes PBN neurons, tanycytes, and arcuate nucleus neurons, serves as a neural substrate that converts heat sensation into appetite suppression. However, it is unclear whether the proposed circuit is required for heat-induced food intake reduction. Are tanycytes necessary for this phenomenon? Demonstrating the necessity of this circuit is necessary for their claims. This can be accomplished by measuring food intake after a thermal challenge during acute inhibition and/or chronic tanycyte ablation.'

We appreciate these concerns. During our experimental work, we have measured food intake. We have also done so when knocking down *Vegfa* expression in tanycytes *in vivo* ($n = 8$ group/treatment in repeated measures). We found that *Vegfa*-RNAi reduced the acute heat-induced suppression of food intake. **These data support and corroborate the involvement of tanycytes in modulating food intake following either acute heat or the direct activation of PBN neurons by chemogenetics. Data were shown in: Fig. 1, Fig. 4g,h, ED1b and ED9i-l.**

Q2: 'The authors also suggested that thermosensory signals are relayed to tanycytes by parabrachial glutamatergic neurons (e.g., "thermosensing glutamatergic neurons of the PBN innervate tanycytes"). They did show that thermal challenge increases Fos and pErk1/2 levels in tanycytes, but it is unclear whether parabrachial neurons are required for tanycytic activation after heat exposure. Indeed, the authors did not demonstrate that tanycytes in the PBN are innervated by "thermosensing" glutamatergic neurons. Given that primary tanycytes express thermosensitive TRPV channels, it appears necessary to show that the parabrachial nucleus is essential for tanycytic activation in response to thermal challenge.'

Thank you for identifying these critical points. 1) We support our findings by CRACM: AAV1-CAG-FLEXFRT-ChR2(H134R)-mCherry was stereotaxically delivered into the PBN of *Slc17a6-IRES2-FlpO-D* mice, allowing for the Flp-dependent expression of ChR2(H134R)-mCherry specifically in VGLUT2⁺ neurons of the PBN. Subsequently, 470 nm light could evoke optogenetically induced EPSCs (oi-EPCs) in tanycytes apposing ChR2-tagged afferents (patch-clamp experiments *ex vivo*). 2) We showed an increase in cFos⁺-mCherry neurons following the chemogenetic activation of VGLUT2⁺ neurons in the PBN of *Slc17a6-IRES2-FlpO-D* mice injected with AAV-FRT-hM3D(Gq)-mCherry viruses, which coincided with many more tanycytes expressing cFos. 3) We used high-resolution microscopy to show that glutamatergic efferents originating from the PBN are juxtaposed with cFos⁺ tanycytes. **These data were included in: Fig. 3n,f-g₁, ED4,5.**

Q3: 'The authors argue that monosynaptic PBN input to tanycytes mediates the heat-induced orexigenic effects and mention the possibility of second-order relay by intrahypothalamic neurons to tanycyte in the discussion section. However, evidence suggesting that tanycytes are directly modulated by PBN projection is insufficient to underline the importance of the monosynaptic PBN-tanycyte connections. It would be better to directly demonstrate modulation of

tanycyte by direct PBN projection, such as Chr2-assisted circuit mapping (CRACM) experiment (that expresses Chr2 virus in the PBN and records from tanycyte while activating the PBN terminal) or the author should tone down their arguments.'

We provided data supporting our hypothesis on monosynaptic PBN-to-tanycyte connections by CRACM. **Data were included in: Fig. 3e-g₁.**

Q4: 'In Fig. 3, the authors illustrate the differences between α -tanycytes and β -tanycytes in a variety of ways (e.g. VGLUT2+ input density, AMPAR subunit expression, and EPSC features). However, there is no discussion about the significance of tanycyte type distinctions in reduced food seeking by heat perception. Also, the authors should explain why they focused on α -tanycytes (despite the fact that EPSC intensity is higher in β -tanycytes) and how their data that tanycytes in the caudal subdivision preferentially accumulate pErk1/2 after 40°C exposure can be reconciled (Fig. 1d).'

We are grateful for these great questions. We noted that acute heat exposure activates a small cluster of tanycytes, which are predominantly α -tanycytes. However, tanycytes can amplify this signal through gap-junctions, which was shown to connect 1-to-60 cells (e.g., Recabal A *et al.*, Connexin-43 Gap Junctions Are Responsible for the Hypothalamic Tanycyte-Coupled Network. *Front Cell Neurosci.* (2018) 12:406). We could reconcile the broader and increased expression of pErk1/2 in other tanycyte domains as the result of such amplification, whereby neighboring tanycytes could be efficiently recruited following acute heat exposure. **We have now discussed this topic, as requested.**

Q5: 'The authors attempt to emphasize the importance of this study in the Abstract and Introduction sections by mentioning climate change and global warming, which represent an increase in environmental temperature over longer timescales. This is misleading, especially given that mice are only exposed to hot environments for a few hours in their experiments. They should revise the Abstract and Introduction sections in order to properly summarize and introduce the research (i.e. the function of tanycytes during acute heat exposure).'

Indeed, and also considering the sentiments of Referees #1 & #2, **we revised the Abstract and Introduction** and deleted any unwanted reference to climate change. Instead, we contextualized our model as heat exposure, e.g., upon leaving temperature-controlled environments or sauna.

Q6: 'In Extended Data Fig. 1a, the authors found no significant change in skin temperature after a 3-h exposure to 40°C. However, this observation is inconsistent with ref 16. As shown in Fig. 1a of ref 16, the skin temperature of mice increases as the ambient temperature rises. Please explain this discrepancy.'

Unfortunately, our initial temperature recordings were suboptimal. Even if we used infrared guns to measure surface temperature, the fur of the animals was not shaven off. Therefore, we repeated this experiment, **removed the fur, and measured body temperature over brown adipose fat depots (in the neck/back), as well as at the tail area.** We could show that the body temperature transiently increased in both regions, in agreement with Ref. 16. **Data were shown in: Fig. 1c and ED1c.**

Q7: 'The authors use the phrase "food seeking" throughout the manuscript. This refers to the appetitive phase of feeding rather than the consummatory phase. What was measured,

however, was the amount of time spent eating, which is indicative of the consummatory phase of feeding behavior. Please change the phrase "food seeking" to something more appropriate, such as "food intake" or "food consumption."

Given the additional parameters measured during the revisions phase, we now phrase **'food intake'** throughout, and expressed it as the difference in food weight (Δg) consumed before and in a 24-h interval after acute thermal challenge.

Q8: 'PBN neurons that project to the arcuate nucleus (Fig. 2e) and tanycytes (Fig. 2f) appear to have distinct anatomical locations. It would be useful to have images that could identify the LPB subregion.'

We have collected new images showing the PBN subregions with the sub-cerebellar peduncle indicated and the PBN projections terminating around the hypothalamic 3rd ventricle. **Data were shown in: Fig. 2e,f, ED4a-c.**

Q9: 'Please add quantitative information to the data that is currently only presented as images (e.g. Fig. 2h, i).'

We have added **these data: Fig. ED2e,h.**

Q10: 'Few casual words such as "liquor". in Line 40, may be better rephrased.'

We have changed this archaic wording to 'cerebrospinal fluid' throughout.

Q11: 'Lines 164 and 811. Fig. 2l should be renamed Fig. 2j.'

This error was corrected.

Q12: 'Lines 242 and 243. Fig. 5a,b and 5c should be renamed Fig. 4a,b and 4c.'

This error was corrected.

Q13: 'Line 154. "Effernets" should be spelled "efferents".'

This error was corrected.

Q14: 'Line 272. "VGEFA" should be spelled "VEGFA".'

This error was corrected.

Referee #4:

Thank you for judging our study as ‘Conceptually, the message of the study is very interesting and provides a new perspective on the functions of tanycytes as integrators of environmental signals (i.e. nutrients and now thermal responses).’ We also appreciated your insightful queries, for which you find our point-by-point responses as follows.

Q1: ‘- Fig.1A. Total food intake, rather than number of bouts should be shown.’

We measured food intake and expressed it as the difference in food weight consumed in grams (g) before and after acute heat exposure in both males and females. **Data were included in: Fig. 1b (and in all genetic studies).**

Q2: ‘- Fig 1C. It would be nice to see a picture taken at reduced magnification to appreciate the entire (or larger portion) wall of the 3rd ventricle.’

We show both survey images (at low power), as well as at high resolution, in both male and female mice. **Data were included in: Fig. ED2.**

Q3: ‘- Ext Fig 1B. These images actually do not support the text. In fact, it seems that 40°C decreases FOS in the 3V wall of the ARC and also in the ARC region.’

We have captured another set of images *in both male and female subjects* to show that the density of cFos⁺ tanycytes is significantly increased at the 3rd ventricle of the ARC region (-1.94 mm). **Data were included in: Fig. ED2.**

Q4: ‘- Ext Fig 1D. There is also increased FOS in both alpha and beta tanycytes (text only states alpha tanycytes).’

We indeed find other tanycyte populations positive for cFos, too. However, we revealed that α -tanycytes are more prone to express cFos following acute heat exposure in both sexes. **Data are referred to: Fig. ED2.**

Q5: ‘- Fig 2. In general, images here should be improved by adding higher magnification insets (C-D), showing homologous areas (E-F), spanning larger areas (H-I) or providing additional images (J).’

We provided high-resolution insets. Additionally, we show panoramic views related to Fig. 2. **Data can be found in: Fig. 2c,d, ED2,3.**

Q6: ‘- The authors do not provide evidence that the VGLUT2 synapses are coming from PBN neurons. They should perform experiments injecting Cre-dependent synaptophysin AAVs into the PBN and count the number of synapses in tanycytes.’

Thank you for raising this important point, which resonates on comments from Referee #1 (Q1), but also from Referees #2 & #3. We have injected an AAV1-Syn-FRT-hM3D(Gq)-mCherry construct in the PBN of *Slc17a6-IRES2-FlpO-D* mice to drive the specific recombination and expression of DREADD(Gq) in *Vglut2/Slc17a6*-mCherry transduced neurons.

Reminiscent to data we obtained with an AAV1-EF1a-FRT-hM3D(Gq)-mCherry construct, **chemogenetic induction of PBN neurons elicited cFos expression in tanycytes**. Instead of a static anatomical assessment (synapse counts), we functionally characterized PBN-to-tanycyte projections using CRACM, in which we illuminated ChR2-mCherry terminals in the vicinity of tanycytes in brain slices from *Slc17a6-IRES2-FlpO-D* mice injected with an AAV1-CAG-FLEXFRT-ChR2(H134R)-mCherry construct in the PBN. **We report that light-evoked stimuli elicit optogenetically induced EPSCs (oi-EPSCs) in patch-clamped tanycytes, suggesting that PBN glutamatergic neurons indeed form monosynaptic connections with tanycytes. Data were shown in: Fig. 2e-j₁, 3f-g₁.**

Q7: '- Lines 147-149 overstate their findings (as VGLUT2 nerve endings could be projected from many other regions than the PBN).'

We have revised our conclusions. Besides discussing at least 4 different experimental reasons (WGA-Cre retrograde labeling, CRACM, electrophysiology, *Vegfa* knock-down), we have toned down our conclusion that the PBN is the primary source of glutamatergic inputs to tanycytes.

Q8: '- Line 152- 153 is not demonstrated by the results.'

Our additional experiments justify the use of *Slc17a6-IRES2-FlpO-D* mice to study glutamatergic projections from the PBN towards the 3rd ventricle. We highlight that **both chemogenetic and optogenetic activation of glutamatergic efferents of the PBN were sufficient to activate cFos, and evoke EPSCs in tanycytes** in *Slc17a6-IRES2-FlpO-D* mice that were injected for FLP-dependent AAVs in the PBN. **Data were shown in: Fig. 3f-g₁.**

Q9: '- Ext FIG2B is wrongly placed.'

This mistake has been corrected.

Q10: '- Ext FIG2D. PDYN staining is not exclusive of PBN neurons and could actually come from other neurons.'

We thank you for raising this point. **We have deleted any reference to prodynorphin⁺ neurons.** We only mention that glutamatergic projections from the PBN to tanycytes link somatosensation to the modulation of an endocrine output. Testing the specific PBN subpopulation would require a screen of 21 Cre lines (Pauli *et al.*, 2022), which is certainly beyond the scope of this study. However, recent findings from Palmiter and colleagues show that *Adcyap1*, *Tac1*, *ChaT* and *Pdyn* expressing glutamatergic neurons, amongst others, could project from the PBN towards the hypothalamic 3rd ventricle, where α -tanycytes reside (Pauli *et al.*, 2022, doi: 10.5281/zenodo.6707404), supporting our findings.

Q11: '- FIG 3D. Images should be at the same magnification. It does not seem that GluA2 preferably stains alpha tanycytes.'

We have captured another set of confocal images using the same number of tiles. Even though the pattern of GluA2 expression supports our initial description of this subunit being preferentially expressed in α -tanycytes, we have carefully rephrased the relevant results section. Moreover, and in accord with Referee #2's Q9, we have revised the statement on GluA1

being specifically expressed in β -tanycytes, and state that its expression is well detectable in $\alpha 2$ -tanycytes, too.

Q12: '- It is difficult to understand why the authors included the electrophysiology data shown in FIG 3. These results do not demonstrate that tanycytes are excited via glutamatergic PBN neurons. A more direct way to show that tanycytes respond to glutamate would be to perfuse actual glutamate in the electrophysiology setting (rather than receptor agonists). Statement in lines 202-203 is false.'

The reason we included the electrophysiological data in original Fig. 3 was because we initially thought to provide functional evidence on **tanycytes direct responding to neuronal activity**. In this section, we have not stated that the recorded EPSCs and synaptically evoked Ca^{2+} transients were generated by PBN neurons. We merely aimed to show that 'synaptoid contacts' on tanycytes could be functional. Nevertheless, **we have brought this concept forward during the revision process by using CRACM to show that monosynaptic inputs from PBN neurons to tanycytes indeed exist and generate optically evoked EPSCs**. Lastly, **we see problems with using glutamate**: 1) it is entirely non-specific, and besides being a neurotransmitter, it is also a metabolite. Thus, glutamate superfusion is unlikely to mimic synaptic glutamatergic neurotransmission. 2) Glutamate superfusion will activate all types of glia (excitatory amino acid transporters). Therefore, its effects on synaptic neurotransmission could be confounded by perisynaptic astroglial activity. *Data were reported in: Fig. 3e-g1.*

Q13: '- Ext FIG 3A-C. Again, authors do not demonstrate that VGLUT2 synapses are coming from PBN neurons.'

Please see our responses to Q6-Q8 above. We have performed the requested experiment by recombining AAVs specifically in the PBN of *Slc17a6-IRES2-FlpO-D* mice.

Q14: '- Line 237 (Ext FIG 5A) is not true. According to the heat map, VegfA is also expressed in other cell types at similar levels as tanycytes.'

As requested by Referee #3, this **statement/conclusion was revised**.

Q15: '- Lines 242-245 the authors referred to FIG 5 when they meant FIG 4.'

This unfortunate error is now **corrected**.

Q16: '- Line 253. KCl depolarizes all cell types, not only tanycytes. This approach is too drastic and inespecific, and therefore VEGF in CSF may not necessarily come from tanycytes.'

Thank you for highlighting this potential problem. **We have repeated this experiment by using the physiological phenomenon that is the core of our report**: We exposed rats to 40 °C for 1h, sampled their cerebrospinal fluid, and compared its VEGF content to rats kept at 25 °C. **These data are shown in: Fig. ED9c.**

Q17: '- Images Ext Fig 5 D-E are misleading. Larger fields with more POMC neurons should be shown (in E) and provide separate images for dyes.'

We have **produced image surveys** along the arcuate nucleus, including in tissues with POMC neurons. **Data were shown in: Fig. 4d, ED9f-h.**

Q18: '- The authors should take advantage of publicly available sequencing data, as done in other sections of the manuscript, to provide evidence of the expression of *Flt1* in diverse subsets of neurons. ISH data for AgRP should be shown as it seems it is an important mediator of the appetite effects of tanycytes.'

Thank you for making this important point. We preferred to perform fluorescence *in situ* hybridization, which showed that ***Flt1* is indeed co-expressed in AgRP and Th neurons**, but not in *Pomc* neurons. **Data were included in: Fig. 4d, ED9f-h.**

Q19: '- Lines 282-283 is an overinterpretation of the data.'

The statement was **toned down and corrected.**

Q20: '- FIG 4G. Authors should also provide ultrastructural data showing TH cells in close apposition to tanycytes.'

We have performed this experiment and provided ultrastructural data on TH⁺ neurons being in close apposition to tanycyte processes. **Data were presented in: Fig. 4c.**

Q21: '- In the model suggested by the authors, it is not clear how VEGFA released by tanycytes leads to reduced food intake. Is VEGFA inhibiting TH and AgRP neurons thus enhancing the anorexigenic output? The effects of VEGFA on the activity of TH and AGRP neurons should be assessed. Chemogenetic/optogenetic studies could be also performed in TH and/or AgRP neurons to demonstrate that they mediate the feeding effects of this circuit.'

We embarked on a more direct approach by knocking down *Vegfa* in tanycytes, and measuring food intake following acute heat exposure. To do so, we infused siRNAs into the 3rd ventricle of *C57Bl6/J* mice. We found that reducing *Vegfa* expression in tanycytes prevented a reduction in food intake after acute heat exposure. **Data were shown in: Fig. 4f-h.**

Q22: '- To prove that is the VEGFA derived from tanycytes the link between thermo-sensitive PBN neurons and feeding, the authors should repeat the experiment shown in FIG 5 using mice lacking VEGFA in tanycytes.'

Please see our response to your preceding query (Q21).

Q23: '- Do all thermo-sensitive PBN neurons send projections to tanycytes? What is the molecular identity of glutamate PBN neurons that interact with tanycytes?'

Please see our response to Q10 for details.

Reviewer Reports on the First Revision:

Referees' comments:

Referee #1 (Remarks to the Author):

All questions have been answered satisfactorily. The manuscript improved further. This reviewer has no additional concerns.

Referee #2 (Remarks to the Author):

Benevento et al 2022-08-23268B

This study aims to delineate the cellular and molecular pathways linking PBN to an observed transient phase of anorexia after heat stress. The study suggests that PBN innervates a subset of tanycytes, and upon acute heat stresses, induces VEGF expression and release onto hypothalamic neurons to induce reduced food intake.

The revised manuscript has addressed most of the concerns and included some fresh data to strengthen the arguments posed in the main text.

The revision has created some new points and formatting issues, which need addressing.

1. The revised text (particularly blue highlighted areas) requires some further editing to replace verbose expressions, improve grammar and punctuation and ensure text is consistently in the past tense.

- Line 91: 'attenuation' not loss, as this was not a Knockout of Vegf.

- Line 370: add ' of mice' after nervous system.

2. Line 251 to 258 (described lines 681-694) – Experiments to show lack of VEGF secretion into CSF is not very robust. Ideally, a control protein would have been measured alongside the assays presented.

Figures:

1C – add 'minutes' to labelling axis

4G – capitals VEGF on Y axis, as this is a protein measurement.

Extended data Figure

8G – Legends says Tanycytes, but the transcriptomic data label says Ependymocytes. The latter seems to be the correct extraction from Campbell et al 2017 (ref 44). Which is the intended representation?

If Tanycytes, then Pdgfra, Fgf1 and Tgfb3 are the other potential candidates for mediating some of effects described here, as they have the potential to induce ERK and possibly Fos. Therefore, the discussion should say something like: “we have demonstrated a role for VEGF here, but other potential inducing candidates could include....”.

Fgf10 also seems to crop up in extended Figure 8, but not shown in previous version of this figure. Is this a typo or confusion with Fgf1?

Referee #3 (Remarks to the Author):

The authors have made extensive efforts to address the Referees' comments. For example, they have conducted electrophysiology experiments to test the monosynaptic connectivity between PBN neurons and tanycytes, as well as Vegfa knockdown experiments to elucidate the mechanism by which tanycytes signal downstream neurons in the arcuate nucleus to reduce food intake after warm exposure. However, some comments from the previous review (Q1-3) were not adequately addressed, and the authors are encouraged to conduct several additional experiments to fully support their claim, as follows:

1. Although the authors have presented data (Fig. 4f-h) showing that stereotaxically delivering Vegfa siRNAs into the 3rd ventricle reduces Vegfa mRNA in tanycytes and attenuates the heat-induced reduction in food intake, these data only weakly support the authors' claims. Specifically, intraventricular delivery of Vegfa siRNAs may produce off-target effects, rather than acting solely on the tanycytes. Moreover, the effect size of food intake suppression appears to be small, which may not be sufficient to convince readers of the necessity of tanycytes in heat-induced reduction in food intake. To address these concerns, the authors may consider adding another set of loss-of-function data to their study. In particular, investigating whether inactivation of tanycytes using TeLC (as shown in Fig. 5) can attenuate heat-induced feeding suppression would be promising. Such data would provide stronger evidence for the authors' claims.

2-1. The authors have demonstrated anatomical and functional connectivity between PBN glutamatergic neurons and tanycytes using electron microscopy imaging, chemogenetics experiments combined with cFos staining, and CRACM experiments. However, these data still do not sufficiently support the authors' claim that “thermosensing glutamatergic neurons of the parabrachial nucleus innervate tanycytes (lines 37-38)”. Specifically, it remains unclear whether PBN neurons projecting to the tanycytes are thermosensitive. To address this question, the authors may consider retrogradely labelling PBN neurons that project tanycytes and subsequently performing a Fos experiment after exposing mice to warmth. Such experiments could provide crucial insight into the properties of PBN neurons projecting to the tanycytes and further support the authors' claim. Furthermore, the phrase "primary thermoceptive neurons in the pontine parabrachial nucleus" in the Abstract (line 33) may be misreading. Do the authors intend to imply that PBN neurons express thermoreceptors and detect temperature directly?

2-2. The authors showed an increase in body temperature during heat exposure (Fig. 1c). Given that tanycytes express thermosensitive TRPV1 channels, these cells may detect increased body temperature directly, without relying on thermosensory PBN relay neurons. Therefore, the authors should clarify if the PBN is an essential input for the tanycyte-mediated reduction of food intake in response to environmental warmth. Several experiments may be considered, as below:

- If the PBN is an essential input for tanycyte to reduce food intake upon environmental warmth, does inactivating the PBN rescue the anorexic effect during warm exposure?
- The authors may consider testing whether the activity of tanycytes is decreased (by Fos experiments, for example) or the level of VEGFA is lowered when the PBN is inactivated during warm exposure. Such experiments could provide important support for the authors' claim that tanycytes receive input from thermosensory PBN neurons.

3. The newly added Fig. 3f-g shows that optogenetic activation of PBN glutamatergic neurons elicits EPSCs in tanycytes. However, the average latency of 250 ms is quite different from the average spike latency of neurons (~10 ms), and the latency of EPSCs is quite variable. The authors should provide explanation for these slow and variable latencies and include this in the manuscript. Moreover, it would be informative to include additional data such as the failure rate of optogenetically-induced EPSCs.

Referee #4 (Remarks to the Author):

The authors have made a substantial effort in addressing the reviewers' comments and as a result the revised version of the manuscript has significantly improved. Nevertheless, some of the initial questions of this reviewer have not been adequately addressed and the new data raised further questions that need to be considered.

1. The experimental paradigm used consist in submitting the animals at 40C for 1h. According to the results of the authors, acute heat would rapidly activate glutamatergic PBN neurons that in turn would activate tanycytes thus reducing food intake. The authors show food intake data over 24h post heat exposure, but it would be nice to see a time-course of food intake. Given the rapid circuit activation as a consequence of high temperature, it is plausible to expect that food intake is affected at short time-points.

2. Line 206: I guess you mean Fig 3d.

3. I am not sure why the authors changed the original Extended Fig 8. In the original one they showed the expression levels of diverse markers in tanycytes while now they are included in the category ependymocytes (which is more general).

4. Lines 274-283 (Fig 4e). In this paragraph the authors suggest that tanycytes per se are able to sense acute heat, as they express TRPV2 channels and are able to release VEGFa and reduce the excitability of neighbor neurons. Unless this reviewer is missing something, this experimental setting bypasses the PBN glutamatergic input on tanycytes. Then, these results seem to suggest that actually the PBN glutamatergic input on tanycytes is not required to mediate the effects of heat.

5. In the mass spectrometry study, did the authors detect VEGFa? This would be a better way to demonstrate it is expressed in tanycytes.

6. The inhibition of VEGFa is a good strategy to demonstrate that is the molecule mediating the acute effects of heat on food intake. Problematically, the experimental approach is not specific for

tanycytes. Delivering an siRNA into the ventricle could actually target multiple cell types, including some expressing high levels of VEGFa (e.g., endothelial cells). A specific approach is necessary to conclude that VEGFa derived from tanycytes mediates the acute effects of heat on food intake.

7. The validation of the model above, should be done by a method measuring protein levels of VEGFa (rather than RNA – that is RNAscope and qPCR).

8. The authors use a wide range of tools, and this is a strength of the work. However, sometimes the authors do not assess whether these tools perform as they are supposed. For example, in the TeLC study the authors do not show that TeLC is actually expressed in tanycytes after viral infection. They did not show either that they are able to simultaneously activate PBN neurons and inactivate tanycytes (via c-fos). These kind of controls should be provided for all the experimental strategies.

9. Fig 5e, f; Ext Fig 10. Why the authors show “time spent eating/drinking” instead of total amount of food eaten or water drunk?

10. In the model that the authors propose, heat would activate PBN glutamatergic neurons that in turn would activate tanycytes and subsequently inhibit orexigenic (Th and/or AgRP neurons) in the ARC. This would mediate the observed reduction in food intake upon heat exposure. However, this latter part of the model (the role of ARC orexigenic neurons) has been neglected, but it is key to “close the circle” to explain how food intake is attenuated. The authors should address this point.

11. Some images and quantifications are difficult to interpret. For example, Fig 2j/j1. What is within the inset (square)? Is this a tanycyte? Does it stain for vimentin? Similarly, in Fig 4a-b the authors quantify VEGFa puncta in a-tanycytes. How do they know they are alpha? Because of their location? Is the number of puncta referred as to the average in a single cell? Or in the whole area shown in the image? The same for Fig 4g.

12. Extended Fig 1C: x-axis lacks units.

13. Line 336. C-fos expression is not a feature of only a-tanycytes. In fact, in the response letter, the authors acknowledge that other tanycyte populations are activated although the alpha type represents the majority.

14. Line 361. This is an overstatement. Authors did not show specificity of release from tanycytes.

15. Line 364. Truncated sentence.

16. Considering the findings of the authors, I would suggest changing the title. First, the authors do not show an endocrine mechanism (perhaps paracrine would be more accurate). Also, I think that narrowing the findings to only alpha-tanycytes is a step too far, as other types of tanycytes may be involved. Insights on the circuit engaged would be also informative.

Author Rebuttals to First Revision:

Point-by-point responses to the Referees' concerns

(manuscript ID: Nature-2022-08-13268B)

Referee #1:

'All questions have been answered satisfactorily. The manuscript improved further. This r no additional concerns.'

Thank you for approving our manuscript for publication.

Referee #2:

Thank you for appreciating our thorough revision, and the inclusion of fresh data to strengthen our conclusions. We very much appreciated your attention to detail, as well as constructive comments. Please find our point-by-point responses to your queries as follows.

Q1: 'The revised text (particularly blue highlighted areas) requires some further editing to replace verbose expressions, improve grammar and punctuation and ensure text is consistently in the past tense.'

We have addressed the issues you have mentioned throughout the manuscript. Care was directed towards using past tense when describing our results. 'Verbose' expressions were **reduced and made more coherent** with the rest of the manuscript.

Q2: 'Line 91: 'attenuation' not loss, as this was not a Knockout of Vegf. and Line 370: add ' of mice' after nervous system.'

These changes **have been made**.

Q3: 'Line 251 to 258 (described lines 681-694) – Experiments to show lack of VEGF secretion into CSF is not very robust. Ideally, a control protein would have been measured alongside the assays presented.'

Thank you for asking this. Please note that the volume of cerebrospinal fluid that we could isolate from rats (for this experiment mice were too small) was only sufficient for **a single ELISA run for VEGFA**. As such, we have performed these extremely challenging experiments twice to satisfy your previous concern on the number of observations. We do not believe that another protein, whether changing or not, would have made our study better, or more stringent. This is because the release of any other protein could be viewed as independent from VEGFA. Instead, we have performed

histochemistry with an anti-VEGFA antibody for heat induction and RNAi experiments (**Extended Data Fig. 9b,c; Extended Data Fig. 10a**) to strengthen *i*) the localization of VEGFA to tanycytes and *ii*) direct evidence on **successful attenuation of *Vegfa* expression** upon gene silencing.

Q4: 'Figures: 1C – add 'minutes' to labelling axis, 4G – capitals VEGF on Y axis, as this is a protein measurement.'

Changes to Fig. 1C **have been made**. For 4G, this was a panel of quantitative *in situ* hybridization, and therefore the axis label referred to mRNA levels.

Q5: 'Extended data Figure 8G – Legends says Tanycytes, but the transcriptomic data label says Ependymocytes. The latter seems to be the correct extraction from Campbell et al 2017 (ref 44). Which is the intended representation? If Tanycytes, then *Pdgfa*, *Fgf1* and *Tgfb3* are the other potential candidates for mediating some of effects described here, as they have the potential to induce ERK and possibly Fos. Therefore, the discussion should say something like: "we have demonstrated a role for VEGF here, but other potential inducing candidates could include....". *Fgf10* also seems to crop up in extended Figure 8, but not shown in previous version of this figure. Is this a typo or confusion with *Fgf1*?'

We appreciated your request of clarifications. As required after the second round of review, we have returned to the original Campbell *et al.* terminology. As originally presented in Campbell *et al.*, we visualized a heat map reporting the fold change expression of genes expressed in 'tanycytes' vs. all the cell types listed on the horizontal axis. We also subset tanycytes based on *Rax* expression and wrote (lines 241-242 of the previous version) that '**Herein, *Rax*⁺/*Col23a1*⁺ tanycytes expressed *Vegfa*, *Tgfb3*, *Tgfb2*, *Fgf10* and *Pdgfa*.**' Thus, our analysis was and remains correct.

We have **inserted an additional sentence** on alternative mediators in the discussion, as requested. It is *Fgf10* in the Campbell *et al.* paper (and also in **Extended Data Fig. 8a**), and as reported in the manuscript.

Referee #3:

We thank the Referee for recognizing the amount of experimental work that has been made to strengthen this manuscript. We also appreciated the remaining questions below and have addressed these both experimentally and through text edits in the re-revised manuscript. Herein, please find our point-by-point replies to your specific queries.

Q1: 'Although the authors have presented data (Fig. 4f-h) showing that stereotaxically delivering Vegfa siRNAs into the 3rd ventricle reduces Vegfa mRNA in tanycytes and attenuates the heat-induced reduction in food intake, these data only weakly support the authors' claims. Specifically, intraventricular delivery of Vegfa siRNAs may produce off-target effects, rather than acting solely on the tanycytes. Moreover, the effect size of food intake suppression appears to be small, which may not be sufficient to convince readers of the necessity of tanycytes in heat-induced reduction in food intake. To address these concerns, the authors may consider adding another set of loss-of-function data to their study. In particular, investigating whether inactivation of tanycytes using TeLC (as shown in Fig. 5) can attenuate heat-induced feeding suppression would be promising. Such data would provide stronger evidence for the authors' claims.'

We have performed the experiment using intracerebroventricular infusion of an AAV carrying either TeLC-GFP or a control virus expressing only GFP in Rax-CreER^{T2} mice, as requested. **These data are shown in Figure 4j,k and Extended Data Fig. 10f.** In brief, TeLC infusion occluded the heat-induced reduction in food intake. We would also point to **Figure 5e**, which shows that TeLC-GFP was selectively expressed in tanycytes, and did not have cellular off-targets deeper in the hypothalamic parenchyma. Cumulatively, these data strengthen our conclusions.

Q2: '2-1. The authors have demonstrated anatomical and functional connectivity between PBN glutamatergic neurons and tanycytes using electron microscopy imaging, chemogenetics experiments combined with cFos staining, and CRACM experiments. However, these data still do not sufficiently support the authors' claim that "thermosensing glutamatergic neurons of the parabrachial nucleus innervate tanycytes (lines 37-38)". Specifically, it remains unclear whether PBN neurons projecting to the tanycytes are thermosensitive. To address this question, the authors may consider retrogradely labelling PBN neurons that project tanycytes and subsequently performing a Fos experiment after exposing mice to warmth. Such experiments could provide crucial insight into the properties of PBN neurons projecting to the tanycytes and further support the authors' claim. Furthermore, the phrase "primary thermoceptive neurons in the pontine parabrachial nucleus" in the Abstract (line 33) may be misreading. Do the authors intend to imply that PBN neurons express thermoreceptors and detect temperature directly?'

Thank you for asking additional validation of the circuit in question. We have first performed **cFos histochemistry** to show that PBN neurons indeed become activated in parallel with tanycytes upon exposure to 40 °C (**Fig. 1d**). Even without performing retrograde tracing, the **Fos⁺ neurons are in the ventromedial domain of the PBN** that we have shown as the source of glutamatergic input to tanycytes earlier. To support causality, we have injected an **AAV-FlpON-TeLC-GFP construct into the PBN of Slc17a6-FlpO mice and exposed the animals to 40 °C for 1h**. Once expressing TeLC, alike for the experiment you have asked for tanycytes above, and noting that TeLC silences VAMP2-based quantal synaptic neurotransmission, **neither tanycytes became cFos⁺ nor did we record a change in food intake after exposure to 40 °C (Fig. 5a-c)**. These data mechanistically reinforce a link between the PBN and tanycytes.

Q3: '2-2. The authors showed an increase in body temperature during heat exposure (Fig. 1c). Given that tanycytes express thermosensitive TRPV1 channels, these cells may detect increased body temperature directly, without relying on thermosensory PBN relay neurons. Therefore, the authors

should clarify if the PBN is an essential input for the tanycyte-mediated reduction of food intake in response to environmental warmth. Several experiments may be considered, as below:

- If the PBN is an essential input for tanycyte to reduce food intake upon environmental warmth, does inactivating the PBN rescue the anorexic effect during warm exposure?
- The authors may consider testing whether the activity of tanycytes is decreased (by Fos experiments, for example) or the level of VEGFA is lowered when the PBN is inactivated during warm exposure. Such experiments could provide important support for the authors' claim that tanycytes receive input from thermosensory PBN neurons.'

Thank you for these questions. We believe there is a misunderstanding here since we did not detect TRPV1 channels in tanycytes. Instead, we find the presence of mRNA for TRPV2, which are activated only by noxious heat (>50 °C). Moreover, we would caution that recent imaging data suggest the brain to "heat up" to close to 40 °C (and the hypothalamus in particular) when neuronal activity is high (Rzechorzek N.M. *et al.* A daily temperature rhythm in the human brain predicts survival after brain injury. *Brain* 145, 2031–2048 (2022)). Physiologically, it seems inconceivable if all the TRPV channels would then be activated because they could not perform their putative roles in, e.g., modulating endocannabinoid-mediated synaptic neurotransmission (e.g., Maccarrone M. *et al.* Programming of neural cells by (endo)cannabinoids: from physiological rules to emerging therapies. *Nat. Rev. Neurosci.* 15, 786–801 (2014); Maccarrone M. *et al.* Anandamide inhibits metabolism and physiological actions of 2-arachidonoylglycerol in the striatum. *Nat Neurosci.* 11, 152–9 (2008); Li Y. *et al.* Endocannabinoid activation of the TRPV1 ion channel is distinct from activation by capsaicin. *J. Biol. Chem.* 297, 101022 (2021) and others). We believe that performing acute temperature switching in brain slices has nothing to do with the thermosensing circuitry. Instead, we could gain an 'access point' to manipulating tanycytes in this preparation.

Experimentally, we have:

- **Performed the inhibition experiment** as requested (using TeLC), see above (**Figure 4j,k**),
- Performed **exposure to 38 °C** only, which is below the known activation set point of even TRPV2 channels (in fact, none of the TRPV channels should be active at this temperature), and still **recorded reduced food intake (Extended Data Figure 9j)**,
- Pre-treated mice with Tranilast, a selective TRPV2 antagonist that crosses the blood-brain barrier (See G.L. *et al.* Enhanced nose-to-brain delivery of tranilast using liquid crystal formulations. *J. Control. Release* 325, 1–9 (2020)), and found a lack of drug effect (**Extended Data Figure 9j**).

Thus, we are of the view that *i*) tanycytes are likely protected within the brain from being direct heat sensors, and *ii*) even if they were this mechanism would be downstream to PBN activation, and *iii*) unlikely to rely on TRPV2 activation directly.

Q4: '3. The newly added Fig. 3f-g shows that optogenetic activation of PBN glutamatergic neurons elicits EPSCs in tanycytes. However, the average latency of 250 ms is quite different from the average spike latency of neurons (~10 ms), and the latency of EPSCs is quite variable. The authors should provide explanation for these slow and variable latencies and include this in the manuscript. Moreover, it would be informative to include additional data such as the failure rate of optogenetically-induced EPSCs.

Thank you for requesting these additional details. They can be found in **Extended Data Figure 6m**. We show that the **time-lag**, whilst slow, does not vary excessively (we used a binning plot to show that the bulk of events is close to the end of the light stimulus). Likewise, the **amplitude** of the evoked events is stable. Admittedly, the **failure rate** of these EPSCs is high (only those cells were included that had at least 1 positive response), which we attributed to coupling being activity-dependent and sensitive probably to high-frequency stimulation (alike bursting in the PBN, which however was not tested here).

Referee #4:

Thank you for appreciating our work that had gone into the revised version of this manuscript. In this round, we have again performed additional experiments, as well as introduced clarifications in the manuscript to satisfy your remaining concerns. Our replies to your queries are as follows.

Q1: 'The experimental paradigm used consist in submitting the animals at 40C for 1h. According to the results of the authors, acute heat would rapidly activate glutamatergic PBN neurons that in turn would activate tanycytes thus reducing food intake. The authors show food intake data over 24h post heat exposure, but it would be nice to see a time-course of food intake. Given the rapid circuit activation as a consequence of high temperature, it is plausible to expect that food intake is affected at short time-points.'

Thank you for asking. We have extended our experiments (in view of the question of TRPV channels by yourself and Referee #3) to both 38 °C and 40 °C, and in both cases **measured food intake at 2h, 4h, and 24h. Data on controls were included in Extended Data Figure 1a, while with tranelast (TRPV2 antagonist) and 38 °C are shown in Extended Data Figure 9j.** Indeed, a reduction in food intake, which is more variable than the cumulative value after 24h, is apparent already shortly after heat exposure, which is consistent with the rapid activation of the neurocircuit.

Q2: 'Line 206: I guess you mean Fig 3d.'

Figure designations **were corrected throughout.**

Q3: 'I am not sure why the authors changed the original Extended Fig 8. In the original one they showed the expression levels of diverse markers in tanycytes while now they are included in the category ependymocytes (which is more general).'

We **have reverted to the original figure** given the concerns of the other Referees, and to be coherent with the terminology used in the source paper (Campbell et al., 2017). In both **Extended Data Fig. 8** and in the text, we report a heat map representing fold changes of the selected genes in 'tanycytes 1' vs. all the other cell types listed by Campbell and colleagues. However, in the text we state that we did subset those cells that expressed *Rax*, and thereby focused on genuine tanycytes. Therefore, we believe the presentation of the source data is correct and allows us to make the point that *Rax*⁺ tanycytes are indeed a subset.

Q4: 'Lines 274-283 (Fig 4e). In this paragraph the authors suggest that tanycytes per se are able to sense acute heat, as they express TRPV2 channels and are able to release VEGFa and reduce the excitability of neighbor neurons. Unless this reviewer is missing something, this experimental setting bypasses the PBN glutamatergic input on tanycytes. Then, these results seem to suggest that actually the PBN glutamatergic input on tanycytes is not required to mediate the effects of heat.'

Thank you for this question, which resonates on a concern from Referee 3. We believe there is a misunderstanding here since we did not imply that tanycytes are direct heat sensors. Indeed, our mass spectrometry analysis revealed the presence of TRPV2 in tanycytes. But TRPV2 channels are only activated by noxious heat (>50 °C) (Liu B. & Qin F. Use dependence of heat sensitivity of vanilloid receptor TRPV2. *Biophys. J.* 110, 1523–1537 (2016)). Moreover, recent data show that the brain changes its temperature upon activity (and also diurnally; Rzechorzek N.M. *et al.* A daily temperature rhythm in the human brain predicts survival after brain injury. *Brain* 145, 2031–2048 (2022)), yet the recorded parenchymal temperatures are below the activation set points of TRPV family channels.

We view acute temperature switching in brain slices to have nothing to do with direct temperature sensing or 'bypassing' the thermosensing circuitry. Instead, we exploit an 'access point' to manipulating tanycytes in this preparation.

To exclude the possibility of direct temperature sensing by tanycytes, we have:

- **Performed the inhibition experiment** as requested by Referee 3 (using TeLC; **Figure 4j,k**),
- Performed exposure to 38 °C only, which is below the known activation set point of even TRPV1 channels (in fact, none of the TRPV channels should be active at this point), and still **recorded reduced food intake (Extended Data Figure 9j)**,
- Pre-treated mice with tranilast, a selective TRPV2 antagonist that crosses the blood brain barrier (See G.L. *et al.* Enhanced nose-to-brain delivery of tranilast using liquid crystal formulations. *J. Control. Release* 325, 1–9 (2020), and found a lack of drug effect (**Extended Data Figure 9j**).

Thus, we are of the view that *i*) tanycytes are likely protected within the brain from being direct heat sensors, and *ii*) even if they were this mechanism would be downstream to PBN activation, and *iii*) unlikely to rely on TRPV2 activation directly.

Q5: 'In the mass spectrometry study, did the authors detect VEGFa? This would be a better way to demonstrate it is expressed in tanycytes.'

Our mass spectrometry analysis failed to detect VEGF in primary tanycyte cultures. This is not surprising when considering that roughly 50% of proteins are detected, if ionized, by this method. But we were able to validate VEGFA expression in tanycytes with an anti-VEGFA antibody, further corroborating the scRNA-seq data from Campbell *et al.* (Campbell J.N. *et al.* A molecular census of arcuate hypothalamus and median eminence cell types. *Nat. Neurosci.* 20, 484–496 (2017)). Moreover, we show *in two independent experiments* that VEGFA protein expression in tanycytes is increased by both heat exposure and chemogenetic activation of PBN-glutamatergic neurons (**Extended Data Figure 9b,c**).

Q6: 'The inhibition of VEGFa is a good strategy to demonstrate that is the molecule mediating the acute effects of heat on food intake. Problematically, the experimental approach is not specific for tanycytes. Delivering an siRNA into the ventricle could actually target multiple cell types, including some expressing high levels of VEGFa (e.g., endothelial cells). A specific approach is necessary to conclude that VEGFa derived from tanycytes mediates the acute effects of heat on food intake.'

We appreciate your concern. However, **two sets of data** mitigate this. 1) We have used a tetanus toxin light chain construct (**TeLC-GFP**) to 'silence' tanycytes given that TeLC blocks VAMP2-dependent exocytosis. These experiments led to the same outcome as the siRNA experiment (**Figure 4j,k**). 2) At the same time, we used a tamoxifen inducible TeLC that was *directly fused* with GFP in *Rax-CreER^{T2}* mice. Therefore, we could follow GFP expression throughout the brain. We only found GFP expression in tanycytes lining the wall of the 3rd ventricle, but neither in parenchyma nor in endothelial cells. Thus, and with this 2nd experiment at hand, we are confident to have reduced the likelihood of an 'off-target' bias.

Q7: 'The validation of the model above, should be done by a method measuring protein levels of VEGFa (rather than RNA – that is RNAscope and qPCR).'

We have performed histochemistry using **an anti-VEGF antibody** with data included in **Extended. Data Figure 9b,c**.

Q8: 'The authors use a wide range of tools, and this is a strength of the work. However, sometimes the authors do not assess whether these tools perform as they are supposed. For example, in the TeLC study the authors do not show that TeLC is actually expressed in tanycytes after viral infection. They did not show either that they are able to simultaneously activate PBN neurons and inactivate tanycytes (via c-fos). These kind of controls should be provided for all the experimental strategies.'

Thank you for asking this. Please note that the **TeLC construct we used is fused to GFP**. Therefore, we can use GFP as a reporter for expression/cell type-specific localization throughout. We also have performed the cFos experiment you have requested, including the inactivation of PBN neurons by TeLC, with data included in **Figure 5a-c, Extended Data Fig. 10f,h**. Thus, we ascertained that each experiment is quality controlled.

Q9: 'Fig 5e, f; Ext Fig 10. Why the authors show "time spent eating/drinking" instead of total amount of food eaten or water drunk?'

We have used a **Noldus Phenotyper system** to continuously monitor the animals after heat exposure, and to collect multiple variables. However, this system does not record actual food (g) or water (ml) intake but instead assigns fields of view for the animal to be in direct contact with the food or liquid source. This is why we were limited in expressing the amounts, and use the 'time spent' parameter instead. However, **in the experiments performed during the revision process using, e.g., TeLC injection in the PBN, we have measured food intake (g), and even at 2h, 4h, and 24h time-points (Extended Data Fig. 1a)**. These experiments support and justify that "time spent eating/drinking" is a **sufficiently tight surrogate** of the actual amount of food consumed.

Heat exposure increases the population of TH⁺ neurons (a) and the phosphorylation of TH (b-d) in the arcuate nucleus.

The size of the cFos⁺ cell group was significantly larger, too, even 24h after exposing mice to 40 °C (e). Amongst the cFos⁺ cells, the population of TH⁺ neurons was significantly increased. **p* < 0.05, ***p* < 0.01

Q10: 'In the model that the authors propose, heat would activate PBN glutamatergic neurons that in turn would activate tanycytes and subsequently inhibit orexigenic (Th and/or AgRP neurons) in the ARC. This would mediate the observed reduction in food intake upon heat exposure. However, this latter part of the model (the role of ARC orexigenic neurons) has been neglected, but it is key to "close the circle" to explain how food intake is attenuated. The authors should address this point.'

We appreciate you asking this. However, we feel that the core food intake circuit is sufficiently mapped to allow for our hypothesis to stand considering the electrophysiology data and the behavioral phenotype we demonstrated. We believe that an extensive exploration beyond these parameters

would distract from the main thrust of our manuscript and unnecessarily fragment it. Nevertheless, to meet your expectations, we have taken advantage of a biochemical surrogate of tyrosine hydroxylase (TH) activity, that is its phosphorylation at Ser²¹. This modification is necessary for an increased enzymatic activity (Alpar A *et al.* Hypothalamic CNTF volume transmission shapes cortical

noradrenergic excitability upon acute stress. *EMBO J.* 37, e100087 (2018)). Given that downstream modulation of TH activity mediated by heat induced tanyocyte-derived VEGFA has an anorexigenic effect, and that upstream dopamine efferents inhibit AgRP neurons in the arcuate nucleus, we have performed quantitative histochemistry to test parameters of dopamine neurons. We used 3-4 mice/group with 3 sections each within the area of -1.82 mm to -2.30 mm and determined cell numbers, fluorescence intensity for phosphorylated and total TH, and cFos. As our data show (see *attached figure overleaf*), the number of TH⁺ neurons is increased upon heat exposure (noting that similar increases have been seen by psychostimulants, cannabinoids). Not only that but TH phosphorylation was also increased. These data suggest increased dopamine production in a larger contingent of TH⁺ neurons in the arcuate nucleus. Notably, cFos localization was also elevated in TH⁺ neurons in mice exposed to 40 °C 24 h prior. Thus, we suggest that increased dopamine output could indeed effectively inhibit AgRP neurons, thus lending mechanistic support within the local arcuate circuit to our findings.

Q11: 'Some images and quantifications are difficult to interpret. For example, Fig 2j/j1. What is within the inset (square)? Is this a tanyocyte? Does it stain for vimentin? Similarly, in Fig 4a-b the authors quantify VEGFa puncta in a-tanycytes. How do they know they are alpha? Because of their location? Is the number of puncta referred as to the average in a single cell? Or in the whole area shown in the image? The same for Fig 4g.'

Thank you for asking these details. We have moved the arrowheads on Fig. 2j₁ given that the intended meaning is that vimentin⁺ structures were innervated by mCherry⁺ varicose axons. One of those is identified in the present version. Tanyocyte subtypes are indeed inferred by location but also by GluA2 expression. We **revised the figure legends** to improve clarity.

Q12: Extended Fig 1C: x-axis lacks units.

The axis label has been added.

Q13: 'Line 336.C-fos expression is not a feature of only a-tanycytes. In fact, in the response letter, the authors acknowledge that other tanyocyte populations are activated although the alpha type represents the majority.'

We have **replaced the word 'only' with 'preferentially' upon heat manipulation** to be entirely correct.

Q14: 'Line 361. This is an overstatement. Authors did not show specificity of release from tanycytes.'

Indeed, the original statement was an interpretation only. However, **with the TeLC experiments at hand, we can now justify the statement that it is exocytosis that indeed occurs in tanycytes.**

Q15: 'Line 364. Truncated sentence.'

Thank you for drawing our attention to this error. The **sentence has been corrected.**

Q16: 'Considering the findings of the authors, I would suggest changing the title. First, the authors do not show an endocrine mechanism (perhaps paracrine would be more accurate). Also, I think that narrowing the findings to only alpha-tanycytes is a step too far, as other types of tanycytes may be involved. Insights on the circuit engaged would be also informative.'

After a candid discussion with Dr. Rowland, we have simplified the title which addresses both of your queries, and reads: '**A brain stem-hypothalamus neurocircuit links environmental heat sensing to reduced food intake in mice.**'

Reviewer Reports on the Second Revision:

Referees' comments:

Referee #3 (Remarks to the Author):

After substantial revision, the manuscript has been notably improved and all previously raised concerns have been satisfactorily addressed, leaving no remaining questions for this reviewer except for a minor clarification:

Line 344-345 : Figure 5C is associated with the amount of food intake, not body weight. Please correct the sentence accordingly.

Referee #4 (Remarks to the Author):

I would like to thank the authors for their efforts in revising the study in response to the comments of this reviewer. Most of my concerns have been satisfactorily addressed and the new additions have significantly improved the manuscript, providing important insights to the story. The response letter also addresses my comments adequately. Nevertheless, I would like to raise few points that need the attention of the authors.

- Some of the images included are not compelling. For example, Fig 2j, j1. The c-fos positive cell that is highlighted, does not seem to express vimentin and their location is not consistent with tanycytes (thought it could have been migrated). Another example is Extended Fig 9g, the cell that is highlighted by the arrow does not seem to express Flt1. Furthermore, in Figure Extended Fig 9i it seems that Flt1 is not express at all in the whole section, while in the previous one it can be seen that other ARC cells express it suggesting perhaps that the immunofluorescence study did not properly work. A third example can be found in Fig 5 e, f. In these figures the showed field is so limited that it makes very difficult to assess if the staining is specific or just background.

- The data shown in lines 238-239 does not seem to fit with the graphs in Fig 3g1 (in particular the time).

- In line 337, the authors mention that heat induced a “quasi-equivalent c-fos expression...”.

However, this assertion is based on the visual inspection of one image. The authors should provide actual quantifications. Similarly, the data in Fig 5b should be quantified rather than just showing an image.

- Response letter Q10. I agree that the ARC core food intake circuit is well mapped, but I do not think that this is a sufficient reason to superficially address this aspect in the context of this research.

There is no functional data indicating that VEGFA modulate the aactivity of TH or AgRP neurons, only anatomical evidence of the expression of the receptor. So it is not known if VEGFA “affected” the activity of these neurons as suggested in line 410. This should be rephrased. Finally, in the response letter the authors state that “upstream dopamine efferents inhibit AgRP neurons in the ARC...”. This should be substantiated by the authors as previous reports suggest tha dopamine inhibits POMC neurons but excites AgRP neurons.

Author Rebuttals to Second Revision:

Point-by-point responses to the Referees' concerns

(manuscript ID: Nature-2022-08-13268C)

Referee #3:

Thank you for appreciating our thorough revision, and considering the manuscript as *proved*, with all *'concerns satisfactorily addressed'*. Please find our response to your minor clarification as follows.

Q1: 'Line 344-345 : Figure 5C is associated with the amount of food intake, not body weight. Please correct the sentence accordingly.'

The sentence has been corrected to accurately reflect the data.

Referee #4:

We very much appreciate the Referee's support of the revised version of our manuscript, and recognizing the amount of effort that had gone into satisfying the various points of concern. We were also glad to learn that the additional data placed in the response letter were found of interest. At the same time, we thank the Referee for their remaining points, which we have addressed as follows.

Q1: 'Some of the images included are not compelling. For example, Fig 2j, j1. The c-fos positive cell that is highlighted, does not seem to express vimentin and their location is not consistent with tanycytes (thought it could have been migrated). Another example is Extended Fig 9g, the cell that is highlighted by the arrow does not seem to express Flt1. Furthermore, in Figure Extended Fig 9i it seems that Flt1 is not express at all in the whole section, while in the previous one it can be seen that other ARC cells express it suggesting perhaps that the immunofluorescence study did not properly work. A third example can be found in Fig 5 e, f. In these figures the showed field is so limited that it makes very difficult to assess if the staining is specific or just background.'

Fig. 1 – Histochemical data related to 'Fig. 5e,f'. Please note the specificity of both genetic tags in larger fields of views. *Arrowheads* in 'a' point to the ventricular layer of genetically-tagged tanycytes. *Abbreviations*: 3V, 3rd ventricle, scp, superior cerebellar peduncle. *Scale bars* = 50 μ m.

a) We have replaced Fig. 2j,j₁ with images that unambiguously identify a vimentin⁺/Fos⁺ tanycyte. We included Hoechst 33,342 nuclear labeling to identify the soma of the tanycyte, and to also define their cell-dense ventricular layer. Arrows in Fig. 2j pinpoint putative contacts along several tanycyte processes. We have also replaced the inset (Fig. 2j₁) that

focuses on the apical process of the tanycyte in Fig. 2j, which is apposed by genetically tagged varicosities, conforming to **putative contacts**.

b) It is indeed difficult to find the color balance when showing panoramic overviews of *in situ* hybridization, as you have requested. With all due respect, all colors were visible, albeit their balance was less than optimal in the images. Yet, we have **replaced both Extended Data Fig. 9g and 9i** to show that the staining protocols had worked appropriately, with the data supporting our conclusions. Even more so, another high-power photomicrograph was placed into **Extended Data Fig. 9g**.

c) Thank you for requesting larger views of **Fig. 5e,f**. Please note that we have had much larger fields of views for both images in the original and earlier revised versions of the manuscript (see **Fig. 1** here), but then had to reduce their sizes. Please note that these images show the distribution of genetic tags and are not histochemical images. Moreover, we have additional controls for specificity (e.g., panoramic views in **Extended Data Fig. 10f,h**) for the same constructs. Therefore, we are of the view that the specificity of both constructs is sufficiently assured and documented.

Q2: 'The data shown in lines 238-239 does not seem to fit with the graphs in Fig 3g1 (in particular the time).'

Thank you for this point. Data in the text were expressed as **means \pm s.e.m.**, whereas the box and whisker plot in Fig. 3g,g1 showed **medians \pm interquartile ranges**. This difference was made clear in the figure legend, and also in the statistics description of the methods section. Moreover, we have **corrected the figure to include all the observations ($n = 50$) that were analyzed and part of the source data file**.

Q3: 'In line 337, the authors mention that heat induced a "quasi-equivalent c-fos expression...". However, this assertion is based on the visual inspection of one image. The authors should provide actual quantifications. Similarly, the data in Fig 5b should be quantified rather than just showing an image.'

Thank you for asking these additional data. **Data on cFOS expression in the PBN were inserted in the text (lines 338-339)** due to space constraints in the relevant figure. For **Fig. 5b**, quantitative data were inserted, as per your request. These data lend additional support to our conclusions.

Q4: 'Response letter Q10. I agree that the ARC core food intake circuit is well mapped, but I do not think that this is a sufficient reason to superficially address this aspect in the context of this research. There is no functional data indicating that VEGFA modulate the activity of TH or AgRP neurons, only anatomical evidence of the expression of the receptor. So it is not known if VEGFA "affected" the activity of these neurons as suggested in line 410. This should be rephrased. Finally, in the response letter the authors state that "upstream dopamine efferents inhibit AgRP neurons in the ARC...". This should be substantiated by the authors as previous reports suggest tha dopamine inhibits POMC neurons but excites AgRP neurons.'

Thank you for these comments. As have stated in our earlier response, we think that it is not only anatomical data but also *ex vivo* electrophysiology and pharmacology results were presented to link VEGFA release from tanycytes to the altered activity of hypothalamic (ARC) neurons. We showed that the spike threshold of ARC neurons significantly increases in the hypothalamus at elevated temperatures. The physiological sign of this change shall be reduced action potential firing. VEGFA receptor involvement was shown pharmacologically, by using Axitinib. Together with the ultrastructural data on tanycytes preferentially contacting TH⁺ and AgRP⁺ neurons, we are of the view that our data cumulatively support a sequence of heat-induced PBN inputs stimulate tanycytes → increased VEGFA expression in and release from tanycytes → increased spike threshold → reduced activity of TH⁺ and AgRP⁺ neurons, which is compatible with lowered food intake transiently.

The data on TH⁺ neurons (increased phosphorylation and cFos expression) we have presented previously were interpreted as an adaptive response to thermal stimulation. We have cautioned that the data are from 24h after treatment, thus likely suggesting the role of dopamine neurons to reset the food intake circuit to its physiological optimum after the heat (VEGFA)-induced inhibition of its orexigenic components.

You are correct; Zhang & van den Pol (2016, Ref.⁶⁵) have shown that a population of dopamine neurons in the ARC differentially affects POMC vs. AgRP neurons. Optogenetic and electrophysiology data together showed that local TH neurons in the ARC inhibited POMC neurons and excited AgRP neurons. Food deprivation in their paper increased cFos expression in TH⁺ neurons. This latter finding is compatible with our above suggestion that increased cFos expression in TH⁺ neurons at a delayed time point could be reflective of an adaptive cellular response to restart feeding.

What was misspelled in our response and data interpretation is that we contextualized the changes of the food intake circuit with the VEGFA effect (that is, increased firing threshold) in mind. Accordingly, the suggestion can be put forward that the reduced excitability of TH⁺ neurons after heat (VEGFA) exposure lessens their stimulatory drive of AgRP⁺ neurons. Using 'inhibition' was incorrect at this point. At the same time, the inhibition of POMC neurons by TH⁺ neurons (Ref.⁶⁵) might also be reduced, inferring that the net output of the food intake circuit will shift towards imposing an orexigenic phenotype (that is, phasic reduction in food intake). This is exactly what we have observed.

Given the above, **we thank you for asking us to rephrase line 410, which has been done to communicate our findings clearly and correctly.**

Reviewer Reports on the Third Revision:

Referees' comments:

Referee #4 (Remarks to the Author):

The authors have addressed all the concerns of this reviewer. I thank the authors for their comments and extra experiments they performed.